# Revisiting Area Convexity: Faster Box-Simplex Games and Spectrahedral Generalizations

**Arun Jambulapati**
Simons Institute*
jmblpati@berkeley.edu

**Kevin Tian**
University of Texas at Austin†
kjtian@cs.utexas.edu

## Abstract

We investigate different aspects of area convexity [She17], a mysterious tool introduced to tackle optimization problems under the challenging $\ell_\infty$ geometry. We develop a deeper understanding of its relationship with more conventional analyses of extragradient methods [Nem04, Nes07]. We also give improved solvers for the subproblems required by variants of the [She17] algorithm, designed through the lens of relative smoothness [BBT17, LFN18].

Leveraging these new tools, we give a state-of-the-art first-order algorithm for solving box-simplex games (a primal-dual formulation of $\ell_\infty$ regression) in a $d \times n$ matrix with bounded rows, using $O(\log d \cdot \epsilon^{-1})$ matrix-vector queries. Our solver yields improved runtimes for approximate maximum flow, optimal transport, min-mean-cycle, and other basic combinatorial optimization problems. We also develop a near-linear time algorithm for a matrix generalization of box-simplex games, capturing a family of problems closely related to semidefinite programs recently used as subroutines in robust statistics and numerical linear algebra.

## 1 Introduction

Box-simplex, or $\ell_\infty$-$\ell_1$, games are a family of optimization problems of the form

$$\min_{x \in [-1,1]^n} \max_{y \in \Delta^d} x^\top \mathbf{A} y - b^\top y + c^\top x. \tag{1}$$

These problems are highly expressive, as high-accuracy solutions capture linear programs (LPs) as a special case [LS15].[3] More recently, approximation algorithms for the family (1) have been applied to improve runtimes for applications captured by LPs naturally bounded in the $\ell_\infty$ geometry. The current state-of-the-art first-order method for (1) is by Sherman [She17], who used the resulting solver to obtain a faster algorithm for the maximum flow problem. As further consequences, solvers for (1) have sped up runtimes for optimal transport [JST19], a common task in modern computer vision [KPT⁺17], and min-mean-cycle [AP20], a basic subroutine in solving Markov decision processes [ZP96], among other combinatorial optimization problems [AJJ⁺22, JJST22].

The runtimes of first-order methods for approximating (1) are parameterized by an additive accuracy $\epsilon > 0$ and a Lipschitz constant[4] $L := \|\mathbf{A}\|_{1 \to 1}$; in several prominent applications of (1) where the relevant $\mathbf{A}$ has favorable structure, e.g. the column-sparse edge-vertex incidence matrix of a graph, $L$ is naturally small. Standard non-Euclidean variants of gradient descent solve a smoothed variant of (1) using $\widetilde{O}(L^2 \epsilon^{-2})$ matrix-vector products with $\mathbf{A}$ and $\mathbf{A}^\top$ [Nes05, KLOS14]. Until

---

*Work completed at the University of Washington.

†Work completed at Microsoft Research.

[3]This follows as (1) generalizes $\ell_\infty$ regression $\min_{x \in [-1,1]^n} \|\mathbf{A}x - b\|_\infty$, see Section 3.1 of [ST18].

[4]See Section 1.3 for notation used throughout the paper. $\|\mathbf{A}\|_{1 \to 1}$ is the largest $\ell_1$ norm of any column of $\mathbf{A}$.

37th Conference on Neural Information Processing Systems (NeurIPS 2023).

recently however, the lack of accelerated algorithms for (1) (using a quadratically smaller $\widetilde{O}(L\epsilon^{-1})$ matrix-vector products) was a notorious barrier in optimization theory. This barrier arises due to the provable lack of a strongly convex regularizer in the $\ell_\infty$ geometry with dimension-independent additive range [ST18],[5] a necessary component of standard non-Euclidean acceleration schemes.

Sherman's breakthrough work [She17] overcame this obstacle and gave an algorithm for solving (1) using roughly $O(L\log(d)\log(L\epsilon^{-1})\cdot\epsilon^{-1})$ matrix-vector products.[6] Roughly speaking, Sherman exploited the primal-dual nature of the problem (1) to design a smaller regularizer over the $\ell_\infty$ ball $[-1,1]^n$, which satisfied a weaker condition known as *area convexity* (rather than strong convexity). The algorithm in [She17] combined an extragradient method "outer loop" requiring $O(L\log d\cdot\epsilon^{-1})$ iterations, and an alternating minimization subroutine "inner loop" solving the subproblems required by the outer loop in $O(\log(L\epsilon^{-1}))$ steps. However, Sherman's analysis in [She17] was quite ad hoc, and its relationship to more standard optimization techniques has remained mysterious. These mysteries have made further progress on (1) and related problems challenging in several regards.

1. Sherman's algorithm pays a logarithmic overhead in the complexity of its inner loop to solve subproblems to high accuracy. Does it tolerate more approximate subproblem solutions?

2. The analysis of Sherman's alternating minimization scheme relies on multiplicative stability properties of the dual variable $y$. In generalizations of (1) where the dual variable is unstable, how do we design solvers handling instability to maintain fast subproblem solutions?

3. The relationship between the area convexity definition [She17] and more standard Bregman divergence domination conditions used in classical analyses of extragradient methods [Nem04, Nes07] is unclear. Can we unify these extragradient convergence analyses?

A central goal of this work is to answer Question 3 by revisiting the area convexity technique of [She17], and developing a deeper understanding of its relationship to more standard tools in optimization theory such as relative smoothness [BBT17, LFN18] and the classical extragradient methods of [Nem04, Nes07]. We summarize these insights in Section 2 and give self-contained expositions in Appendices A and B of the supplement. As byproducts, our improved understanding of area convexity results in new state-of-the-art runtimes for (1) by affirmatively answering Question 1, and the first accelerated solver for a matrix generalization by affirmatively answering Question 2.

## 1.1 Our results

**Box-simplex games.** We give an algorithm for solving (1) improving the runtime of [She17] by a logarithmic factor, by removing the need for high-accuracy inner loop solutions.[7]

**Theorem 1.** *There is an algorithm (stated in full as Algorithm 4 in the supplement) deterministically computing an $\epsilon$-approximate saddle point to* (1) *in time*

$$O\left(\mathrm{nnz}(\mathbf{A})\cdot\frac{\|\mathbf{A}\|_{1\to1}\log d}{\epsilon}\right).$$

We define nnz and $\|\mathbf{A}\|_{1\to1}$ in Section 1.3. The most direct comparison to Theorem 1 is the work [She17], which we improve. All other first-order methods for solving (1) we are aware of are slower than [She17] by at least an $\approx\min(\epsilon^{-1},\sqrt{n})$ factor. Higher-order methods for (1) (e.g. interior point methods) obtain improved dependences on $\epsilon$ at the cost of polynomial overhead in the dimension.

As a byproduct of Theorem 1, we improve all the recent applications of (1), including approximate maximum flow [She17], optimal transport [JST19], min-mean-cycle [AP20], and semi-streaming variants of bipartite matching and transshipment [AJJ+22]. These are summarized in Section 5 of the supplement. The design of efficient approximate solvers for optimal transport in particular is a problem which has received a substantial amount of attention from the learning theory community

---

[5]For a function $f$, we call $\max f-\min f$ its additive range.

[6]The runtime for solving subproblems required by [She17] was not explicitly bounded in that work, but a simple initialization strategy bounds the runtime by $O(\log(L\epsilon^{-1}))$, see Proposition 1 of [AJJ+22].

[7]This statement assumes direct access to the entries of $\mathbf{A}$ (as does [She17]). More generally, letting $\mathcal{T}_{\mathrm{mv}}$ be the time it takes to multiply a vector by the argument, we may replace $\mathrm{nnz}(\mathbf{A})$ by $\max(\mathcal{T}_{\mathrm{mv}}(\mathbf{A}),\mathcal{T}_{\mathrm{mv}}(\mathbf{A}^\top),\mathcal{T}_{\mathrm{mv}}(|\mathbf{A}|),\mathcal{T}_{\mathrm{mv}}(|\mathbf{A}^\top|))$ where $|\cdot|$ is entrywise, which may be a significant savings when $\mathbf{A}$ is implicit and nonnegative. An $\epsilon$-approximate saddle point is a point with duality gap $\epsilon$; see Section 1.3.

(see Table 1 and the survey [PC19]). The runtime of Theorem 1 represents a natural conclusion to this line of work.[8] We believe our improvement in Theorem 1 (avoiding a high-accuracy subproblem solve) takes an important step towards bridging the state-of-the-art in the theory and practice of optimal transport, a well-motivated undertaking due to its widespread use in applications.

Table 1: **Runtime complexities of first-order algorithms for optimal transport.** Stated for a $d \times d$ cost matrix with unit-bounded costs, additive error tolerance $\epsilon$. Results labeled "sequential" require a parallel depth which scales polynomially in the problem dimension $d$.

| Method | Runtime | Comments |
|---|---|---|
| [AWR17] | $O(d^2 \log d \cdot \epsilon^{-3})$ | |
| [DGK18] | $O(d^2 \log d \cdot \epsilon^{-2})$ | |
| [DGK18, LHJ19] | $O(d^{2.5}\sqrt{\log d} \cdot \epsilon^{-1})$ | |
| [BJKS18, Qua20] | $O(d^2 \log(d) \log(\epsilon^{-1}) \cdot \epsilon^{-1})$ | sequential, randomized |
| [LMR19] | $O(d^2 \cdot \epsilon^{-1} + d \cdot \epsilon^{-2})$ | sequential |
| [JST19] | $O(d^2 \log(d) \log(\epsilon^{-1}) \cdot \epsilon^{-1})$ | |
| [LHJ22] | $O(d^{7/3}\sqrt[3]{\log d} \cdot \epsilon^{-4/3})$ | |
| Theorem 1 | $O(d^2 \log d \cdot \epsilon^{-1})$ | |

**Box-spectraplex games.** We initiate the algorithmic study of box-spectraplex games of the form

$$\min_{x \in [-1,1]^n} \max_{\mathbf{Y} \in \Delta^{d \times d}} \left\langle \mathbf{Y}, \sum_{i \in [n]} x_i \mathbf{A}_i \right\rangle - \langle \mathbf{B}, \mathbf{Y} \rangle + c^\top x, \tag{2}$$

where $\{\mathbf{A}_i\}_{i \in [n]}$, $\mathbf{B}$ are $d \times d$ symmetric matrices and $\mathbf{Y} \succeq \mathbf{0}$ satisfies $\mathrm{Tr}(\mathbf{Y}) = 1$ (i.e. $\mathbf{Y} \in \Delta^{d \times d}$, cf. Section 1.3). This family of problems is a natural formulation of semidefinite programming (SDP), and generalizes (1), the special case of (2) where all of the matrices $\{\mathbf{A}_i\}_{i \in [n]}$, $\mathbf{B}$, $\mathbf{Y}$ are diagonal.

Our main result on solving (2), stated below as Theorem 2, has already found use in [JRT23] to obtain faster solvers for spectral sparsification and related discrepancy-theoretic primitives, where existing approximate SDP solvers do not apply. Beyond the fundamental nature of the problem solved by Theorem 2, we are optimistic that it will find use in other applications of structured SDPs.

To motivate our investigation of (2), we first describe some history of the study of the simpler problem (1). Many applications captured by (1) are structured settings where optimization is at a *relative scale*, i.e. we wish to obtain an $(1 + \epsilon)$-multiplicative approximation to the value of a LP, $\mathrm{OPT} := \min_{\mathbf{A}x \leq b} c^\top x$. We use bipartite matching as an example: $x$ is a fractional matching and $\mathbf{A}$ is a nonnegative edge-vertex incidence matrix. For LPs with a nonnegative constraint matrix $\mathbf{A}$, custom positive LP solvers achieve $(1 + \epsilon)$-multiplicative approximations at accelerated rates scaling as $\widetilde{O}(\epsilon^{-1})$. This result then implies a $\widetilde{O}(\epsilon^{-1})$-rate algorithm for approximating bipartite matching and its generalization, optimal transport (where the first such algorithm used positive LP solvers [BJKS18, Qua20]). However, existing accelerated positive LP solvers [ZO15] are both sequential and randomized, and whether this is necessary has persisted as a challenging open question.

In several applications with nonnegative constraint matrices [JST19, AP20, AJJ+22], this obstacle was circumvented by recasting the problem as a box-simplex game (1), for which faster, deterministic, and efficiently-parallelizable solvers exist (see e.g. Section 4.1 of [AJJ+22] for the bipartite matching reduction).[9] In these cases, careful use of box-simplex game solvers (and binary searching for the problem scale) match the guarantees of positive LP solvers, with improved parallelism or determinism. Notably, simpler primitives such as simplex-simplex game solvers do not apply in these settings.

The current state-of-affairs in applications of fast SDP solvers is very similar. For various problems in robust statistics (where the goal is to approximate a covariance matrix or detect a corruption

---

[8]Progress on the historically simpler problem of (simplex-simplex) matrix games, a relative of (1) where both players are $\ell_1$ constrained, has stalled since [Nem04] almost 20 years ago, which attains a deterministic runtime of $O(\mathrm{nnz}(\mathbf{A}) \cdot L \log d \cdot \epsilon^{-1})$ where $L$ is the appropriate Lipschitz constant, analogously to Theorem 1.

[9]Two notable additional problems captured by box-simplex games, maximum flow [She17] and transshipment [AJJ+22], do not have a nonnegative constraint matrix (as flows are signed), so positive LP solvers do not apply.

spectrally) [CDG19, CMY20, JLT20] and numerical linear algebra (where the goal is to reweight or sparsify a matrix sum) [LS17, CG18, JLM$^+$21], positive SDP solvers have found utility. However, these uses appear quite brittle: current positive SDP solvers [ZLO16, PTZ16, JLT20] only handle the special case of packing SDPs (a special class of positive SDP with one-sided constraints), preventing their application in more challenging settings (including pure covering SDPs).

In an effort to bypass this potential obstacle when relying on positive SDP solvers more broadly, we therefore develop a nearly-linear time solver for box-spectraplex games in Theorem 2 (we define $\mathcal{T}_{\mathrm{mv}}$, the time required to perform a matrix-vector product, in Section 1.3).

**Theorem 2.** *There is an algorithm which computes an $\epsilon$-approximate saddle point to* (2) *in time*

$$O\left(\left(\mathcal{T}_{\mathrm{mv}}(\mathbf{B}) + \sum_{i \in [n]} \left(\mathcal{T}_{\mathrm{mv}}(\mathbf{A}_i) + \mathcal{T}_{\mathrm{mv}}(|\mathbf{A}_i|)\right)\right) \cdot \frac{L^{3.5} \log^3\left(\frac{Lnd}{\delta\epsilon}\right)}{\epsilon^{3.5}}\right),$$

*with probability $\geq 1 - \delta$, for $L := \|\sum_{i \in [n]} |\mathbf{A}_i|\|_{\mathrm{op}} + \|\mathbf{B}\|_{\mathrm{op}}$. A deterministic variant uses time*

$$O\left(\left(\left(\mathcal{T}_{\mathrm{mv}}(\mathbf{B}) + \sum_{i \in [n]} \left(\mathcal{T}_{\mathrm{mv}}(\mathbf{A}_i) + \mathcal{T}_{\mathrm{mv}}(|\mathbf{A}_i|)\right)\right) \cdot d + d^\omega\right) \cdot \frac{L \log\left(\frac{Lnd}{\delta\epsilon}\right)}{\epsilon}\right).$$

Theorem 2 follows as a special case of Theorem 3 in the supplement, which has refined guarantees stated in terms of the cost to perform certain queries of matrix exponentials required by our algorithm, typical of first-order methods over $\Delta^{d \times d}$. The first runtime in Theorem 2 implements these queries using randomized sketching tools, and the second uses an exact implementation. Beyond its generic utility, we find Theorem 2 conceptually interesting, as its development used our relative smoothness viewpoint on area convexity in Section 2 (for overcoming instability of dual matrix variables), and new techniques for proving area convexity of a matrix variant of the [She17] regularizer.

To put Theorem 2 in context, we compare it to existing solvers for simplex-spectraplex games, a relative of (2) where both players are $\ell_1$ constrained. Simplex-spectraplex games are more well-studied [KV05, WK06, AK07, BBN13, AL17, CDST19], and the state-of-the-art algorithms [BBN13, AL17, CDST19] query $\widetilde{O}(L^{2.5}\epsilon^{-2.5})$ vector products in $\{\mathbf{A}_i\}_{i \in [n]}$ and $\mathbf{B}$. A slight variant of these algorithms (using a separable regularizer in place of area convex techniques) gives a complexity of $\widetilde{O}(n \cdot L^{2.5}\epsilon^{-2.5})$ such queries for the more challenging box-spectraplex games (2), which is the baseline in the literature. In comparison, Theorem 2 requires $\widetilde{O}(L^{3.5}\epsilon^{-3.5})$ products, but assumes access to $\{|\mathbf{A}_i|\}_{i \in [n]}$. This is not without loss of generality, as $|\mathbf{A}|$ can be dense even when $\mathbf{A}$ is sparse, though in important robust statistics or spectral sparsification applications (where $\{\mathbf{A}_i\}_{i \in [n]}$ are rank-one), it is not restrictive. We find it interesting to understand whether access to $\{|\mathbf{A}_i|\}_{i \in [n]}$ and the $\widetilde{O}(L\epsilon^{-1})$ overhead in Theorem 2 (due to higher-rank randomized sketches) can be avoided.

## 1.2 Related work

The family of box-simplex games (1) is captured by linear programming [LS15], where state-of-the-art solvers [CLS21, vdBLL$^+$21] run in time $\widetilde{O}((n+d)^\omega)$ or $\widetilde{O}(nd + \min(n,d)^{2.5})$, where $\omega \approx 2.37$ is the current matrix multiplication constant [AW21, DWZ22]. These LP solvers run in superlinear time, and practical implementations do not currently exist; on the other hand, the convergence rates depend polylogarithmically on the accuracy parameter $L\epsilon^{-1}$. The state-of-the-art approximate solver for (1) (with runtime depending linearly on the input size $\mathrm{nnz}(\mathbf{A})$ and polynomially on $L\epsilon^{-1}$) is the accelerated algorithm of [She17], which is improved by our Theorem 1 by a logarithmic factor. Finally, we mention [BSW19] as another recent work which relies on area convexity techniques to design an algorithm in a different problem setting; their inner loop also requires high-precision solves (as in [She17]), and this similarly results in a logarithmic overhead in the runtime.

For the specific application of optimal transport, a problem captured by (1) receiving significant recent attention from the learning theory community, several alternative solvers have been developed beyond Table 1. By exploiting relationships between Sinkhorn regularization and faster solvers for matrix scaling [CMTV17, ALdOW17], [BJKS18] gave an alternative solver obtaining a $\widetilde{O}(n^2 \cdot \epsilon^{-1})$ rate. These algorithms call specialized graph Laplacian system solvers as a subroutine, which are currently

less practical than our methods based on matrix-vector queries. Finally, the recent breakthrough maximum flow algorithm of [CKL$^+$22] extends to solve optimal transport in $O(n^{2+o(1)})$ time (though its practicality is unclear). For moderate $\epsilon \geq n^{-o(1)}$, this is slower than Theorem 1.

To our knowledge, there have been no solvers developed tailoring specifically to the family (2). These problems are solved by general-purpose SDP solvers, where the state-of-the-art runtimes of $\widetilde{O}(n^\omega \sqrt{d} + nd^{2.5})$ or $\widetilde{O}(n^\omega + d^{4.5} + n^2\sqrt{d})$ [JKL$^+$20, HJS$^+$22] are highly superlinear (though again depend polylogarithmically on the inverse accuracy). We believe our techniques in proving Theorem 2 extend to show that a variant of gradient descent in the $\ell_\infty$ geometry [KLOS14] solves (2) in an unaccelerated $\widetilde{O}(L^2\epsilon^{-2})$ iterations, with the same per-iteration complexity as Theorem 2.

## 1.3 Notation

**General notation.** We use $\widetilde{O}$ to suppress polylogarithmic factors in problem parameters for brevity. Throughout $[n] := \{i \in \mathbb{N} \mid 1 \leq i \leq n\}$ and $\|\cdot\|_p$ is the $\ell_p$ norm. We refer to the dual space (bounded linear operators) of a set $\mathcal{X}$ by $\mathcal{X}^*$. The all-zeroes and all-ones vectors in dimension $d$ are denoted $\mathbb{0}_d$ and $\mathbb{1}_d$. When $u, v$ are vectors of equal dimension, $u \circ v$ denotes their coordinatewise multiplication. For $\mathbf{A} \in \mathbb{R}^{n \times d}$ we denote its $i^{\text{th}}$ row (for $i \in [n]$) by $\mathbf{A}_{i:}$ and $j^{\text{th}}$ column (for $j \in [d]$) by $\mathbf{A}_{:j}$. For $p, q \geq 1$ we define $\|\mathbf{A}\|_{p \to q} := \sup_{\|v\|_p = 1} \|\mathbf{A}v\|_q$. The number of nonzero entries in $\mathbf{A}$ is denoted $\mathrm{nnz}(\mathbf{A})$, and $\mathcal{T}_{\mathrm{mv}}(\mathbf{A})$ is the time it takes to multiply a vector by $\mathbf{A}$. We use $\mathrm{med}(x, -1, 1)$ to project $x$ onto $[-1, 1]$, where med means median. We define the $d$-dimensional simplex and the $d \times d$ spectraplex $\Delta^d := \{y \in \mathbb{R}^d_{\geq 0} \mid \|y\|_1 = 1\}$ and $\Delta^{d \times d} := \{\mathbf{Y} \in \mathbb{S}^{d \times d}_{\succeq \mathbf{0}} \mid \mathrm{Tr}\mathbf{Y} = 1\}$. Throughout we reserve $h(y) := \sum_{j \in [d]} y_j \log y_j$ (for $y \in \Delta^d$) for negated (vector) entropy.

**Optimization.** For convex function $f$, $\partial f(x)$ refers to the subgradient set at $x$; we sometimes use $\partial f$ to denote any (consistently chosen) subgradient. Following [LFN18], we say $f$ is $L$-relatively smooth with respect to convex function $r$ if $Lr - f$ is convex, and we say $f$ is $m$-strongly convex with respect to $r$ if $f - mr$ is convex. For a differentiable convex function $f$ we define the associated Bregman divergence $V_x^f(x') := f(x') - f(x) - \langle \nabla f(x), x' - x \rangle$, which satisfies the identity

$$\langle \nabla V_x^f(x'), u - x' \rangle = \langle \nabla f(x') - \nabla f(x), u - x' \rangle = V_x^f(u) - V_{x'}^f(u) - V_x^f(x'). \qquad (3)$$

For convex function $f$ on two variables $(x, y)$, we use $\partial_y f(x, y)$ to denote the subgradient set at $y$ of the restricted function $f(x, \cdot)$. We call a function $f$ of two variables $(x, y) \in \mathcal{X} \times \mathcal{Y}$ convex-concave if its restrictions to the first and second block are respectively convex and concave. We call $(x, y) \in \mathcal{X} \times \mathcal{Y}$ an $\epsilon$-approximate saddle point if its *duality gap*, $\max_{y' \in \mathcal{Y}} f(x, y') - \min_{x' \in \mathcal{X}} f(x', y)$, is at most $\epsilon$. For differentiable convex-concave function $f$ clear from context, its subgradient operator is $g(x, y) := (\partial_x f(x, y), -\partial_y f(x, y))$; when $f$ is differentiable, we will call this a gradient operator. We say operator $g$ is monotone if for all $w, z$ in its domain, $\langle g(w) - g(z), w - z \rangle \geq 0$: examples are the subgradient of a convex function or subgradient operator of a convex-concave function.

## 1.4 Organization

We summarize our improved area convexity insights in Section 2 (with extended expositions in the supplement), and showcase how they prove Theorem 1 in Section 3. Due to space constraints, we defer proving Theorem 2 and our applications to the supplement (Sections 4 and 5, respectively).

## 2 Revisiting area convexity

In this section and Section 3, we consider *box-simplex* games of the form (1), for $\mathbf{A} \in \mathbb{R}^{n \times d}$, $b \in \mathbb{R}^d$, and $c \in \mathbb{R}^n$. For simplicity we assume $\|\mathbf{A}\|_{1 \to 1} \leq 1$, lifting this assumption (by scale invariance) in the proof of Theorem 1. We approximate saddle points to (1) by using the family of regularizers:

$$r_{\mathbf{A}}^{(\alpha)}(x, y) := \langle |\mathbf{A}|y, x^2 \rangle + \alpha h(y), \text{ where } h(y) := \sum_{j \in [d]} y_j \log y_j \text{ is negative entropy,} \qquad (4)$$

where the absolute value is applied to $\mathbf{A}$ entrywise, and squaring is applied to $x$ entrywise. This family was introduced by [JST19], a minor modification to a similar family given by [She17], and has favorable properties for the geometry present in the problem (1). In the remainder of the paper,

we specialize the notation $\mathcal{X} := [-1, 1]^n$, and for any $v \in \mathbb{R}^n$ we let $\mathbf{\Pi}_{\mathcal{X}}(v) := \mathrm{med}(v, -1, 1)$ be truncation applied entrywise. We also use $\mathcal{Y} := \Delta^d$ and for any $v \in \mathbb{R}^d_{\geq 0}$ we let $\mathbf{\Pi}_{\mathcal{Y}}(v) := v/\|v\|_1$ normalize onto $\mathcal{Y}$. We let $\mathcal{Z} := \mathcal{X} \times \mathcal{Y}$, and define the gradient operator of (1):

$$g(x, y) := \left( \mathbf{A}y + c, b - \mathbf{A}^\top x \right). \tag{5}$$

We refer to the $x$ and $y$ components of $g(x, y)$ by $g^{\mathsf{x}}(x, y) := \mathbf{A}y + c$ and $g^{\mathsf{y}}(x, y) := b - \mathbf{A}^\top x$. Finally, to minimize clutter we denote the Bregman divergence in $r_{\mathbf{A}}^{(\alpha)}$ by $V^{(\alpha)} := V^{r_{\mathbf{A}}^{(\alpha)}}$.

1. Section 2.1 presents our unification of the convergence analysis in [She17] with more standard analyses, by way of interpreting area convexity as Bregman domination (6).

2. Section 2.2 presents several helper results from convex analysis which demonstrates how area convexity implies relative smoothness of the subproblems encountered by [She17], leading to simpler subproblem solvers. This also gives way to developments which generalize to the matrix setting (and are crucial for our proof of Theorem 2).

Ultimately, our proof of Theorem 1 in Section 3 will build upon both of these sets of insights.

## 2.1 Area convexity as Bregman domination

In Appendix B of the supplement, we analyze extragradient algorithms for approximately solving variational inequalities in an operator $g$, i.e. which find $z$ with small $\langle g(z), z - u \rangle$ for all $u$ in the domain (in the minimax optimization setting where $g$ is taken to be the gradient operator, e.g. the operator in (5), this corresponds to duality gap). Specifically, we analyze convergence under the following condition, weaker than previous notions in [She17, CST21].

**Definition 1** (Relaxed relative Lipschitzness). *We say an operator $g : \mathcal{Z} \to \mathcal{Z}^*$ is $\frac{1}{\eta}$-relaxed relatively Lipschitz with respect to $r : \mathcal{Z} \to \mathbb{R}$ if for all $(z, z', z^+) \in \mathcal{Z} \times \mathcal{Z} \times \mathcal{Z}$,*

$$\eta \left\langle g(z') - g(z), z' - z^+ \right\rangle \leq V_z^r(z') + V_{z'}^r(z^+) + V_z^r(z^+).$$

This notion is related to and subsumes the notions of relative Lipschitzness [CST21] and area convexity [She17] which have recently been proposed to analyze extragradient methods, which we define below. In particular, Definitions 2 and 3 were motivated by designing solvers for (1).

**Definition 2** (Relative Lipschitzness [CST21]). *We say an operator $g : \mathcal{Z} \to \mathcal{Z}^*$ is $\frac{1}{\eta}$-relatively Lipschitz with respect to $r : \mathcal{Z} \to \mathbb{R}$ if for all $(z, z', z^+) \in \mathcal{Z} \times \mathcal{Z} \times \mathcal{Z}$,*

$$\eta \left\langle g(z') - g(z), z' - z^+ \right\rangle \leq V_z^r(z') + V_{z'}^r(z^+).$$

**Definition 3** (Area convexity [She17]). *We say convex $r : \mathcal{Z} \to \mathbb{R}$ is $\eta$-area convex with respect to an operator $g : \mathcal{Z} \to \mathcal{Z}^*$ if for all $(z, z', z^+) \in \mathcal{Z} \times \mathcal{Z} \times \mathcal{Z}$, defining $c := \frac{1}{3}(z + z' + z^+)$,*

$$\eta \left\langle g(z') - g(z), z' - z^+ \right\rangle \leq r(z) + r(z') + r(z^+) - 3r(c).$$

*Area convexity is monotone in $\eta$ as the right-hand side is nonnegative for any $z, z', z^+$.*

Relaxed relative Lipschitzness simultaneously generalizes relative Lipschitzness and area convexity. The relationship is obvious for Definition 2, but the generalization of Definition 3 relies on the following simple observation, which has not previously appeared explicitly to our knowledge:

$$r(z) + r(z') + r(z^+) - 3r(c) = V_z^r(z^+) + V_z^r(z') - 3V_z^r(c) \leq V_z^r(z^+) + V_z^r(z'). \tag{6}$$

In Appendix B of the supplement we give a framework unifying previous analyses of [Nem04, Nes07, She17, CST21], possibly of further utility. The realization (6) that area convexity implies a Bregman divergence bound lets us reinterpret the [She17] outer loop to tolerate larger amounts of inexactness in the subproblems, where inexactness is measured in Bregman divergence. We combine our improved outer loop analysis with tools leveraging relative smoothness (to be discussed) to remove alternating minimization from [She17]. We will use the following known fact.

**Fact 1** (Lemma 2, [CST21]). *For convex functions $f, r$, if $f$ is $L$-relatively smooth with respect to $r$, then $\partial f$ is $L$-relatively Lipschitz with respect to $r$.*

We also summarize some useful properties of $r_{\mathbf{A}}^{(\alpha)}$ adapted from [JST19].

**Lemma 1.** *For $\alpha \geq \frac{1}{2}$, $r_{\mathbf{A}}^{(\alpha)}$ is jointly convex; for $\alpha \geq 2$, $r_{\mathbf{A}}^{(\alpha)}$ is $\frac{1}{3}$-area convex with respect to $g$ (5).*

## 2.2 Area convexity as relative smoothness

In [She17], the second property in Lemma 1 was used with an ad hoc outer loop analysis (unified with existing analyses in Appendix B of the supplement) to prove convergence. However, it remains to discuss how to implement steps of the outer loop method, each of which solves a regularized subproblem of the form, for $(g^{\mathsf{x}}, g^{\mathsf{y}}) \in \mathcal{X}^* \times \mathcal{Y}^*$ (and recalling our regularizer (4))

$$\min_{x \in [-1,1]^n, y \in \Delta^d} F(x,y) := \langle g^{\mathsf{x}}, x \rangle + \langle g^{\mathsf{y}}, y \rangle + \langle |\mathbf{A}|y, x^2 \rangle + \alpha h(y).$$

These subproblem forms are standard for mirror descent-based methods (including extragradient methods). In contrast to schemes using separable regularizers (admitting closed-form solutions), the subproblems induced by (4) themselves require an iterative method to solve. In [She17], a linearly-convergent minimization subroutine for $F$ was given based on multiplicative stability of simplex variables, using a fairly nonstandard analysis (see Lemma 6, [JST19]). Our next observation is that a linearly-convergent subproblem solver follows off-the-shelf from [LFN18] and viewing area convexity under the lens of *relative smoothness*, a more standard condition in optimization theory. We use the following convex analysis facts, deferring proofs to Section 3.2 of the supplement.

**Lemma 2.** *Let $\mathcal{X} \subseteq \mathbb{R}^m$ and $\mathcal{Y} \subseteq \mathbb{R}^n$ be convex compact subsets. Suppose $F : \mathcal{X} \times \mathcal{Y} \to \mathbb{R}$ is jointly convex over its argument $(x,y) \in \mathcal{X} \times \mathcal{Y}$. For $y \in \mathcal{Y}$, define $x_{\mathrm{br}}(y) := \operatorname{argmin}_{x \in \mathcal{X}} F(x,y)$ and $f(y) := F(x_{\mathrm{br}}(y), y)$. Then for all $y \in \mathcal{Y}$, $\partial_y F(x_{\mathrm{br}}(y), y) \subset \partial f(y)$.*

**Lemma 3.** *In the setting of Lemma 2, suppose for any $x \in \mathcal{X}$, $F(x, \cdot)$ (as a function over $\mathcal{Y}$) always is $r : \mathcal{Y} \to \mathbb{R}$ plus a linear function (where the linear function may depend on $x$). Then $r - f$ is convex, and $f - q$ is convex for any $q : \mathcal{Y} \to \mathbb{R}$ such that $F - q : \mathcal{X} \times \mathcal{Y} \to \mathbb{R}$ is jointly convex.*

Recall from Lemma 1 that $F$ is jointly convex over $(x,y)$ for any $\alpha \geq \frac{1}{2}$. Hence, for $\alpha = 2$, we may apply Lemma 3 with $r(y) := 2h(y)$ to conclude that $f(y) := \min_{x \in [-1,1]^n} F(x,y)$ is 2-relatively smooth with respect to $h$, as a function over $\mathcal{Y} = \Delta^d$. Moreover, applying Lemma 3 with $q(y) := h(y)$, and again using the joint convexity fact in Lemma 1, shows that $f$ is further 1-relatively strongly convex with respect to $h$. At this point, a direct application of Theorem 3.1 in [LFN18] (which gives an algorithm for optimization under relative smoothness and strong convexity) yields a linearly-convergent algorithm for minimizing $F$. We remark that in light of Lemma 2, we can implement gradient queries to $f$ by computing the best response argument for a given $y$.

Interestingly, this argument used nothing more than Lemma 3, joint convexity of $r_{\mathbf{A}}^{(\alpha)}$, and the ability to tune $\alpha$ to induce relative strong convexity (in contrast to [She17], which requires multiplicative stability properties). An important consequence is that the same technique generalizes to the matrix setting via new joint convexity facts we prove in Section 4 of the supplement, where multiplicative stability breaks due to non-monotonicity of matrix exponentials; the corresponding subproblem analysis in [She17] hence does not apply. This gives a simple proof-of-concept matching Theorem 2 up to logarithmic factors, which we improve via our approximation-tolerant extragradient methods.

# 3 Box-simplex games without alternating minimization

In this section, we show how to combine the insights regarding area convexity from Section 2, along with a careful implementation of our inner loop subproblem solvers, to prove Theorem 1.

## 3.1 Approximation-tolerant extragradient method

We begin by stating two oracles whose guarantees, when combined with Definition 3, give our conceptual algorithm for (1). These oracles can be viewed as approximately implementing steps of the extragradient method framework in Appendix B of the supplement. We develop subroutines which satisfy these relaxed definitions in the following Section 3.2. In the following definitions (and throughout), we let $V^{(\alpha)} := V^{r_{\mathbf{A}}^{(\alpha)}}$ be the divergence associated with $r_{\mathbf{A}}^{(\alpha)}$ for notational convenience.

**Definition 4** (Gradient step oracle). *For a problem* (1)*, we say $\mathcal{O}_{\mathrm{grad}} : \mathcal{Z} \times \mathcal{Z}^* \to \mathcal{Z}$ is an $(\alpha, \beta)$-gradient step oracle if on input $(z, v)$, it returns $z'$ such that*

$$\langle v, z' - u \rangle \leq V_z^{(\alpha+\beta)}(u) - V_{z'}^{(\alpha)}(u) - V_z^{(\alpha)}(z') \text{ for all } u \in \mathcal{Z}.$$

**Definition 5** (Extragradient step oracle). *For a problem* (1)*, we say $\mathcal{O}_{\text{xgrad}} : \mathcal{Z} \times \mathcal{Z}^* \times \mathcal{Y} \to \mathcal{Z}$ is an $(\alpha, \beta)$-extragradient step oracle if on input $(z, v, \bar{y})$, it returns $(z^+, \bar{y}^+)$ such that*

$$\langle v, z^+ - u \rangle \leq V_z^{(\alpha)}(u) - V_{z^+}^{(\alpha)}(u) - V_z^{(\alpha)}(z^+) + V_{\bar{y}}^{\beta h}(u^{\mathsf{y}}) - V_{\bar{y}^+}^{\beta h}(u^{\mathsf{y}}) \text{ for all } u = (u^{\mathsf{x}}, u^{\mathsf{y}}) \in \mathcal{Z}.$$

When $\beta = 0$, Definitions 4 and 5 reduce to the conventional proximal oracle steps used by the extragradient method of [Nem04]. In our solver for (1), we use $\beta > 0$ to compensate for our inexact subproblem solves. The asymmetry in Definitions 4 and 5 reflect an asymmetry in the analyses of extragradient methods. In typical analyses, the regret is bounded for the "gradient oracle" points, but the regret upper bound is stated in terms of the divergences of the "extragradient oracle" points (which our inexact oracles need to compensate for).

The utility of our Definitions 3, 4, and 5 reveals itself through the following lemma.

**Lemma 4.** *Let $z \in \mathcal{Z}$, $\bar{y} \in \mathcal{Y}$, $\alpha \geq 2$, $\beta, \gamma \geq 0$ and $0 \leq \eta \leq \frac{1}{3}$. Let $z' \leftarrow \mathcal{O}_{\text{grad}}(z, \eta g(z))$ and $(z^+, \bar{y}^+) \leftarrow \mathcal{O}_{\text{xgrad}}(z, \frac{\eta}{2} g(z'), \bar{y})$, where $\mathcal{O}_{\text{grad}}$ is an $(\alpha, \beta)$-gradient step oracle and $\mathcal{O}_{\text{xgrad}}$ is an $(\alpha + \beta, \gamma)$-extragradient step oracle. Then for all $u \in \mathcal{Z}$,*

$$\langle \eta g(z'), z' - u \rangle \leq 2V_z^{(\alpha+\beta)}(u) - 2V_{z^+}^{(\alpha+\beta)}(u) + 2V_{\bar{y}}^{\gamma h}(u^{\mathsf{y}}) - 2V_{\bar{y}^+}^{\gamma h}(u^{\mathsf{y}}).$$

*Proof.* By definition of $\mathcal{O}_{\text{grad}}$ (with $u \leftarrow z^+$) and $\mathcal{O}_{\text{xgrad}}$, we have

$$\begin{aligned} \langle \eta g(z), z' - z^+ \rangle &\leq V_z^{(\alpha+\beta)}(z^+) - V_{z'}^{(\alpha)}(z^+) - V_z^{(\alpha)}(z'), \\ \langle \eta g(z'), z^+ - u \rangle &\leq 2V_z^{(\alpha+\beta)}(u) - 2V_{z^+}^{(\alpha+\beta)}(u) - 2V_z^{(\alpha+\beta)}(z^+) + 2V_{\bar{y}}^{\gamma h}(u^{\mathsf{y}}) - 2V_{\bar{y}^+}^{\gamma h}(u^{\mathsf{y}}). \end{aligned} \tag{7}$$

Combining yields

$$\begin{aligned} \langle \eta g(z'), z' - u \rangle &\leq 2V_z^{(\alpha+\beta)}(u) - 2V_{z^+}^{(\alpha+\beta)}(u) + 2V_{\bar{y}}^{\gamma h}(u^{\mathsf{y}}) - 2V_{\bar{y}^+}^{\gamma h}(u^{\mathsf{y}}) \\ &\quad + \eta \left\langle g(z') - g(z), z' - z^+ \right\rangle - V_z^{(\alpha)}(z') - V_z^{(\alpha+\beta)}(z^+) - V_{z'}^{(\alpha)}(z^+) \\ &\leq 2V_z^{(\alpha+\beta)}(u) - 2V_{z^+}^{(\alpha+\beta)}(u) + 2V_{\bar{y}}^{\gamma h}(u^{\mathsf{y}}) - 2V_{\bar{y}^+}^{\gamma h}(u^{\mathsf{y}}) \\ &\quad + \eta \left\langle g(z') - g(z), z' - z^+ \right\rangle - V_z^{(\alpha)}(z') - V_z^{(\alpha)}(z^+), \end{aligned}$$

where in the second inequality we used that $V^{(\alpha+\beta)}$ dominates $V^{(\alpha)}$, and $V_{z'}^{(\alpha)}(z^+) \geq 0$ by Lemma 1. The conclusion follows by applying Definition 3 (see (6)) and the second fact in Lemma 1. $\qquad\square$

When $\beta = \gamma = 0$, Lemma 4 is the same as Appendix B of the supplement, and hence yields similar implications as standard extragradient methods: a scaling of the left-hand side upper bounds duality gap of $z'$, and the right-hand side telescopes (and is bounded using the following standard fact).

**Lemma 5.** *Let $\alpha, \gamma \geq 0$, and let $z_0 = (x_0, y_0)$ where $x_0 = \mathbb{0}_n$ and $y_0 = \frac{1}{d}\mathbb{1}_d$. Then $z_0$ is the minimizer of $r_{\mathbf{A}}^{(\alpha)}$ over $\mathcal{Z}$, and $V_{z_0}^{(\alpha)}(u) \leq 1 + \alpha \log d$, $V_{y_0}^{\gamma h}(u^{\mathsf{y}}) \leq \gamma \log d$ for all $u = (u^{\mathsf{x}}, u^{\mathsf{y}}) \in \mathcal{Z}$.*

Finally, for convenience to the reader, we put together Lemma 4 and 5 to obtain an analysis of the following conceptual "outer loop" extragradient algorithm (subject to the implementation of gradient and extragradient step oracles), Algorithm 1. Our end-to-end algorithm will be an explicit implementation of the framework in Algorithm 1; we provide a runtime analysis and error guarantee for our complete algorithm in Theorem 1, as well as pseudocode in Algorithm 4 of the supplement.

---

**Algorithm 1:** ConceptualBoxSimplex($\mathbf{A}, b, c, \mathcal{O}_{\text{grad}}, \mathcal{O}_{\text{xgrad}}$)

---

1 **Input:** $\mathbf{A} \in \mathbb{R}^{n \times d}$ with $L := \|\mathbf{A}\|_{1 \to 1}$, desired accuracy $\epsilon \in (0, L)$, $\mathcal{O}_{\text{grad}}$ a $(2, 2)$-gradient step oracle, $\mathcal{O}_{\text{xgrad}}$ a $(4, 4)$-extragradient step oracle

2 Initialize $x_0 \leftarrow \mathbb{0}_n$, $y_0 \leftarrow \frac{1}{d}\mathbb{1}_d$, $\bar{y}_0 \leftarrow \frac{1}{d}\mathbb{1}_d$, $\hat{x} \leftarrow \mathbb{0}_n$, $\hat{y} \leftarrow \mathbb{0}_d$, $T \leftarrow \lceil \frac{6(8 \log d + 1)L}{\epsilon} \rceil$, $\eta \leftarrow \frac{1}{3}$

3 Rescale $\mathbf{A} \leftarrow \frac{1}{L}\mathbf{A}$, $b \leftarrow \frac{1}{L}b$, $c \leftarrow \frac{1}{L}c$

4 **for** $t = 0$ **to** $T - 1$ **do**

5     $g_t \leftarrow (\mathbf{A}y_t + c, b - \mathbf{A}^\top x_t)$

6     $z_t' := (x_t', y_t') \leftarrow \mathcal{O}_{\text{grad}}(z_t, \eta g_t)$

7     $g_t' \leftarrow (\mathbf{A}y_t' + c, b - \mathbf{A}^\top x_t')$

8     $(z_{t+1}, \bar{y}_{t+1}) := (x_{t+1}, y_{t+1}, \bar{y}_{t+1}) \leftarrow \mathcal{O}_{\text{xgrad}}(z_t, \frac{\eta}{2}g_t', \bar{y}_t)$

9 **end**

10 **Return:** $(\hat{x}, \hat{y}) \leftarrow \frac{1}{T}\sum_{t=0}^{T-1}(x_t', y_t')$

---

**Corollary 1.** *Algorithm 1 deterministically computes an $\epsilon$-approximate saddle point to* (1).

*Proof.* First, clearly the rescaling in Line 3 multiplies the entire problem (1) by $\frac{1}{L}$, so an $\frac{\epsilon}{L}$-approximate saddle point to the new problem becomes an $\epsilon$-approximate saddle point for the original. Throughout the rest of the proof it suffices to treat $L = 1$. Next, by telescoping and averaging Lemma 4 with $\alpha = \beta = 2$, $\gamma = 4$, and $\eta = \frac{1}{3}$, we have for $z_0 = (x_0, y_0)$ and any $u = (u^\times, u^y) \in \mathcal{Z}$

$$\frac{1}{T}\sum_{t=0}^{T-1} \langle g(z_t'), z_t' - u \rangle \leq \frac{6\left(V_{z_0}^{(4)}(u) + V_{\bar{y}_0}^{4Lh}(u^y)\right)}{T} \leq \epsilon.$$

The last inequality used the bounds in Lemma 5 and the definition of $T$. Moreover since $g$ is bilinear, and $\hat{z} := (\hat{x}, \hat{y})$ is the average of the $z_t'$ iterates, we have $\langle g(\hat{z}), \hat{z} - u \rangle \leq \epsilon$. Taking the supremum over $u \in \mathcal{Z}$ bounds the duality gap of $\hat{z}$ and gives the conclusion. $\qquad\square$

### 3.2 Implementing oracles

In this section, we give generic constructions of gradient and extragradient step oracles.[10]

---

**Algorithm 2:** GradStepOracle($z, v, \alpha, \beta$)

---

1 **Input:** $z = (x, y) \in \mathcal{Z}$, $v = (v^\times, v^y) \in \mathcal{Z}^*$, $\alpha, \beta \geq 0$

2 $y' \leftarrow \operatorname{argmin}_{\hat{y} \in \mathcal{Y}} \left\langle v^y + \nabla_y r_{\mathbf{A}}^{(\alpha)}(x_{\text{br}}(y), y) - \nabla_y r_{\mathbf{A}}^{(\alpha)}(z), \hat{y} \right\rangle + V_y^{\beta h}(\hat{y})$, where for all $\hat{y} \in \mathcal{Y}$,

$$x_{\text{br}}(\hat{y}) := \operatorname{argmin}_{\hat{x} \in \mathcal{X}} \left\langle v^\times - \nabla_x r_{\mathbf{A}}^{(\alpha)}(z), \hat{x} \right\rangle + r_{\mathbf{A}}^{(\alpha)}(\hat{z}) \text{ where } \hat{z} := (\hat{x}, \hat{y}) \qquad (8)$$

3 $x' \leftarrow x_{\text{br}}(y')$

4 **Return:** $(x', y')$

---

**Lemma 6.** *For $\beta \geq \alpha \geq \frac{1}{2}$, Algorithm 2 is an $(\alpha, \beta)$-gradient step oracle.*

*Proof.* By (3) and the first-order optimality condition for $y'$, we have for any $u = (u^\times, u^y) \in \mathcal{Z}$,

$$\left\langle v^y + \nabla_y r_{\mathbf{A}}^{(\alpha)}(x_{\text{br}}(y), y) - \nabla_y r_{\mathbf{A}}^{(\alpha)}(z), y' - u^y \right\rangle \leq V_y^{\beta h}(u^y) - V_{y'}^{\beta h}(u^y) - V_y^{\beta h}(y'). \quad (9)$$

Further, define for any $\hat{y} \in \mathcal{Y}$,

$$f(\hat{y}) := \langle v, \hat{z} \rangle + V_z^{(\alpha)}(\hat{z}) \text{ where } \hat{z} := (x_{\text{br}}(\hat{y}), \hat{y}). \quad (10)$$

Note that $f$ is a partial minimization of a function on two variables which is a linear term plus $r_{\mathbf{A}}^{(\alpha)}$, which is convex by Lemma 1. For any fixed $x$, $r_{\mathbf{A}}^{(\alpha)}(x, y)$ is itself $\alpha h(y)$ plus a linear function.

---

[10]We slightly abuse notation and use $x_{\text{br}}$ in (8) in a consistent way with how it is used in Lemma 2, where $F$ in Lemma 2 is taken to be the jointly convex function of $(x, y)$ in (8) before minimizing over $\mathcal{X}$.

Lemma 3 then shows $f$ is $\alpha$-relatively smooth with respect to $h$. We then have for all $\hat{y}, u^{\mathsf{y}} \in \mathcal{Y}$,

$$
\begin{aligned}
\left\langle \nabla_y r_{\mathbf{A}}^{(\alpha)}(x_{\mathrm{br}}(\hat{y}), \hat{y}) - \nabla_y r_{\mathbf{A}}^{(\alpha)}(x_{\mathrm{br}}(y), y), \hat{y} - u^{\mathsf{y}} \right\rangle &= \langle \partial f(\hat{y}) - \partial f(y), \hat{y} - u^{\mathsf{y}} \rangle \\
&\leq V_y^{\alpha h}(\hat{y}) + V_{\hat{y}}^{\alpha h}(u^{\mathsf{y}}).
\end{aligned}
\tag{11}
$$

Here the equality used Lemma 2 where the linear shift between $r_{\mathbf{A}}^{(\alpha)}$ and the minimization problem inducing $f$ cancels in the expression $\partial f(\hat{y}) - \partial f(y)$, and the inequality used Fact 1. Combining (9) and (11) (with $\hat{y} \leftarrow y'$) and using $V^{\beta h}$ dominates $V^{\alpha h}$ yields

$$
\left\langle v^{\mathsf{y}} + \nabla_y r_{\mathbf{A}}^{(\alpha)}(z') - \nabla_y r_{\mathbf{A}}^{(\alpha)}(z), y' - u^{\mathsf{y}} \right\rangle \leq V_y^{\beta h}(u^{\mathsf{y}}).
\tag{12}
$$

Finally, first-order optimality of $x'$ with respect to the objective induced by $y'$ implies for all $u^{\mathsf{x}} \in \mathcal{X}$,

$$
\left\langle v^{\mathsf{x}} + \nabla_x r_{\mathbf{A}}^{(\alpha)}(z') - \nabla_x r_{\mathbf{A}}^{(\alpha)}(z), x' - u^{\mathsf{x}} \right\rangle \leq 0,
\tag{13}
$$

so combining with (12) we have for $u = (u^{\mathsf{x}}, u^{\mathsf{y}})$, $\langle v + \nabla r_{\mathbf{A}}^{(\alpha)}(z') - \nabla r_{\mathbf{A}}^{(\alpha)}(z), z' - u \rangle \leq V_y^{\beta h}(u^{\mathsf{y}})$. The conclusion follows by using the identity (3) to rewrite $\langle \nabla r_{\mathbf{A}}^{(\alpha)}(z') - \nabla r_{\mathbf{A}}^{(\alpha)}(z), u - z' \rangle$. □

---

**Algorithm 3:** XGradStepOracle$(z, v, \bar{y}, \alpha, \beta)$

---

1 **Input:** $z = (x, y) \in \mathcal{Z}$, $v = (v^{\mathsf{x}}, v^{\mathsf{y}}) \in \mathcal{Z}^*$, $\bar{y} \in \mathcal{Y}$, $\alpha, \beta \geq 0$

2 $y^+ \leftarrow \operatorname{argmin}_{\hat{y} \in \mathcal{Y}} \left\langle v^{\mathsf{y}} + \nabla_y r_{\mathbf{A}}^{(\alpha)}(x_{\mathrm{br}}(\bar{y}), \bar{y}) - \nabla_y r_{\mathbf{A}}^{(\alpha)}(z), \hat{y} \right\rangle + V_{\bar{y}}^{\beta h}(\hat{y})$ (following (8))

3 $x^+ \leftarrow x_{\mathrm{br}}(y^+)$

4 $\bar{y}^+ \leftarrow \operatorname{argmin}_{\hat{y} \in \mathcal{Y}} \left\langle v^{\mathsf{y}} + \nabla_y r_{\mathbf{A}}^{(\alpha)}(x^+, y^+) - \nabla_y r_{\mathbf{A}}^{(\alpha)}(z), \hat{y} \right\rangle + V_{\bar{y}}^{\beta h}(\hat{y})$

5 **Return:** $(x^+, y^+, \bar{y}^+)$

---

We defer a proof of the following to Section 3.2 of the supplement, as it is similar to Lemma 6. We remark that the auxiliary sequence $\bar{y}, \bar{y}^+$ used in the extragradient step oracle definition is itself inspired by running a few steps of a descent method on the subproblem, inducing a telescoping divergence sequence used by the outer loop to pay for inaccuracy in the subproblem solution.

**Lemma 7.** *For $\beta \geq \alpha \geq \frac{1}{2}$, Algorithm 3 is an $(\alpha, \beta)$-extragradient step oracle.*

### 3.3 Proof of Theorem 1

**Theorem 1.** *There is an algorithm (stated in full as Algorithm 4 in the supplement) deterministically computing an $\epsilon$-approximate saddle point to* (1) *in time*

$$
O\left( \mathrm{nnz}(\mathbf{A}) \cdot \frac{\|\mathbf{A}\|_{1 \to 1} \log d}{\epsilon} \right).
$$

*Proof.* We briefly describe the algorithm from the supplement for completeness here, as correctness follows straightforwardly from combining Corollary 1 with Lemmas 6 and 7. The algorithm is an instance of the conceptual Algorithm 1. Each step of Algorithm 1 then first implements Algorithm 2 (used in Lemma 6), with the parameters required by Lemma 4, to obtain a gradient step oracle. Similarly, it next implements Algorithm 3 (used in Lemma 7), with the parameters required by Lemma 4, to obtain an extragradient step oracle. Correctness follows from Corollary 1. For the runtime, without loss of generality $\mathrm{nnz}(\mathbf{A}) \geq \max(n, d)$ (else we may drop columns or rows appropriately), and each of $T$ iterations is dominated by a constant number of matrix-vector multiplications. □

## Acknowledgements

We would like thank our long-term collaborators Yujia Jin and Aaron Sidford for many helpful discussions at earlier stages of this project, which improved our understanding of area convexity. We also thank Victor Reis for his collaboration on related ideas to this work in [JRT23]. Finally, we thank anonymous reviewers for their helpful suggestions in improving our presentation.

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
