]. For convenience to the reader, we give self-contained expositions of these relationships in Appendices A and B. As byproducts, our improved understanding of area convexity results in new state-of-the-art runtimes for (1) by affirmatively answering Question 1, and the first accelerated solver for a matrix generalization by affirmatively answering Question 2.

## 1.1 Our results

We begin by stating our new runtime results in this section, deferring a more extended discussion of how we achieve them via improved analyses of the [She17] framework to the following Section 1.2.

**Box-simplex games.** We give an algorithm for solving (1) improving the runtime of [She17] by a logarithmic factor, by removing the need for high-accuracy inner loop solutions.

**Theorem 1.** *Algorithm 4 deterministically computes an $\epsilon$-approximate saddle point to* (1) *in time[5]*

$$O\left(\text{nnz}(\mathbf{A}) \cdot \frac{\|\mathbf{A}\|_{1\to 1}\log d}{\epsilon}\right).$$

We define nnz and $\|\mathbf{A}\|_{1\to 1}$ in Section 2.1. The most direct comparison to Theorem 1 is the work [She17], which we improve. All other first-order methods for solving (1) we are aware of are slower than [She17] by at least an $\approx \min(\epsilon^{-1}, \sqrt{n})$ factor. Higher-order methods for (1) (e.g. interior point methods) obtain improved dependences on $\epsilon$ at the cost of polynomial overhead in the dimension.

As a byproduct of Theorem 1, we improve all the recent applications of (1), including approximate maximum flow [She17], optimal transport [JST19], min-mean-cycle [AP20], and semi-streaming variants of bipartite matching and transshipment [AJJ+22]. These are summarized in Section 5. The design of efficient approximate solvers for optimal transport in particular is a problem which has received a substantial amount of attention from the learning theory community (see Table 1 and the survey [PC19]). The runtime of Theorem 1 represents a natural conclusion to this line of work.[6] We believe our improvement in Theorem 1 (avoiding a high-accuracy subproblem solve) takes an important step towards bridging the state-of-the-art in the theory and practice of optimal transport, a well-motivated undertaking due to its widespread use in applications.

**Box-spectraplex games.** We initiate the algorithmic study of box-spectraplex games of the form

$$\min_{x\in[-1,1]^n} \max_{\mathbf{Y}\in\Delta^{d\times d}} \left\langle \mathbf{Y}, \sum_{i\in[n]} x_i \mathbf{A}_i \right\rangle - \langle \mathbf{B}, \mathbf{Y}\rangle + c^\top x, \tag{2}$$

where $\{\mathbf{A}_i\}_{i\in[n]}$, $\mathbf{B}$ are $d\times d$ symmetric matrices and $\mathbf{Y}\succeq\mathbf{0}$ satisfies $\text{Tr}(\mathbf{Y})=1$ (i.e. $\mathbf{Y}\in\Delta^{d\times d}$, cf. Section 2). This problem family is a natural formulation of semidefinite programming (SDP), and generalizes (1), the special case of (2) where all of the matrices $\{\mathbf{A}_i\}_{i\in[n]}$, $\mathbf{B}$, $\mathbf{Y}$ are diagonal.

---

[5]This statement assumes direct access to the entries of $\mathbf{A}$. More generally, letting $\mathcal{T}_{\text{mv}}$ be the time it takes to multiply a vector by the argument, we may replace nnz($\mathbf{A}$) by $\max(\mathcal{T}_{\text{mv}}(\mathbf{A}), \mathcal{T}_{\text{mv}}(\mathbf{A}^\top), \mathcal{T}_{\text{mv}}(|\mathbf{A}|), \mathcal{T}_{\text{

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

 incurred by Theorem 2 (due to the higher ranks in the randomized sketches we use) can be avoided, and discuss the latter in Section 1.2.

## 1.2 Our approach

We begin our overview of our techniques with a short overview of area convexity from the perspective of both the extragradient method "outer loop" and the alternating minimization "inner loop" used by the [She17] algorithm. We then summarize the new perspectives on both loops developed in this paper, and describe how these perspectives allow us to obtain our main results, Theorems 1 and 2.

**Area convexity.** Extragradient methods [Nem04, Nes07] are a powerful general-purpose tool for solving convex-concave minimax optimization problems with a $L$-Lipschitz gradient operator $g$. To obtain an $\epsilon$-approximate saddle point, extragradient methods take descent-ascent steps regularized by a function $r$, which is 1-strongly convex in the same norm $g$ is Lipschitz in; their convergence rates then scale as $L\Theta\cdot\epsilon^{-1}$, where $\Theta$ is the additive range of $r$. For $\ell_1$-$\ell_1$ or $\ell_2$-$\ell_1$ games, regularizers with $\Theta = \widetilde{O}(1)$ exist, and hence the results of [Nem04, Nes07] immediately imply accelerated $\widetilde{O}(L\epsilon^{-1})$ rates for approximating equilibria. For $\ell_\infty$-$\ell_1$ games however, a fundamental difficulty is that under the strong convexity requirement over $\mathcal{X} := [-1,1]^n$, $\Theta$ is necessarily $\Omega(n)$ [ST18].

To bypass this issue, Sherman introduced a regularizer capturing the "local geometry" of (1).[8] A (slight variant) of the [She17] regularizer has the form, for some parameter $\alpha > 0$,

$$r(x, y) := \left\langle |\mathbf{A}| y, x^2 \right\rangle + \alpha h(y), \text{ where } h(y) := \sum_{j \in [d]} y_j \log y_j \text{ is negative entropy.} \tag{3}$$

Here, $|\mathbf{A}|$ and $x^2$ are applied entrywise. Sherman shows (3) satisfies a weaker condition known as *area convexity* (Definition 3), and modifies standard extragradient methods to converge under this condition. However, the area convexity definition is fairly unusual from the perspective of standard analyses, which typically bound regret notions via Bregman divergences in the regularizer used.

Intuitively, (3) obtains a smaller range by placing weights $|\mathbf{A}| y$ on the quadratic $x^2$. Moreover, the reweighted quadratic $\left\langle |\mathbf{A}| y, x^2 \right\rangle$ still has enough strong convexity in multiplicative neighborhoods of a dual variable $y$ for extragradient methods to converge. Sherman's proof that (3) satisfies area convexity also implies $r$ is jointly convex for sufficiently large $\alpha$, which means that by increasing $\alpha$, we can ensure $\nabla^2 r$ is dominated by a multiple of the entropy component $\nabla^2 h$. Sherman used this Hessian domination condition to efficiently minimize subproblems of the form $f_v(z) := \langle v, z \rangle + r(z)$, for $z \in [-1, 1]^n \times \Delta^d$, required by the outer loop. These subproblems $f_v$, which are non-separable, do not admit closed form solutions, but [She17] showed they are well-conditioned enough (due to entropy domination) for alternating minimization to converge rapidly.

**Improved subproblem solvers.** Our first observation is that the Hessian domination condition used by [She17] to ensure rapid convergence of subproblems can be viewed through the lens of *relative smoothness* [BBT17, LFN18], a more standard condition in optimization theory. Simple facts from convex analysis (Lemma 5) imply that after minimizing over the box, the induced function $\tilde{f}_v(y) := \min_{x \in [-1,1]^n} f_v(x, y)$ is relatively smooth in $h$. By tuning $\alpha$ in (3), we can further ensure $\tilde{f}_v$ is relatively strongly convex in $h$, and use methods for relatively well-conditioned optimization from [LFN18] off-the-shelf to minimize $\tilde{f}_v$ (and hence also $f_v$). We give a self-contained presentation of this strategy in Appendix A, yielding a simplified solver for the subproblems in [She17].

While this is a seemingly a small difference in proof strategy, the convergence analysis in [She17] strongly used that $\nabla^2 f_v = \nabla^2 r$ is multiplicatively stable in small neighborhoods of a dual variable $y$. Our new subproblem solvers, which are based on relative smoothness, avoid this requirement and hence apply to the setting of Theorem 2, where matrix variables on $\Delta^{d \times d}$ do not satisfy the required multiplicative stability. Specifically, $\exp(\log \mathbf{Y} + \mathbf{G})$ for operator-norm bounded $\mathbf{G}$ is not necessarily multiplicatively well-approximated by $\mathbf{Y}$, which is required by a natural generalization of the [She17] subproblem analysis (the corresponding statement for vectors is true). On the other hand, our relative smoothness analysis avoids the need for dual variable local stability altogether.

**Improved extragradient analyses.** Our next observation is that Sherman's area convexity condition (Definition 3) actually implies a natural Bregman divergence domination condition (8), which is closely related to more standard analyses of extragradient methods. We discuss this relationship in more detail in Appendix B, where we unify [She17] with [Nem04, Nes07, CST21] and propose a weaker condition than used in either existing analysis which suffices for convergence.

The simple realization that area convexity implies a bound based on Bregman divergences, which to our knowledge is novel, allows us to reinterpret the [She17] outer loop to tolerate larger amounts of inexactness in the solutions for the subproblems $\min_{z \in [-1,1]^n} f_v(z)$ defined earlier, where

---

[8]Sherman actually studied a more general problem setting than (1), where the simplex is replaced by an $\ell_1$-constrained product set. For most applications of area convexity, it suffices to use simplex dual variables, and we use a different, simpler regularizer introduced by a follow-up work specifically for box-simplex games [JST19].

inexactness is measured in Bregman divergence. We show that by combining a tolerant variant of the [She17] outer loop with our relative smoothness analysis, a constant number of alternating steps solves subproblems in $f_v$ to sufficient accuracy. More specifically, we maintain an auxiliary sequence of dual variables to warm-start our subproblems (see $\bar{y}$ in Definition 5), and show that a telescoping entropy divergence bound on this auxiliary sequence pays for the inexactness of subproblem solves. Combining these pieces yields our faster box-simplex game solver, Theorem 1. Interestingly, its proof builds upon our insights regarding area convexity in both Appendices A and B.

**Spectrahedral generalizations.** To generalize these methods to solve box-spectraplex games, we introduce a natural matrix variant of the regularizer (3) in (7). Proving it satisfies area convexity (or even joint convexity) introduces new challenges, as the corresponding proofs for (3) exploit coordinatewise-decomposability [She17, JST19], whereas matrix regularizers induce incompatible eigenspaces. We use tools from representation theory to avoid this issue and prove convexity of (7).

Finally, a major computational obstacle in minimax problems over $\mathbf{Y} := \Delta^{d \times d}$ is that the standard regularizer over this set, von Neumann entropy $H(\mathbf{Y}) := \langle \mathbf{Y}, \log \mathbf{Y} \rangle$, induces iterates as matrix exponentials; explicitly computing even a single iterate typically takes superlinear time. In the simpler setting of simplex-spectraplex games, these computations can be implicitly performed using polynomial approximations to the exponential and randomized sketches [AK07]. The tolerance required by analogous computations in the box-spectraplex setting is more fine-grained, especially when computing the best responses $\operatorname{argmin}_{x \in [-1,1]^n} f_v(x, \mathbf{Y})$ required by our subproblem solvers, which can be quite unstable. By carefully using multiplicative-additive approximations to the matrix exponential (Definition 6), we show how to efficiently implement all the operations needed using Johnson-Lindenstrauss sketches of rank $\widetilde{O}(L^2 \epsilon^{-2})$. An interesting open problem suggested by our work is whether our methods tolerate lower-rank sketches of rank $\widetilde{O}(L\epsilon^{-1})$, which would improve Theorem 2 by this factor. These lower-rank sketches suffice for simplex-spectraplex games [BBN13], though we suspect corresponding analyses for box-spectraplex games will also need to use more fine-grained notions of error tolerance, which we defer to future work.

## 1.3 Related work

The family of box-simplex games (1) is captured by linear programming [LS15], where state-of-the-art solvers [CLS21, vdBLL+21] run in time $\widetilde{O}((n+d)^\omega)$ or $\widetilde{O}(nd+\min(n,d)^{2.5})$, where $\omega \approx 2.37$ is the current matrix multiplication constant [AW21, DWZ22]. These LP solvers run in superlinear time, and practical implementations do not currently exist; on the other hand, the convergence rates depend polylogarithmically on the accuracy parameter $L\epsilon^{-1}$. The state-of-the-art approximate solver for (1) (with runtime depending linearly on the input size $\operatorname{nnz}(\mathbf{A})$ and polynomially on $L\epsilon^{-1}$) is the accelerated algorithm of [She17], which is improved by our Theorem 1 by a logarithmic factor. Finally, we mention [BSW19] as another recent work which relies on area convexity techniques to design an algorithm in a different problem setting; their inner loop also requires high-precision solves (as in [She17]), and this similarly results in a logarithmic overhead in the runtime.

For the specific application of optimal transport, an optimization problem captured by (1) receiving significant recent attention from the learning theory community, several alternative specialized solvers have been developed beyond those in Table 1. By exploiting relationships between Sinkhorn regularization and faster solvers for matrix scaling [CMTV17, ALdOW17], [BJKS18] gave an alternative solver obtaining a $\widetilde{O}(n^2 \cdot \epsilon^{-1})$ rate. These matrix scaling algorithms call specialized graph Laplacian system solvers as a subroutine, which are currently less practical than our methods based on matrix-vector queries. Finally, the recent breakthrough maximum flow algorithm of [CKL+22]

also extends to solve optimal transport in $O(n^{2+o(1)})$ time (though its practicality at this point remains unclear). For moderate $\epsilon \geq n^{-o(1)}$, this is slower than Theorem 1.

To our knowledge, there have been no solvers developed in the literature which tailor specifically to the family of box-spectraplex games (2). These problems are solved by general-purpose SDP solvers, where the state-of-the-art runtimes of $\widetilde{O}(n^\omega \sqrt{d} + nd^{2.5})$ or $\widetilde{O}(n^\omega + d^{4.5} + n^2\sqrt{d})$ [JKL+20, HJS+22] are highly superlinear (though again, they depend polylogarithmically on the inverse accuracy). We believe our techniques in Section 4 extend straightforwardly to show that a variant of gradient descent in the $\ell_\infty$ geometry [KLOS14] solves (2) in an unaccelerated $\widetilde{O}(L^2\epsilon^{-2})$ iterations, with the same per-iteration complexity as Theorem 2. As discussed in Section 1.2, an interesting open direction is to improve the per-iteration complexity of Theorem 2 using sketches of lower rank $\widetilde{O}(L\epsilon^{-1})$, paralleling developments for simplex-spectraplex games [BBN13, AL17, CDST19].

## 2 Preliminaries

### 2.1 Notation

**General notation.** We use $\widetilde{O}$ to suppress polylogarithmic factors in problem parameters for brevity. Throughout $[n] := \{i \in \mathbb{N} \mid 1 \leq i \leq n\}$. When applied to a vector $\|\cdot\|_p$ is the $\ell_p$ norm. We refer to the dual space (bounded linear operators) of a set $\mathcal{X}$ by $\mathcal{X}^*$. The all-zeroes and all-ones vectors in dimension $d$ are denoted $\mathbb{0}_d$ and $\mathbb{1}_d$. When $u, v$ are vectors of equal dimension, we let $u \circ v$ denote their coordinatewise multiplication. Matrices are denoted in boldface. The symmetric $d \times d$ matrices are $\mathbb{S}^d$, equipped with the Loewner partial ordering $\preceq$ and $\langle \mathbf{A}, \mathbf{B} \rangle = \mathrm{Tr}(\mathbf{AB})$. The $d \times d$ positive semidefinite cone is $\mathbb{S}^d_{\succeq \mathbf{0}}$, and the set of $d \times d$ positive definite matrices is $\mathbb{S}^d_{\succ \mathbf{0}}$. For $\mathbf{Y} \in \mathbb{S}^d$ with eigendecomposition $\mathbf{Y} = \mathbf{U}^\top \mathbf{\Lambda} \mathbf{U}$ for $\mathbf{\Lambda} = \mathbf{diag}\,(\lambda)$ we define $\exp(\mathbf{Y}) := \mathbf{U}^\top \mathbf{diag}\,(\exp(\lambda))\,\mathbf{U}$ (where $\exp(\lambda)$ is entrywise) and similarly define $|\mathbf{Y}|$ and $\log \mathbf{Y}$ (when $\mathbf{Y} \succ \mathbf{0}$). The operator norm of $\mathbf{Y} \in \mathbb{S}^d$ is denoted $\|\mathbf{Y}\|_{\mathrm{op}}$ and is the largest eigenvalue of $|\mathbf{Y}|$; the trace norm $\|\mathbf{Y}\|_{\mathrm{tr}}$ is $\mathrm{Tr}|\mathbf{Y}|$. The all-zeroes and identity matrices of dimension $d$ are $\mathbf{0}_d$ and $\mathbf{I}_d$. For $\mathbf{A} \in \mathbb{R}^{n \times d}$ we denote its $i^{\text{th}}$ row (for $i \in [n]$) by $\mathbf{A}_{i:}$ and $j^{\text{th}}$ column (for $j \in [d]$) by $\mathbf{A}_{:j}$. For $p, q \geq 1$ we define $\|\mathbf{A}\|_{p \to q} := \sup_{\|v\|_p = 1} \|\mathbf{A}v\|_q$. The number of nonzero entries in $\mathbf{A}$ is denoted $\mathrm{nnz}(\mathbf{A})$, and $\mathcal{T}_{\mathrm{mv}}(\mathbf{A})$ is the time it takes to multiply a vector by $\mathbf{A}$. We use $\mathrm{med}(x, -1, 1)$ to mean the projection of $x$ onto $[-1, 1]$, where med means median. We define the $d$-dimensional simplex and the $d \times d$ spectraplex

$$\Delta^d := \{y \in \mathbb{R}^d_{\geq 0} \mid \|y\|_1 = 1\} \text{ and } \Delta^{d \times d} := \{\mathbf{Y} \in \mathbb{S}^{d \times d}_{\succeq \mathbf{0}} \mid \mathrm{Tr}\mathbf{Y} = 1\}.$$

Throughout we reserve the function names $h(y) := \sum_{j \in [d]} y_j \log y_j$ (for $y \in \Delta^d$) and $H(\mathbf{Y}) := \langle \mathbf{Y}, \log \mathbf{Y} \rangle$ (for $\mathbf{Y} \in \Delta^{d \times d}$) for negated (vector) entropy and (matrix) von Neumann entropy.

**Optimization.** For convex function $f$, $\partial f(x)$ refers to the subgradient set at $x$; we sometimes use $\partial f$ to denote any (consistently chosen) member of the subgradient set. Following [LFN18], we say $f$ is $L$-relatively smooth with respect to convex function $r$ if $Lr - f$ is convex, and we say $f$ is $m$-strongly convex with respect to $r$ if $f - mr$ is convex. For a differentiable convex function $f$ we define the associated (nonnegative, convex) Bregman divergence $V_x^f(x') := f(x') - f(x) - \langle \nabla f(x), x' - x \rangle$. The Bregman divergence satisfies the well-known identity

$$\langle \nabla V_x^f(x'), u - x' \rangle = \langle \nabla f(x') - \nabla f(x), u - x' \rangle = V_x^f(u) - V_{x'}^f(u) - V_x^f(x'). \tag{4}$$

For convex function $f$ on two variables $(x, y)$, we use $\partial_y f(x, y)$ to denote the subgradient set at $y$ of the restricted function $f(x, \cdot)$. We call a function $f$ of two variables $(x, y) \in \mathcal{X} \times \mathcal{Y}$ convex-concave if its restrictions to the first and second block are respectively convex and concave. We call

$(x, y) \in \mathcal{X} \times \mathcal{Y}$ an $\epsilon$-approximate saddle point if its *duality gap*, $\max_{y' \in \mathcal{Y}} f(x, y') - \min_{x' \in \mathcal{X}} f(x', y)$, is at most $\epsilon$. For differentiable convex-concave function $f$ clear from context, its subgradient operator is $g(x, y) := (\partial_x f(x, y), -\partial_y f(x, y))$; when $f$ is differentiable, we will call this a gradient operator. We say operator $g$ is monotone if for all $w, z$ in its domain, $\langle g(w) - g(z), w - z \rangle \geq 0$: examples are the subgradient of a convex function or subgradient operator of a convex-concave function.

## 2.2 Box-simplex games

In Section 3, we develop a solver for *box-simplex* games of the form

$$\min_{x \in [-1,1]^n} \max_{y \in \Delta^d} x^\top \mathbf{A} y - b^\top y + c^\top x,$$

for some $\mathbf{A} \in \mathbb{R}^{n \times d}$, $b \in \mathbb{R}^d$, and $c \in \mathbb{R}^n$. We will design methods computing approximate saddle points to (1) which leverage the following family of regularizers (recalling the definition of $h$ from Section 2.1):

$$r_{\mathbf{A}}^{(\alpha)}(x, y) := \langle |\mathbf{A}| y, x^2 \rangle + \alpha h(y), \tag{5}$$

where the absolute value is applied to $\mathbf{A}$ entrywise, and squaring is applied to $x$ entrywise. In the context of Section 3 only, $|\mathbf{A}|$ will be applied entrywise (rather than to the spectrum). This family of regularizers was introduced by [JST19], a minor modification to a similar family given by [She17], and has multiple favorable properties for the geometry present in the problem (1). In this context, we will use $\mathcal{X} := [-1, 1]^n$, and for any $v \in \mathbb{R}^n$ we let $\mathbf{\Pi}_{\mathcal{X}}(v) := \mathrm{med}(v, -1, 1)$ be truncation applied entrywise. We also use $\mathcal{Y} := \Delta^d$ and for any $v \in \mathbb{R}_{\geq 0}^d$ we let $\mathbf{\Pi}_{\mathcal{Y}}(v) := \frac{v}{\|v\|_1}$ normalize onto $\mathcal{Y}$.

## 2.3 Box-spectraplex games

In Section 4, we develop a solver for *box-spectraplex* games, which are bilinear minimax problems defined with respect to a set of matrices $\mathcal{A} := \{\mathbf{A}_i\}_{i \in [n]} \subset \mathbb{S}^d$. We define $|\mathcal{A}| := \{|\mathbf{A}_i|\}_{i \in [n]}$ where the absolute value is applied spectrally. We also denote for $x \in \mathbb{R}^n$ and $\mathbf{Y} \in \mathbb{R}^{d \times d}$,

$$\mathcal{A}(x) := \sum_{i \in [n]} x_i \mathbf{A}_i, \ \mathcal{A}^*(\mathbf{Y}) := \{\langle \mathbf{A}_i, \mathbf{Y} \rangle\}_{i \in [n]}.$$

The box-spectraplex games we study are of the form

$$\min_{x \in [-1,1]^n} \max_{\mathbf{Y} \in \Delta^{d \times d}} \langle \mathbf{Y}, \mathcal{A}(x) \rangle - \langle \mathbf{B}, \mathbf{Y} \rangle + c^\top x,$$

for some $\mathcal{A} \subset \mathbb{S}^d$, $\mathbf{B} \in \mathbb{S}^d$, and $c \in \mathbb{R}^n$. The Lipschitz constant of problem (2) is denoted

$$L_{\mathcal{A}} := \left\| \sum_{i \in [n]} |\mathbf{A}_i| \right\|_{\mathrm{op}}. \tag{6}$$

Analogously to the box-simplex setting, we use the following two-parameter family of regularizers (recalling the definition of $H$ from Section 2.1):

$$r_{\mathcal{A}}^{(\alpha,\mu)}(x, \mathbf{Y}) := \langle |\mathcal{A}|^*(\mathbf{Y}), x^2 \rangle + \alpha H(\mathbf{Y}) + \frac{\mu}{2} \|x\|_2^2. \tag{7}$$

When $\mu = 0$, we denote (7) by $r_{\mathcal{A}}^{(\alpha)}$. We define $\mathcal{X}$ and $\mathbf{\Pi}_{\mathcal{X}}$ similarly to Section 2.2 in the context of Section 4, where we also overload $\mathcal{Y} := \Delta^{d \times d}$ and for any $\mathbf{V} \in \mathbb{S}_{\succeq \mathbf{0}}^d$ we let $\mathbf{\Pi}_{\mathcal{Y}}(\mathbf{V}) := \frac{\mathbf{V}}{\mathrm{Tr}\mathbf{V}}$.

## 2.4 Extragradient methods

In Appendix B, we analyze extragradient algorithms for approximately solving variational inequalities in an operator $g$, i.e. which find $z$ with small $\langle g(z), z - u \rangle$ for all $u$ in the domain. We analyze convergence under the following condition, which is weaker than previous notions in [She17, CST21].

**Definition 1** (Relaxed relative Lipschitzness). *We say an operator* $g : \mathcal{Z} \to \mathcal{Z}^*$ *is* $\frac{1}{\eta}$*-relaxed relatively Lipschitz with respect to* $r : \mathcal{Z} \to \mathbb{R}$ *if for all* $(z, z', z^+) \in \mathcal{Z} \times \mathcal{Z} \times \mathcal{Z}$,

$$\eta \left\langle g(z') - g(z), z' - z^+ \right\rangle \leq V_z^r(z') + V_{z'}^r(z^+) + V_z^r(z^+).$$

This notion is related to and subsumes the notions of relative Lipschitzness [CST21] and area convexity [She17] which have recently been proposed to analyze extragradient methods, which we define below. In particular, Definitions 2 and 3 were motivated by designing solvers for (1).

**Definition 2** (Relative Lipschitzness [CST21]). *We say an operator* $g : \mathcal{Z} \to \mathcal{Z}^*$ *is* $\frac{1}{\eta}$*-relatively Lipschitz with respect to* $r : \mathcal{Z} \to \mathbb{R}$ *if for all* $(z, z', z^+) \in \mathcal{Z} \times \mathcal{Z} \times \mathcal{Z}$,

$$\eta \left\langle g(z') - g(z), z' - z^+ \right\rangle \leq V_z^r(z') + V_{z'}^r(z^+).$$

**Definition 3** (Area convexity [She17]). *We say convex* $r : \mathcal{Z} \to \mathbb{R}$ *is* $\eta$*-area convex with respect to an operator* $g : \mathcal{Z} \to \mathcal{Z}^*$ *if for all* $(z, z', z^+) \in \mathcal{Z} \times \mathcal{Z} \times \mathcal{Z}$, *defining* $c := \frac{1}{3}(z + z' + z^+)$,

$$\eta \left\langle g(z') - g(z), z' - z^+ \right\rangle \leq r(z) + r(z') + r(z^+) - 3r(c).$$

*Area convexity is monotone in* $\eta$ *as the right-hand side is nonnegative for any* $z, z', z^+$.

As shown in Appendix B, relaxed relative Lipschitzness simultaneously generalizes relative Lipschitzness and area convexity. The relationship is obvious for Definition 2, but the generalization of Definition 3 relies on the following simple observation, which has not previously appeared explicitly:

$$r(z) + r(z') + r(z^+) - 3r(c) = V_z^r(z^+) + V_z^r(z') - 3V_z^r(c) \leq V_z^r(z^+) + V_z^r(z'). \tag{8}$$

In Appendix B we give an analysis framework unifying previous analyses of [Nem04, Nes07, She17, CST21], possibly of further utility. Our algorithms for box-simplex and box-spectrahedra games employ variants of our relaxed relative Lipschitzness solver in Appendix B, extending it to robustly handle inexactness from subproblem solves (and other approximate computations involving matrices). Finally, we will use the following fact from prior work.

**Fact 1** (Lemma 2, [CST21]). *For convex functions* $f, r$, *if* $f$ *is* $L$*-relatively smooth with respect to* $r$, *then* $\partial f$ *is* $L$*-relatively Lipschitz with respect to* $r$.

## 3 Box-simplex games without alternating minimization

In this section, we develop algorithms for computing approximate saddle points to (1). We will follow notation of Section 2.2, and in the context of this section only we let $\mathcal{X} := [-1, 1]^n$, $\mathcal{Y} := \Delta^d$, and $\mathcal{Z} := \mathcal{X} \times \mathcal{Y}$. To minimize notational clutter, we assume throughout the section that $\|\mathbf{A}\|_{1 \to 1} \leq 1$, lifting this assumption in the proof of Theorem 1 only.[9] We also define the gradient operator of (1):

$$g(x, y) := \left( \mathbf{A}y + c, b - \mathbf{A}^\top x \right). \tag{9}$$

---

[9] As we demonstrate in the proof of Theorem 1, this is without loss of generality via rescaling.

We refer to the $x$ and $y$ components of $g(x, y)$ by $g^{\mathsf{x}}(x, y) := \mathbf{A}y + c$ and $g^{\mathsf{y}}(x, y) := b - \mathbf{A}^\top x$. Finally, for notational convenience, we define the Bregman divergence associated with $r_{\mathbf{A}}^{(\alpha)}$ as

$$V^{(\alpha)} := V^{r_{\mathbf{A}}^{(\alpha)}}.$$

## 3.1 Approximation-tolerant extragradient method

We begin by stating two oracles whose guarantees, when combined with Definition 3, give our conceptual algorithm for (1). These oracles can be viewed as approximately implementing steps of the extragradient method in Appendix B (which assumes exact proximal oracle steps, see Definition 11). We refer the reader to Section 3 and Appendix D.2 of [CST21] for a recent tutorial. We develop subroutines which satisfy these relaxed definitions in the following Section 3.2.

**Definition 4** (Gradient step oracle). *For a problem* (1), *we say* $\mathcal{O}_{\mathrm{grad}} : \mathcal{Z} \times \mathcal{Z}^* \to \mathcal{Z}$ *is an* $(\alpha, \beta)$-*gradient step oracle if on input* $(z, v)$, *it returns* $z'$ *such that for all* $u \in \mathcal{Z}$,

$$\langle v, z' - u \rangle \le V_z^{(\alpha+\beta)}(u) - V_{z'}^{(\alpha)}(u) - V_z^{(\alpha)}(z').$$

**Definition 5** (Extragradient step oracle). *For a problem* (1), *we say* $\mathcal{O}_{\mathrm{xgrad}} : \mathcal{Z} \times \mathcal{Z}^* \times \mathcal{Y} \to \mathcal{Z}$ *is an* $(\alpha, \beta)$-*extragradient step oracle if on input* $(z, v, \bar{y})$, *it returns* $(z^+, \bar{y}^+)$ *such that*

$$\begin{aligned}\langle v, z^+ - u \rangle &\le V_z^{(\alpha)}(u) - V_{z^+}^{(\alpha)}(u) - V_z^{(\alpha)}(z^+) \\ &\quad + V_{\bar{y}}^{\beta h}(u^{\mathsf{y}}) - V_{\bar{y}^+}^{\beta h}(u^{\mathsf{y}}) \text{ for all } u = (u^{\mathsf{x}}, u^{\mathsf{y}}) \in \mathcal{Z}.\end{aligned}$$

When $\beta = 0$, Definitions 4 and 5 reduce to the conventional proximal oracle steps used by the extragradient method of [Nem04]. In our solver for (1), we use $\beta > 0$ to compensate for our inexact subproblem solves. The asymmetry in Definitions 4 and 5 reflect an asymmetry in the analyses of extragradient methods. In typical analyses, the regret is bounded for the "gradient oracle" points, but the regret upper bound is stated in terms of the divergences of the "extragradient oracle" points (which our inexact oracles need to compensate for).

We now state some useful properties of $r_{\mathbf{A}}^{(\alpha)}$ adapted from [JST19].

**Lemma 1.** *The following properties of* $r_{\mathbf{A}}^{(\alpha)}$ *hold.*

1. *For* $\alpha \ge \frac{1}{2}$, $r_{\mathbf{A}}^{(\alpha)}$ *is jointly convex over* $\mathcal{Z}$.

2. *For* $\alpha \ge 2$, $r_{\mathbf{A}}^{(\alpha)}$ *is* $\frac{1}{3}$-*area convex with respect to* $g$ *defined in* (9).

*Proof.* We prove generalizations of these statements in Section 4, but briefly comment on both parts. The first is a tightening of Lemma 6 in [JST19], and follows immediately from the specialization of Proposition 1 to the case of diagonal $\{\mathbf{A}_i\}_{i \in [n]}$. The second is a tightening of Lemma 4 in [JST19], and follows immediately from the same diagonal matrix specialization of Corollary 2. $\square$

The utility of our Definitions 3, 4, and 5 reveals itself through the following lemma.

**Lemma 2.** *Let* $z \in \mathcal{Z}$, $\bar{y} \in \mathcal{Y}$, $\alpha \ge 2$, $\beta, \gamma \ge 0$ *and* $0 \le \eta \le \frac{1}{3}$. *Let* $z' \leftarrow \mathcal{O}_{\mathrm{grad}}(z, \eta g(z))$ *and* $(z^+, \bar{y}^+) \leftarrow \mathcal{O}_{\mathrm{xgrad}}(z, \frac{\eta}{2} g(z'), \bar{y})$, *where* $\mathcal{O}_{\mathrm{grad}}$ *is an* $(\alpha, \beta)$-*gradient step oracle and* $\mathcal{O}_{\mathrm{xgrad}}$ *is an* $(\alpha + \beta, \gamma)$-*extragradient step oracle. Then for all* $u \in \mathcal{Z}$,

$$\langle \eta g(z'), z' - u \rangle \le 2V_z^{(\alpha+\beta)}(u) - 2V_{z^+}^{(\alpha+\beta)}(u) + 2V_{\bar{y}}^{\gamma h}(u^{\mathsf{y}}) - 2V_{\bar{y}^+}^{\gamma h}(u^{\mathsf{y}}).$$

*Proof.* By definition of $\mathcal{O}_{\text{grad}}$ (with $u \leftarrow z^+$) and $\mathcal{O}_{\text{xgrad}}$, we have

$$
\begin{aligned}
\langle \eta g(z), z' - z^+ \rangle &\leq V_z^{(\alpha+\beta)}(z^+) - V_{z'}^{(\alpha)}(z^+) - V_z^{(\alpha)}(z'), \\
\langle \eta g(z'), z^+ - u \rangle &\leq 2V_z^{(\alpha+\beta)}(u) - 2V_{z^+}^{(\alpha+\beta)}(u) - 2V_z^{(\alpha+\beta)}(z^+) \\
&\quad + 2V_{\bar{y}}^{\gamma h}(u^{\mathsf{y}}) - 2V_{\bar{y}^+}^{\gamma h}(u^{\mathsf{y}}).
\end{aligned}
\tag{10}
$$

Combining yields

$$
\begin{aligned}
\langle \eta g(z'), z' - u \rangle &\leq 2V_z^{(\alpha+\beta)}(u) - 2V_{z^+}^{(\alpha+\beta)}(u) + 2V_{\bar{y}}^{\gamma h}(u^{\mathsf{y}}) - 2V_{\bar{y}^+}^{\gamma h}(u^{\mathsf{y}}) \\
&\quad + \eta \left\langle g(z') - g(z), z' - z^+ \right\rangle - V_z^{(\alpha)}(z') - V_z^{(\alpha+\beta)}(z^+) - V_{z'}^{(\alpha)}(z^+) \\
&\leq 2V_z^{(\alpha+\beta)}(u) - 2V_{z^+}^{(\alpha+\beta)}(u) + 2V_{\bar{y}}^{\gamma h}(u^{\mathsf{y}}) - 2V_{\bar{y}^+}^{\gamma h}(u^{\mathsf{y}}) \\
&\quad + \eta \left\langle g(z') - g(z), z' - z^+ \right\rangle - V_z^{(\alpha)}(z') - V_z^{(\alpha)}(z^+),
\end{aligned}
$$

where in the second inequality we used that $V^{(\alpha+\beta)}$ dominates $V^{(\alpha)}$, and $V_{z'}^{(\alpha)}(z^+) \geq 0$ by Lemma 1. The conclusion follows by applying Definition 3 (see (8)) and the second fact in Lemma 1. $\qquad\square$

It is straightforward to check that when $\beta = \gamma = 0$, the proof of Lemma 2 is exactly the same as the analysis in Appendix B, and hence it yields similar implications as standard extragradient methods. In particular, a scaling of the left-hand side upper bounds duality gap of the point $z'$, and the right-hand side telescopes (and may be bounded using the following fact).

**Lemma 3.** *Let* $\alpha, \gamma \geq 0$, *and let* $z_0 = (x_0, y_0)$ *where* $x_0 = \mathbb{0}_n$ *and* $y_0 = \frac{1}{d}\mathbb{1}_d$. *Then* $z_0$ *is the minimizer of* $r_{\mathbf{A}}^{(\alpha)}$ *over* $\mathcal{Z}$, *and* $V_{z_0}^{(\alpha)}(u) \leq 1 + \alpha \log d$, $V_{y_0}^{\gamma h}(u^{\mathsf{y}}) \leq \gamma \log d$ *for all* $u = (u^{\mathsf{x}}, u^{\mathsf{y}}) \in \mathcal{Z}$.

*Proof.* We can verify that $z_0$ minimizes $r_{\mathbf{A}}^{(\alpha)}$ by computing $\nabla r_{\mathbf{A}}^{(\alpha)}(z_0)$ and checking that it is orthogonal to $z - z_0$ for all $z \in \mathcal{Z}$. Moreover by first-order optimality of $z_0$, we have the first conclusion:

$$
V_{z_0}^{(\alpha)}(u) = r_{\mathbf{A}}^{(\alpha)}(u) - r_{\mathbf{A}}^{(\alpha)}(z_0) - \left\langle \nabla r_{\mathbf{A}}^{(\alpha)}(z_0), u - z_0 \right\rangle = r_{\mathbf{A}}^{(\alpha)}(u) - r_{\mathbf{A}}^{(\alpha)}(z_0) \leq 1 + \alpha \log d,
$$

where we used that over $\mathcal{Z}$, the range of $h$ is bounded by $\log d$, and the range of the quadratic portion of $r_{\mathbf{A}}^{(\alpha)}$ is bounded by 1 by using $\|x^2\|_\infty \leq 1$ and $\|y\|_1 = 1$. The second conclusion is similar, where we again use the range of $h$ and that $y_0$ minimizes it. $\qquad\square$

Finally, for convenience to the reader, we put together Lemma 2 and 3 to obtain an analysis of the following conceptual "outer loop" extragradient algorithm (subject to the implementation of gradient and extragradient step oracles), Algorithm 1. Our end-to-end algorithm in Section 3.3, Algorithm 4, will be an explicit implementation of the framework in Algorithm 1; we provide a runtime analysis and error guarantee for Algorithm 4 in Theorem 1.

---

**Algorithm 1:** ConceptualBoxSimplex($\mathbf{A}, b, c, \mathcal{O}_{\text{grad}}, \mathcal{O}_{\text{xgrad}}$)

---

**1 Input:** $\mathbf{A} \in \mathbb{R}^{n \times d}$ with $L := \|\mathbf{A}\|_{1 \to 1}$, desired accuracy $\epsilon \in (0, L)$, $\mathcal{O}_{\text{grad}}$ a $(2, 2)$-gradient step oracle, $\mathcal{O}_{\text{xgrad}}$ a $(4, 4)$-extragradient step oracle

**2** Initialize $x_0 \leftarrow \mathbb{0}_n$, $y_0 \leftarrow \frac{1}{d}\mathbb{1}_d$, $\bar{y}_0 \leftarrow \frac{1}{d}\mathbb{1}_d$, $\hat{x} \leftarrow \mathbb{0}_n$, $\hat{y} \leftarrow \mathbb{0}_d$, $T \leftarrow \lceil \frac{6(8 \log d + 1)L}{\epsilon} \rceil$, $\eta \leftarrow \frac{1}{3}$

**3** Rescale $\mathbf{A} \leftarrow \frac{1}{L}\mathbf{A}$, $b \leftarrow \frac{1}{L}b$, $c \leftarrow \frac{1}{L}c$

**4 for** $t = 0$ **to** $T - 1$ **do**

**5** $\quad$ $g_t \leftarrow (\mathbf{A}y_t + c, b - \mathbf{A}^\top x_t)$

**6** $\quad$ $z_t' := (x_t', y_t') \leftarrow \mathcal{O}_{\text{grad}}(z_t, \eta g_t)$

**7** $\quad$ $g_t' \leftarrow (\mathbf{A}y_t' + c, b - \mathbf{A}^\top x_t')$

**8** $\quad$ $(z_{t+1}, \bar{y}_{t+1}) := (x_{t+1}, y_{t+1}, \bar{y}_{t+1}) \leftarrow \mathcal{O}_{\text{xgrad}}(z_t, \frac{\eta}{2}g_t', \bar{y}_t)$

**9 end**

**10 Return:** $(\hat{x}, \hat{y}) \leftarrow \frac{1}{T}\sum_{t=0}^{T-1}(x_t', y_t')$

---

**Corollary 1.** *Algorithm 1 deterministically computes an $\epsilon$-approximate saddle point to (1).*

*Proof.* First, clearly the rescaling in Line 3 multiplies the entire problem (1) by $\frac{1}{L}$, so an $\frac{\epsilon}{L}$-approximate saddle point to the new problem becomes an $\epsilon$-approximate saddle point for the original. Throughout the rest of the proof it suffices to treat $L = 1$. Next, by telescoping and averaging Lemma 2 with $\alpha = \beta = 2$, $\gamma = 4$, and $\eta = \frac{1}{3}$, we have for $z_0 = (x_0, y_0)$ and any $u = (u^{\mathsf{x}}, u^{\mathsf{y}}) \in \mathcal{Z}$

$$\frac{1}{T}\sum_{t=0}^{T-1} \langle g(z_t'), z_t' - u \rangle \leq \frac{6\left(V_{z_0}^{(4)}(u) + V_{\bar{y}_0}^{4Lh}(u^{\mathsf{y}})\right)}{T} \leq \epsilon.$$

The last inequality used the bounds in Lemma 3 and the definition of $T$. Moreover since $g$ is bilinear, and $\hat{z} := (\hat{x}, \hat{y})$ is the average of the $z_t'$ iterates, we have $\langle g(\hat{z}), \hat{z} - u \rangle \leq \epsilon$. Taking the supremum over $u \in \mathcal{Z}$ bounds the duality gap of $\hat{z}$ and gives the conclusion. $\qquad\square$

## 3.2 Implementing oracles

In this section, we give generic constructions of gradient and extragradient step oracles. We will rely on the following claims on optimizing jointly convex functions of two variables.

**Lemma 4.** *Let $\mathcal{X} \subseteq \mathbb{R}^m$ and $\mathcal{Y} \subseteq \mathbb{R}^n$ be convex compact subsets. Suppose $F : \mathcal{X} \times \mathcal{Y} \to \mathbb{R}$ is jointly convex over its argument $(x, y) \in \mathcal{X} \times \mathcal{Y}$. For $y \in \mathcal{Y}$, define $x_{\text{br}}(y) := \operatorname{argmin}_{x \in \mathcal{X}} F(x, y)$ and $f(y) := F(x_{\text{br}}(y), y)$. Then for all $y \in \mathcal{Y}$, $\partial_y F(x_{\text{br}}(y), y) \subset \partial f(y)$.*

*Proof.* Let $x := x_{\text{br}}(y)$, $z \in \mathcal{Y}$, and $w := x_{\text{br}}(z)$. We first claim that $0 \in \partial_x F(x, y)$. To see this, the definition of the subgradient set implies that it suffices to show for all $x' \in \mathcal{X}$, $F(x, y) \leq F(x', y)$, which is true by definition. Hence by convexity of $F$ from $(x, y)$ to $(w, z)$, we have the desired

$$f(z) = F(w, z) \geq F(x, y) + \langle \partial_y F(x, y), z - y \rangle = f(y) + \langle \partial_y F(x, y), z - y \rangle.$$

$\qquad\square$

**Lemma 5.** *In the setting of Lemma 4, suppose for any $x \in \mathcal{X}$, $F(x, \cdot)$ (as a function over $\mathcal{Y}$) always is $r : \mathcal{Y} \to \mathbb{R}$ plus a linear function (where the linear function may depend on $x$). Then $r - f$ is convex, and $f - q$ is convex for any $q : \mathcal{Y} \to \mathbb{R}$ such that $F - q : \mathcal{X} \times \mathcal{Y} \to \mathbb{R}$ is jointly convex.*

*Proof.* For the first claim, for any $y, z \in \mathcal{Y}$, letting $x := x_{\mathrm{br}}(y)$, we have

$$(r(z) - r(y) - \langle \partial r(y), z - y \rangle) - (f(z) - f(y) - \langle \partial f(y), z - y \rangle)$$
$$= (F(x, z) - F(x, y) - \langle \partial_y F(x, y), z - y \rangle) - (f(z) - f(y) - \langle \partial f(y), z - y \rangle) = F(x, z) - f(z) \geq 0.$$

The first equality used that the first-order expansion of $F(x, \cdot)$ agrees with the first-order expansion of $r$ (as they only differ by a linear term), and the second equality used $F(x, y) = f(y)$ and Lemma 4. The only inequality used the definition of $f$. For the second claim, note that $x_{\mathrm{br}}(y)$ also minimizes $F - q$ over $\mathcal{X}$ for any fixed $y$, as $q(y)$ is a constant in this objective. Since $F - q$ is convex and partial minimization of a convex function preserves convexity, we have the conclusion. $\square$

The first part of Lemma 5 implies a relative smoothness statement, i.e. if jointly convex $F$ equals $r$ up to a linear term, then minimizing $F$ over $\mathcal{X}$ yields a function which is relatively smooth with respect to $r$. The second part implies an analogous relative strong convexity statement. In Appendix A, we show these implications yield a linearly-convergent method for the subproblems in algorithms for (1) using $r_{\mathbf{A}}^{(\alpha)}$, via off-the-shelf tools from [LFN18]. This observation already matches the subproblem solver in [She17] without relying on multiplicative stability properties.

---

**Algorithm 2:** GradStepOracle$(z, v, \alpha, \beta)$

---

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

*Proof.* The optimality conditions for $y^+$ (with respect to $\bar{y}^+$) and $\bar{y}^+$ (with respect to $u^{\mathsf{y}} \in \mathcal{Y}$) yield:

$$\left\langle v^{\mathsf{y}} + \nabla_y r_{\mathbf{A}}^{(\alpha)}(x_{\mathrm{br}}(\bar{y}), \bar{y}) - \nabla_y r_{\mathbf{A}}^{(\alpha)}(z), y^+ - \bar{y}^+ \right\rangle \leq V_{\bar{y}}^{\beta h}(\bar{y}^+) - V_{y^+}^{\beta h}(\bar{y}^+) - V_{\bar{y}}^{\beta h}(y^+),$$

$$\left\langle v^{\mathsf{y}} + \nabla_y r_{\mathbf{A}}^{(\alpha)}(z^+) - \nabla_y r_{\mathbf{A}}^{(\alpha)}(z), \bar{y}^+ - u^{\mathsf{y}} \right\rangle \leq V_{\bar{y}}^{\beta h}(u^{\mathsf{y}}) - V_{\bar{y}^+}^{\beta h}(u^{\mathsf{y}}) - V_{\bar{y}}^{\beta h}(\bar{y}^+).$$

Combining the above gives

$$\left\langle v^{\mathsf{y}} + \nabla_y r_{\mathbf{A}}^{(\alpha)}(z^+) - \nabla_y r_{\mathbf{A}}^{(\alpha)}(z), y^+ - u^{\mathsf{y}} \right\rangle \leq V_{\bar{y}}^{\beta h}(u^{\mathsf{y}}) - V_{\bar{y}^+}^{\beta h}(u^{\mathsf{y}}) - V_{y^+}^{\beta h}(\bar{y}^+) - V_{\bar{y}}^{\beta h}(y^+)$$
$$+ \left\langle \nabla_y r_{\mathbf{A}}^{(\alpha)}(z^+) - \nabla_y r_{\mathbf{A}}^{(\alpha)}(x_{\mathrm{br}}(\bar{y}), \bar{y}), y^+ - \bar{y}^+ \right\rangle, \tag{17}$$

where we observe that

$$\left\langle v^{\mathsf{y}} + \nabla_y r_{\mathbf{A}}^{(\alpha)}(z^+) - \nabla_y r_{\mathbf{A}}^{(\alpha)}(z), y^+ - u^{\mathsf{y}} \right\rangle - \left\langle \nabla_y r_{\mathbf{A}}^{(\alpha)}(z^+) - \nabla_y r_{\mathbf{A}}^{(\alpha)}(x_{\mathrm{br}}(\bar{y}), \bar{y}), y^+ - \bar{y}^+ \right\rangle$$
$$= \left\langle v^{\mathsf{y}} + \nabla_y r_{\mathbf{A}}^{(\alpha)}(z^+) - \nabla_y r_{\mathbf{A}}^{(\alpha)}(z), \bar{y}^+ - u^{\mathsf{y}} \right\rangle + \left\langle v^{\mathsf{y}} + \nabla_y r_{\mathbf{A}}^{(\alpha)}(x_{\mathrm{br}}(\bar{y}), \bar{y}) - \nabla_y r_{\mathbf{A}}^{(\alpha)}(z), y^+ - \bar{y}^+ \right\rangle.$$

Next, we claim that (following the notation (13)), analogously to (15),

$$\left\langle \nabla_y r_{\mathbf{A}}^{(\alpha)}(z^+) - \nabla_y r_{\mathbf{A}}^{(\alpha)}(x_{\mathrm{br}}(\bar{y}), \bar{y}), y^+ - \bar{y}^+ \right\rangle = \left\langle \partial f(y^+) - \partial f(\bar{y}), y^+ - \bar{y}^+ \right\rangle$$
$$\leq V_{y^+}^{\beta h}(\bar{y}^+) + V_{\bar{y}}^{\beta h}(y^+). \tag{18}$$

Plugging (18) into (17) gives

$$\left\langle v^{\mathsf{y}} + \nabla_y r_{\mathbf{A}}^{(\alpha)}(z^+) - \nabla_y r_{\mathbf{A}}^{(\alpha)}(z), y^+ - u^{\mathsf{y}} \right\rangle \leq V_{\bar{y}}^{\beta h}(u^{\mathsf{y}}) - V_{\bar{y}^+}^{\beta h}(u^{\mathsf{y}}),$$

and combined with first-order optimality of $x^+$ with respect to $u^\mathsf{x} \in \mathcal{X}$ again gives for $u = (u^\mathsf{x}, u^\mathsf{y})$,

$$\left\langle v + \nabla r_\mathbf{A}^{(\alpha)}(z^+) - \nabla r_\mathbf{A}^{(\alpha)}(z), z^+ - u \right\rangle \le V_{\bar{y}}^{\beta h}(u^\mathsf{y}) - V_{\bar{y}^+}^{\beta h}(u^\mathsf{y}).$$

The conclusion again follows by using the identity (4). $\qquad\square$

### 3.3 Algorithm

Finally, we put the pieces of this section together to prove a convergence rate on Algorithm 4.

---

**Algorithm 4:** BoxSimplex$(\mathbf{A}, b, c, \epsilon)$

---

**1 Input:** $\mathbf{A} \in \mathbb{R}^{n \times d}$ with $L := \|\mathbf{A}\|_{1 \to 1}$, desired accuracy $\epsilon \in (0, L)$

**2** Initialize $x_0 \leftarrow \mathbb{0}_n$, $y_0 \leftarrow \frac{1}{d}\mathbb{1}_d$, $\bar{y}_0 \leftarrow \frac{1}{d}\mathbb{1}_d$, $\hat{x} \leftarrow \mathbb{0}_n$, $\hat{y} \leftarrow \mathbb{0}_d$, $T \leftarrow \lceil \frac{6(8\log d + 1)L}{\epsilon} \rceil$

**3** Rescale $\mathbf{A} \leftarrow \frac{1}{L}\mathbf{A}$, $b \leftarrow \frac{1}{L}b$, $c \leftarrow \frac{1}{L}c$

**4 for** $t = 0$ **to** $T - 1$ **do**

**5** $\quad (g_t^\mathsf{x}, g_t^\mathsf{y}) \leftarrow \frac{1}{3}(\mathbf{A}y_t + c, b - \mathbf{A}^\top x_t)$ $\qquad\qquad\qquad$ // Gradient oracle start.

**6**

**7** $\quad x_t^\star \leftarrow \mathbf{\Pi}_\mathcal{X}\left(-\frac{g_t^\mathsf{x} - 2\mathbf{diag}(x_t)|\mathbf{A}|y_t}{2|\mathbf{A}|y_t}\right)$

**8** $\quad y_t' \leftarrow \mathbf{\Pi}_\mathcal{Y}\left(y_t \circ \exp\left(-\frac{1}{\beta}(g_t^\mathsf{y} + |\mathbf{A}|^\top(x_t^\star)^2 - |\mathbf{A}|^\top x_t^2)\right)\right)$

**9** $\quad x_t' \leftarrow \mathbf{\Pi}_\mathcal{X}\left(-\frac{g_t^\mathsf{x} - 2\mathbf{diag}(x_t)|\mathbf{A}|y_t}{2|\mathbf{A}|y_t'}\right)$

**10** $\quad (\hat{x}, \hat{y}) \leftarrow (\hat{x}, \hat{y}) + \frac{1}{T}(x_t', y_t')$ $\qquad\qquad\qquad\qquad$ // Running average maintenance.

**11**

**12** $\quad (g_t^\mathsf{x}, g_t^\mathsf{y}) \leftarrow \frac{1}{6}(\mathbf{A}y_t' + c, b - \mathbf{A}^\top x_t')$ $\qquad\qquad\qquad$ // Extragradient oracle start.

**13**

**14** $\quad \bar{x}_t^\star \leftarrow \mathbf{\Pi}_\mathcal{X}\left(-\frac{g_t^\mathsf{x} - 2\mathbf{diag}(x_t)|\mathbf{A}|y_t}{2|\mathbf{A}|\bar{y}_t}\right)$

**15** $\quad y_{t+1} \leftarrow \mathbf{\Pi}_\mathcal{Y}\left(\bar{y}_t \circ \exp\left(-\frac{1}{\beta}(g_t^\mathsf{y} + |\mathbf{A}|^\top(\bar{x}_t^\star)^2 + \alpha\log\bar{y}_t - |\mathbf{A}|^\top x_t^2 - \alpha\log y_t)\right)\right)$

**16** $\quad x_{t+1} \leftarrow \mathbf{\Pi}_\mathcal{X}\left(-\frac{g_t^\mathsf{x} - 2\mathbf{diag}(x_t)|\mathbf{A}|y_t}{2|\mathbf{A}|y_{t+1}}\right)$

**17** $\quad \bar{y}_{t+1} \leftarrow \mathbf{\Pi}_\mathcal{Y}\left(y_t \circ \exp\left(-\frac{1}{\beta}(g_t^\mathsf{y} + |\mathbf{A}|^\top(x_{t+1})^2 + \alpha\log y_{t+1} - |\mathbf{A}|^\top x_t^2 - \alpha\log y_t)\right)\right)$

**18 end**

**19 Return:** $(\hat{x}, \hat{y})$

---

**Theorem 1.** *Algorithm 4 deterministically computes an $\epsilon$-approximate saddle point to (1) in time*[10]

$$O\left(\mathrm{nnz}(\mathbf{A}) \cdot \frac{\|\mathbf{A}\|_{1 \to 1}\log d}{\epsilon}\right).$$

*Proof.* By observation, Lines 5 to 9 implement Algorithm 2 (used in Lemma 6), with the parameters required by Lemma 2. Similarly, Lines 12 to 17 implement Algorithm 3 (used in Lemma 7), with the parameters required by Lemma 2. Correctness thus follows from Corollary 1. For the runtime, without loss of generality $\mathrm{nnz}(\mathbf{A}) \ge \max(n, d)$ (else we may drop columns or rows appropriately), and each of $T$ iterations is dominated by a constant number of matrix-vector multiplications. $\qquad\square$

---

[10]This statement assumes direct access to the entries of $\mathbf{A}$. More generally, letting $\mathcal{T}_{\mathrm{mv}}$ be the time it takes to multiply a vector by the argument, we may replace $\mathrm{nnz}(\mathbf{A})$ by $\max(\mathcal{T}_{\mathrm{mv}}(\mathbf{A}), \mathcal{T}_{\mathrm{mv}}(\mathbf{A}^\top), \mathcal{T}_{\mathrm{mv}}(|\mathbf{A}|), \mathcal{T}_{\mathrm{mv}}(|\mathbf{A}^\top|))$ where $|\cdot|$ is entrywise, which may be a significant savings when $\mathbf{A}$ is implicitly accessible and nonnegative. An $\epsilon$-approximate saddle point to a minimax problem is a point with duality gap $\epsilon$; see Section 2.

# 4 Box-spectraplex games

In this section, we develop algorithms for computing approximate saddle points to (2). We will follow the notation of Section 2.3, and in the context of this section only we let $\mathcal{X} := [-1, 1]^n$, $\mathcal{Y} := \Delta^{d \times d}$, and $\mathcal{Z} := \mathcal{X} \times \mathcal{Y}$. As in Section 3, we assume throughout the section that (following (6)) $L_{\mathcal{A}} \leq 1$, except when proving Theorem 3. We also define the gradient operator of (2):

$$g(x, \mathbf{Y}) := (\mathcal{A}^*(\mathbf{Y}) + c, \mathbf{B} - \mathcal{A}(x)). \tag{19}$$

We refer to the $x$ and $\mathbf{Y}$ components of $g(x, \mathbf{Y})$ by $g^{\mathsf{x}}(x, \mathbf{Y}) := \mathcal{A}^*(\mathbf{Y}) + c$ and $g^{\mathsf{y}}(x, \mathbf{Y}) := \mathbf{B} - \mathcal{A}(x)$. Finally, for notational convenience, we define the Bregman divergence associated with $r_{\mathcal{A}}^{(\alpha, \mu)}$ as

$$V^{(\alpha, \mu)} := V^{r_{\mathcal{A}}^{(\alpha, \mu)}}.$$

When $\mu = 0$, we will simply denote the divergence as $V^{(\alpha)}$.

## 4.1 Regularizer properties

In this section, we state properties about our regularizer $r_{\mathcal{A}}^{(\alpha, \mu)}$, used to prove the following.

**Proposition 1.** *Let $\mathcal{A} := \{\mathbf{A}_i\}_{i \in [n]} \subset \mathbb{S}^d$. Let $\mathbf{J} \in \mathbb{R}^{(n+d^2) \times (n+d^2)}$ be an associated (skew-symmetric) linear operator mapping $\mathbb{R}^n \times \mathbb{S}^d \to \mathbb{R}^n \times \mathbb{S}^d$, such that for all $v \in \mathbb{R}^n$ and $\mathbf{M} \in \mathbb{S}^d$,*

$$\mathbf{J}(v, \mathbf{M}) = (\mathcal{A}^*(\mathbf{M}), -\mathcal{A}(v)).$$

*$r_{\mathcal{A}}^{(\alpha, \mu)}$ satisfies the following properties:*

- *For any $\alpha \geq \frac{1}{2}$, $\mu \geq 0$, $r_{\mathcal{A}}^{(\alpha, \mu)}$ is jointly convex over $[-1, 1]^n \times \Delta^{d \times d}$.*

- *For $\alpha \geq 2, \mu \geq 0$ and any $(x, \mathbf{Y}) \in [-1, 1] \times \Delta^{d \times d}$, the following matrix is positive semidefinite:*

$$\begin{pmatrix} \nabla^2 r_{\mathcal{A}}^{(\alpha, \mu)}(x, \mathbf{Y}) & -\mathbf{J} \\ \mathbf{J}^\top & \nabla^2 r_{\mathcal{A}}^{(\alpha, \mu)}(x, \mathbf{Y}) \end{pmatrix}.$$

This proposition generalizes Lemma 1, which was proven (up to constant factors) in [JST19]. However, unlike the vector-vector setting, the Hessian of matrix entropy is significantly less well-behaved due to monotonicity bounds which do not apply to exp, a function which is not operator monotone. Our proofs make use of the following nontrivial lower bound on the Hessian of $H(\mathbf{Y})$ in Appendix C, where we recall from Section 2 that $H$ is the negated von Neumann entropy function.

**Lemma 8.** *Let $\mathcal{A} := \{\mathbf{A}_i\}_{i \in [n]} \subset \mathbb{S}_{\succeq \mathbf{0}}^d$ satisfy $\sum_{i \in [n]} \mathbf{A}_i = \mathbf{I}_d$. For any $\mathbf{M} \in \mathbb{S}^d$ and $\mathbf{Y} \in \mathbb{S}_{\succ \mathbf{0}}^d$ we have $\nabla^2 H(\mathbf{Y})[\mathbf{M}, \mathbf{M}] \geq \nabla^2 h(y)[m, m]$ for $y := \mathcal{A}^*(\mathbf{Y})$, $m := \mathcal{A}^*(\mathbf{M})$.*

**Lemma 9.** *Let $\mathcal{A} := \{\mathbf{A}_i\}_{i \in [n]} \subset \mathbb{S}^d$ satisfy (6). For $\mathbf{Y} \in \Delta^{d \times d}$, $\tau > 0$, vectors $v \in \mathbb{R}^n$ and $x \in [-1, 1]^n$, and matrix $\mathbf{M} \in \mathbb{S}^d$, defining $v \circ x$ to be the entrywise product of vectors,*

$$2 \langle v \circ x, \mathcal{A}^*(\mathbf{M}) \rangle \leq \tau \mathbf{diag}\left(|\mathcal{A}|^*(\mathbf{Y})\right)[v, v] + \frac{1}{\tau} \nabla^2 H(\mathbf{Y})[\mathbf{M}, \mathbf{M}].$$

*Proof.* By the assumption that $L_{\mathcal{A}} \leq 1$, $\sum_{i \in [n]} |\mathbf{A}|_i \preceq \mathbf{I}$. For all $i \in [n]$ define $\mathbf{A}_i^+, \mathbf{A}_i^- \in \mathbb{S}_{\succeq \mathbf{0}}^d$ such that $|\mathbf{A}| = \mathbf{A}_i^+ + \mathbf{A}_i^-$, where $\mathbf{A}_i^+$ keeps the positive eigenvalues of $\mathbf{A}$ (with the same eigenspaces), and $\mathbf{A}_i^-$ negates the negative eigenvalues; note also that $\mathbf{A}_i = \mathbf{A}_i^+ - \mathbf{A}_i^-$. Finally let $\mathbf{A}_0 = \mathbf{I} - \sum_{i \in [n]} |\mathbf{A}_i|$ (which is positive semidefinite since $L_{\mathcal{A}} \leq 1$) and let $\mathcal{A}' = \{\mathbf{A}_0\} \cup \{\mathbf{A}_i^+, \mathbf{A}_i^-\}_{i \in [n]}$. By Lemma 8 applied to the set of matrices $\mathcal{A}'$ we conclude

$$\nabla^2 H(\mathbf{Y})[\mathbf{M}, \mathbf{M}] \geq \sum_{i \in [n]} \left( \frac{\langle \mathbf{A}_i^+, \mathbf{M} \rangle^2}{\langle \mathbf{A}_i^+, \mathbf{Y} \rangle} + \frac{\langle \mathbf{A}_i^-, \mathbf{M} \rangle^2}{\langle \mathbf{A}_i^-, \mathbf{Y} \rangle} \right), \tag{20}$$

where we drop the (nonnegative) term in Lemma 8 corresponding to the diagonal element $\langle \mathbf{A}_0, \mathbf{Y} \rangle$. The conclusion follows by plugging in the above inequality, yielding

$$\tau \mathbf{diag} \left( |\mathcal{A}|^*(\mathbf{Y}) \right) [v, v] + \frac{1}{\tau} \sum_{i \in [n]} \left( \frac{\langle \mathbf{A}_i^+, \mathbf{M} \rangle^2}{\langle \mathbf{A}_i^+, \mathbf{Y} \rangle} + \frac{\langle \mathbf{A}_i^-, \mathbf{M} \rangle^2}{\langle \mathbf{A}_i^-, \mathbf{Y} \rangle} \right)$$

$$= \sum_{i \in [n]} \tau v_i^2 \left( \langle \mathbf{A}_i^+, \mathbf{Y} \rangle + \langle \mathbf{A}_i^-, \mathbf{Y} \rangle \right) + \frac{1}{\tau} \left( \frac{\langle \mathbf{A}_i^+, \mathbf{M} \rangle^2}{\langle \mathbf{A}_i^+, \mathbf{Y} \rangle} + \frac{\langle \mathbf{A}_i^-, \mathbf{M} \rangle^2}{\langle \mathbf{A}_i^-, \mathbf{Y} \rangle} \right)$$

$$\geq 2 \sum_{i \in [n]} v_i x_i \left( \langle \mathbf{A}_i^+, \mathbf{M} \rangle - \langle \mathbf{A}_i^-, \mathbf{M} \rangle \right) = 2 \langle v \circ x, \mathcal{A}^*(\mathbf{M}) \rangle.$$

The last line above used Young's inequality which shows for all $i \in [n]$ (recalling $|x_i| \leq 1$)

$$2 v_i x_i \langle \mathbf{A}_i^+, \mathbf{M} \rangle \leq \tau v_i^2 \langle \mathbf{A}_i^+, \mathbf{Y} \rangle + \frac{1}{\tau} \cdot \frac{\langle \mathbf{A}_i^+, \mathbf{M} \rangle^2}{\langle \mathbf{A}_i^+, \mathbf{Y} \rangle},$$

$$-2 v_i x_i \langle \mathbf{A}_i^-, \mathbf{M} \rangle \leq \tau v_i^2 \langle \mathbf{A}_i^-, \mathbf{Y} \rangle + \frac{1}{\tau} \cdot \frac{\langle \mathbf{A}_i^-, \mathbf{M} \rangle^2}{\langle \mathbf{A}_i^-, \mathbf{Y} \rangle}.$$

$\square$

With this bound, we prove Proposition 1.

*Proof of Proposition 1.* For both claims, it suffices to show the case $\mu = 0$, as the sum of convex functions is convex and the sum of positive semidefinite matrices is positive semidefinite. We begin with the first claim. Joint convexity of $r_{\mathcal{A}}^{(\alpha)}$ is equivalent to showing that the quadratic form of $\nabla^2 r_{\mathcal{A}}^{(\alpha)}$ (viewed as a $(n + d^2) \times (n + d^2)$ matrix) with respect to $(v, \mathbf{M})$ is nonnegative for any $v \in \mathbb{R}^n, \mathbf{M} \in \mathbb{S}^d$: in other words

$$\alpha \nabla^2 H(\mathbf{Y})[\mathbf{M}, \mathbf{M}] + 2 \langle v \circ x, |\mathcal{A}|^*(\mathbf{M}) \rangle + 2 \mathbf{diag} \left( |\mathcal{A}|^*(\mathbf{Y}) \right) [v, v] \geq 0.$$

This follows from Lemma 9 with $\tau = 2$, where we replace $x \leftarrow -x$ and use $\alpha \geq \frac{1}{2}$. For the second claim, let $v, u \in \mathbb{R}^n$ and $\mathbf{M}, \mathbf{N} \in \mathbb{S}^d$. Consider the quadratic form of

$$\begin{pmatrix} \nabla^2 r_{\mathcal{A}}^{(\alpha)}(x, \mathbf{Y}) & -\mathbf{J} \\ \mathbf{J}^\top & \nabla^2 r_{\mathcal{A}}^{(\alpha)}(x, \mathbf{Y}) \end{pmatrix}.$$

with $(v, \mathbf{M}, u, \mathbf{N})$. We obtain

$$\alpha \left( \nabla^2 H(\mathbf{Y})[\mathbf{M}, \mathbf{M}] + \nabla^2 H(\mathbf{Y})[\mathbf{N}, \mathbf{N}] \right) + 2 \mathbf{diag} \left( |\mathcal{A}|^*(\mathbf{Y}) \right) [v, v] + 2 \mathbf{diag} \left( |\mathcal{A}|^*(\mathbf{Y}) \right) [u, u]$$

$$+ 2 \langle v \circ x, |\mathcal{A}|^*(\mathbf{M}) \rangle + 2 \langle u \circ x, |\mathcal{A}|^*(\mathbf{N}) \rangle + 2 \langle v, \mathcal{A}^*(\mathbf{N}) \rangle - 2 \langle u, \mathcal{A}^*(\mathbf{M}) \rangle. \tag{21}$$

By applying Lemma 9 with $\tau = 1$, we have

$$-2\langle v \circ x, |\mathcal{A}|^*(\mathbf{M})\rangle \leq \mathbf{diag}\left(|\mathcal{A}|^*(\mathbf{Y})\right)[v,v] + \nabla^2 H(\mathbf{Y})[\mathbf{M},\mathbf{M}],$$
$$-2\langle u \circ x, |\mathcal{A}|^*(\mathbf{N})\rangle \leq \mathbf{diag}\left(|\mathcal{A}|^*(\mathbf{Y})\right)[u,u] + \nabla^2 H(\mathbf{Y})[\mathbf{N},\mathbf{N}],$$
$$-2\langle v, \mathcal{A}^*(\mathbf{N})\rangle \leq \mathbf{diag}\left(|\mathcal{A}|^*(\mathbf{Y})\right)[v,v] + \nabla^2 H(\mathbf{Y})[\mathbf{N},\mathbf{N}],$$
$$2\langle u, \mathcal{A}^*(\mathbf{M})\rangle \leq \mathbf{diag}\left(|\mathcal{A}|^*(\mathbf{Y})\right)[u,u] + \nabla^2 H(\mathbf{Y})[\mathbf{M},\mathbf{M}].$$

Combining these four equations shows that the quantity in (21) is nonnegative as desired. $\qquad\square$

As a corollary of known tools [She17], this implies that $r_{\mathcal{A}}^{(\alpha,\mu)}$ is an area convex regularizer.

**Corollary 2.** *Let* $g(x, \mathbf{Y})$ *be the gradient operator of* (2). *Then for* $\alpha \geq 2$, $\mu \geq 0$, $r_{\mathcal{A}}^{(\alpha,\mu)}$ *is* $\frac{1}{3}$-*area convex with respect to* $g$ *(Definition 3).*

*Proof.* It suffices to check for $\mu = 0$ as increasing $\mu$ only makes the right-hand side larger. This case follows from the second conclusion of Proposition 1 and Theorem 1.6 of [She17], a generic way of proving area convexity via checking a second-order positive semidefiniteness condition. $\qquad\square$

We collect a few additional tools which we use in harnessing (7) for our algorithms later in this section. We first state a bound on the divergence of $r_{\mathcal{A}}^{(\alpha,\mu)}$ from its minimizer.

**Lemma 10.** *Let* $\alpha, \mu \geq 0$, *and let* $z_0 = (x_0, \mathbf{Y}_0)$ *where* $x_0 = \mathbb{0}_n$ *and* $\mathbf{Y}_0 = \frac{1}{d}\mathbf{I}_d$. *Then* $z_0$ *is the minimizer of* $r_{\mathcal{A}}^{(\alpha,\mu)}$ *over* $\mathcal{Z}$, *and*

$$V_{z_0}^{(\alpha,\mu)}(u) \leq 1 + \alpha \log d + \frac{\mu n}{2} \text{ for all } u = (u^{\mathsf{x}}, u^{\mathsf{y}}) \in \mathcal{Z}.$$

*Proof.* The proof is almost identical to Lemma 3 so we only discuss differences. First, the additive range of the squared regularizer on $\mathcal{X}$ from $x_0$ is bounded by $\frac{\mu n}{2}$. Also, the additive range of von Neumann entropy is $\log d$ (similarly to vector entropy), and it is minimized by $\mathbf{Y}_0$. $\qquad\square$

The gradient of our regularizer $r_{\mathcal{A}}^{(\alpha,\mu)}$ is difficult to evaluate exactly due to the presence of matrix exponentials arising from recursive descent steps. We formalize the approximate gradient access required by our algorithms in the following definition.

**Definition 6** (MEQ oracle). *Let* $\epsilon, \delta, \gamma \in (0,1)$. *We say* $\mathcal{O}_{\mathrm{meq}}$ *is an* $(\epsilon, \delta, \gamma)$-*matrix exponential query (MEQ) oracle for* $\{\mathbf{A}_i\}_{i\in[n]} \subset \mathbb{S}^d$ *and* $\mathbf{M} \in \mathbb{S}^d$, *if it returns* $\{V_i\}_{i\in[n]}$ *such that with probability* $\geq 1 - \delta$, $|V_i - \langle \mathbf{A}_i, \mathbf{Y}\rangle| \leq \epsilon \langle |\mathbf{A}_i|, \mathbf{Y}\rangle + \gamma \mathrm{Tr}(|\mathbf{A}_i|)$ *for all* $i \in [n]$, *where* $\mathbf{Y} := \mathbf{\Pi}_{\mathcal{Y}}(\exp \mathbf{M})$.

Definition 6 returns approximations of all $\langle \mathbf{A}_i, \mathbf{Y}\rangle$ up to $\epsilon$-multiplicative error (in $\mathbf{Y}$'s product through $|\mathbf{A}_i|$, instead of $\mathbf{A}_i$), and an additive $\gamma \mathrm{Tr}(|\mathbf{A}_i|)$ error. In Appendix D, we give an implementation of $\mathcal{O}_{\mathrm{meq}}$ whose runtime depends polynomially on $\epsilon$ and polylogarithmically on $\gamma$.[11]

**Proposition 2.** *Let* $\|\mathbf{M}\|_{\mathrm{op}} \leq R$. *We can implement an* $(\epsilon, \delta, \gamma)$-*MEQ oracle for* $\{\mathbf{A}_i\}_{i\in[n]} \subset \mathbb{S}^d$, $\mathbf{M} \in \mathbb{S}^d$ *in time*

$$O\left(\mathcal{T}_{\mathrm{mv}}(\mathbf{M}) \cdot \sqrt{R + \log\frac{1}{\gamma\epsilon}} \cdot \frac{\log^{1.5}(\frac{Rnd}{\gamma\delta\epsilon})}{\epsilon^2} + \left(\sum_{i\in[n]} \mathcal{T}_{\mathrm{mv}}(\mathbf{A}_i)\right) \cdot \frac{\log\frac{nd}{\delta}}{\epsilon^2}\right).$$

*Proof.* It suffices to combine Lemma 19 and 20 in Appendix D, adjusting $\epsilon$ by a constant factor. $\qquad\square$

---

[11] Similar guarantees appear in the literature on approximately solving SDPs, but we could not find a statement with the additive-multiplicative guarantees our method requires, so we provide a self-contained proof for completeness.

## 4.2 Approximation-tolerant mirror prox

We next provide approximation-tolerant variants of the algorithms in Section 3. This tolerance is necessitated by error introduced by approximations to the matrix exponential we develop in Appendix D. We begin by formalizing the notions of approximation required for our framework.

**Definition 7** (Approximate gradient oracle). *We say $\tilde{g} : \mathcal{Z} \to \mathcal{Z}^*$ is a $\Delta$-approximate gradient oracle for $g : \mathcal{Z} \to \mathcal{Z}^*$ if for all $z, z' \in \mathcal{Z}$, $|\langle \tilde{g}(z) - g(z), z' \rangle| \leq \Delta$.*

**Definition 8** (Approximate best response oracle). *We say $\tilde{x} : \mathcal{X}^* \to \mathcal{X}$ is a $\Delta$-approximate best response oracle for $r : \mathcal{Z} \to \mathbb{R}$ and $\mathbf{Y} \in \mathcal{Y}$ if for all $v \in \mathcal{X}^*$ and $u^{\mathsf{x}} \in \mathcal{X}$, the following hold:*

$$\|\tilde{x}(\mathbf{Y}, v) - (\mathrm{argmin}_{x \in \mathcal{X}} \langle v, x \rangle + r(x, \mathbf{Y}))\|_\infty \leq \Delta,$$
$$\langle v + \nabla_x r(x, \mathbf{Y}), x - u^{\mathsf{x}} \rangle \leq \Delta.$$

We pause to remark that for $g$ in (19), the component $g^{\mathsf{y}}(x, \mathbf{Y}) = \mathbf{B} - \mathcal{A}(x)$ is simple to explicitly write down given $x$, and will incur no error in our implementation. Further, the term $g^{\mathsf{x}}(x, \mathbf{Y}) - c = \mathcal{A}^*(\mathbf{Y})$ can be efficiently approximated to the accuracy required by Definition 7 by taking $\epsilon, \gamma \leftarrow O(\Delta)$ in Definition 6. Under the scaling $L_{\mathcal{A}} \leq 1$, the $\ell_1$ error incurred by $\mathcal{O}_{\mathrm{meq}}$ is $O(\epsilon + \gamma)$, which satisfies Definition 7 as $\mathcal{X}$ is $\ell_\infty$-constrained. In the following Section 4.3 we will exploit the stronger multiplicative-additive guarantee afforded by Definition 6 to meet Definition 8. Equipped with Definitions 7 and 8, we give simple error-tolerant extensions of Sections 3.1 and 3.2.

**Definition 9** (Approximate gradient step oracle). *For a problem (2), we say $\widetilde{\mathcal{O}}_{\mathrm{grad}} : \mathcal{Z} \times \mathcal{Z}^* \to \mathcal{Z}$ is an $(\alpha, \beta, \mu, \Delta)$-approximate gradient step oracle if on input $(z, v)$, it returns $z'$ such that for all $u \in \mathcal{Z}$,*

$$\langle v, z' - u \rangle \leq V_z^{(\alpha+\beta, \mu)}(u) - V_{z'}^{(\alpha, \mu)}(u) - V_z^{(\alpha, \mu)}(z') + \Delta.$$

**Definition 10** (Approximate extragradient step oracle). *For a problem (2), we say $\widetilde{\mathcal{O}}_{\mathrm{xgrad}} : \mathcal{Z} \times \mathcal{Z}^* \times \mathcal{Y} \to \mathcal{Z}$ is an $(\alpha, \beta, \mu, \Delta)$-approximate extragradient step oracle if on input $(z, v, \overline{\mathbf{Y}})$ it returns $(z^+, \overline{\mathbf{Y}}^+)$ such that*

$$\langle v, z^+ - u \rangle \leq V_z^{(\alpha, \mu)}(u) - V_{z^+}^{(\alpha, \mu)}(u) - V_z^{(\alpha, \mu)}(z^+)$$
$$+ V_{\overline{\mathbf{Y}}}^{\beta H}(u^{\mathsf{y}}) - V_{\overline{\mathbf{Y}}^+}^{\beta H}(u^{\mathsf{y}}) + \Delta \text{ for all } u = (u^{\mathsf{x}}, u^{\mathsf{y}}) \in \mathcal{Z}.$$

Definitions 9 and 10 are exactly the same as Definitions 4 and 5, except they are generalized to the matrix setting, tolerate regularizers $V^{(\cdot, \mu)}$ with $\mu \geq 0$, and allow for $\Delta$ error in their bounds.

**Corollary 3.** *Let $z \in \mathcal{Z}$, $\overline{\mathbf{Y}} \in \mathcal{Y}$, $\alpha \geq 2$, $\beta, \gamma \geq 0$, and $0 \leq \eta \leq \frac{1}{3}$. Let $\tilde{g}$ be a $\Delta$-approximate gradient oracle for $g$ in (19). Let $z' \leftarrow \widetilde{\mathcal{O}}_{\mathrm{grad}}(z, \eta\tilde{g}(z))$ and $(z^+, \overline{\mathbf{Y}}^+) \leftarrow \widetilde{\mathcal{O}}_{\mathrm{xgrad}}(z, \frac{\eta}{2}\tilde{g}(z'), \overline{\mathbf{Y}})$, where $\widetilde{\mathcal{O}}_{\mathrm{grad}}$ is an $(\alpha, \beta, \mu, \Delta)$-approximate gradient step oracle and $\widetilde{\mathcal{O}}_{\mathrm{xgrad}}$ is an $(\alpha + \beta, \gamma, \mu, \Delta)$-approximate extragradient step oracle. Then for all $u \in \mathcal{Z}$,*

$$\langle \eta g(z'), z' - u \rangle \leq 2V_z^{(\alpha+\beta, \mu)}(u) - 2V_{z^+}^{(\alpha+\beta, \mu)}(u) + 2V_{\overline{\mathbf{Y}}}^{\gamma H}(u^{\mathsf{y}}) - 2V_{\overline{\mathbf{Y}}^+}^{\gamma H}(u^{\mathsf{y}}) + 5\Delta.$$

*Proof.* The proof is the same as Lemma 2, but we incur error due to the approximate guarantees of $\tilde{g}$, $\widetilde{\mathcal{O}}_{\mathrm{grad}}$, and $\widetilde{\mathcal{O}}_{\mathrm{xgrad}}$. It is straightforward to see that using $\widetilde{\mathcal{O}}_{\mathrm{grad}}$ and $\widetilde{\mathcal{O}}_{\mathrm{xgrad}}$ incurs $3\Delta$ additive error on the right-hand sides of (10). Further, using $\tilde{g}$ instead of $g$ implies that the left-hand sides of (10) hold up to $6\eta\Delta$ error via Hölder's inequality. Combining these errors yields the result. $\square$

---

**Algorithm 5:** ApproxGradStepOracle$(z, v, \alpha, \beta, \mu, \tilde{x}, \widetilde{\nabla}_x r_{\mathcal{A}}^{(\alpha,\mu)}(\cdot, \mathbf{Y}))$

---

**1 Input:** $z = (x, \mathbf{Y}) \in \mathcal{Z}$, $v = (v^{\mathsf{x}}, v^{\mathsf{y}}) \in \mathcal{Z}^*$, $\alpha, \beta, \mu \geq 0$, $\tilde{x}$, a $\Delta$-approximate best response
   oracle for $r_{\mathcal{A}}^{(\alpha,\mu)}$ and $\mathbf{Y}$ or $\mathbf{Y}'$ (defined below), $\widetilde{\nabla}_x r_{\mathcal{A}}^{(\alpha,\mu)}(\cdot, \mathbf{Y})$, a $\Delta$-approximate gradient
   oracle for $\nabla_x r_{\mathcal{A}}^{(\alpha,\mu)}(\cdot, \mathbf{Y})$

**2** $w \leftarrow \widetilde{\nabla}_x r_{\mathcal{A}}^{(\alpha,\mu)}(x, \mathbf{Y})$

**3** $\hat{x} \leftarrow \tilde{x}(\mathbf{Y}, v^{\mathsf{x}} - w)$

**4** $\mathbf{Y}' \leftarrow \mathrm{argmin}_{\widehat{\mathbf{Y}} \in \mathcal{Y}} \left\langle v^{\mathsf{y}} + \nabla_y r_{\mathcal{A}}^{(\alpha,\mu)}(\hat{x}, \mathbf{Y}) - \nabla_y r_{\mathcal{A}}^{(\alpha,\mu)}(x, \mathbf{Y}), \widehat{\mathbf{Y}} \right\rangle + V_{\mathbf{Y}}^{\beta H}\left(\widehat{\mathbf{Y}}\right)$

**5** $x' \leftarrow \tilde{x}(\mathbf{Y}', g^{\mathsf{x}} - w)$

**6 Return:** $(x', \mathbf{Y}')$

---

**Corollary 4.** *For $\beta \geq \alpha \geq \frac{1}{2}$, $\mu \geq 0$, Algorithm 5 is an $(\alpha, \beta, \mu, 11\Delta)$-approximate gradient step oracle.*

*Proof.* The proof is the same as Lemma 6, but we incur error due to the approximate computations of $w$, $\hat{x}$, and $x'$. Let $r := r_{\mathcal{A}}^{(\alpha,\mu)}$ for convenience. Let $x_\star$ minimize the subproblem defining $\hat{x}$, and $x'_\star$ minimize the subproblem defining $x'$.

Lemma 6 combines inequalities (12), (14), and (16), and we will bound the error incurred in each. It is straightforward to check that the approximation guarantee on $\hat{x}$ implies $\left\| \hat{x}^2 - x_\star^2 \right\|_\infty \leq 2\Delta$, and so $\|\nabla_y r(\hat{x}, \mathbf{Y}) - \nabla_y r(x_\star, \mathbf{Y})\|_\infty \leq 2\Delta$. Hence, in place of (12), Hölder's inequality yields

$$
\begin{aligned}
\left\langle v^{\mathsf{y}} + \nabla_y r(x_\star, \mathbf{Y}) - \nabla_y r(x, \mathbf{Y}), \mathbf{Y}' - u^{\mathsf{y}} \right\rangle &= \left\langle v^{\mathsf{y}} + \nabla_y r(\hat{x}, \mathbf{Y}) - \nabla_y r(x, \mathbf{Y}), \mathbf{Y}' - u^{\mathsf{y}} \right\rangle \\
&\quad + \left\langle \nabla_y r(x_\star, \mathbf{Y}) - \nabla_y r(\hat{x}, \mathbf{Y}), \mathbf{Y}' - u^{\mathsf{y}} \right\rangle \\
&\leq V_{\mathbf{Y}}^{\beta H}(u^{\mathsf{y}}) - V_{\mathbf{Y}'}^{\beta H}(u^{\mathsf{y}}) - V_{\mathbf{Y}}^{\beta H}(\mathbf{Y}') + 4\Delta.
\end{aligned}
$$

Further, in place of (14) we have by a similar argument

$$
\begin{aligned}
\left\langle \nabla_y r(x', \mathbf{Y}') - \nabla_y r(x_\star, \mathbf{Y}), \mathbf{Y}' - u^{\mathsf{y}} \right\rangle &\leq \left\langle \nabla_y r(x'_\star, \mathbf{Y}') - \nabla_y r(x_\star, \mathbf{Y}), \mathbf{Y}' - u^{\mathsf{y}} \right\rangle \\
&\quad + \left\langle \nabla_y r(x', \mathbf{Y}') - \nabla_y r(x'_\star, \mathbf{Y}'), \mathbf{Y}' - u^{\mathsf{y}} \right\rangle \\
&\leq V_{\mathbf{Y}}^{\alpha H}(\mathbf{Y}') + V_{\mathbf{Y}'}^{\alpha H}(u^{\mathsf{y}}) + 4\Delta.
\end{aligned}
$$

Finally, by the second condition on the oracle $\tilde{x}$ defining $x'$, in place of (16) we have

$$
\begin{aligned}
\left\langle v^{\mathsf{x}} + \nabla_x r(x', \mathbf{Y}') - w, x' - u^{\mathsf{x}} \right\rangle &\leq \Delta \\
\implies \left\langle v^{\mathsf{x}} + \nabla_x r(x', \mathbf{Y}') - \nabla_x r(x, \mathbf{Y}), x' - u^{\mathsf{x}} \right\rangle &\leq \left\langle v^{\mathsf{x}} + \nabla_x r(x', \mathbf{Y}') - w, x' - u^{\mathsf{x}} \right\rangle \\
&\quad + \left\langle w - \nabla_x r(x, \mathbf{Y}), x' - u^{\mathsf{x}} \right\rangle \leq 3\Delta.
\end{aligned}
$$

The last inequality used the guarantee of the oracle $\widetilde{\nabla}_x r$, and Hölder's inequality. $\square$

---

**Algorithm 6:** ApproXGradStepOracle$(z, v, \overline{\mathbf{Y}}, \alpha, \beta, \mu, \tilde{x}, \widetilde{\nabla}_x r_{\mathcal{A}}^{(\alpha,\mu)}(\cdot, \mathbf{Y}))$

---

**1 Input:** $z = (x, \mathbf{Y}) \in \mathcal{Z}$, $v = (v^{\mathsf{x}}, v^{\mathsf{y}}) \in \mathcal{Z}^*$, $\overline{\mathbf{Y}} \in \mathcal{Y}$, $\alpha, \beta, \mu \geq 0$, $\tilde{x}$, a $\Delta$-approximate best response oracle for $r_{\mathcal{A}}^{(\alpha,\mu)}$ and $\overline{\mathbf{Y}}$ or $\mathbf{Y}^+$ (defined below), $\widetilde{\nabla}_x r_{\mathcal{A}}^{(\alpha,\mu)}(\cdot, \mathbf{Y})$, a $\Delta$-approximate gradient oracle for $\nabla_x r_{\mathcal{A}}^{(\alpha,\mu)}(\cdot, \mathbf{Y})$

**2** $w \leftarrow \widetilde{\nabla}_x r_{\mathcal{A}}^{(\alpha,\mu)}(x, \mathbf{Y})$

**3** $\bar{x} \leftarrow \tilde{x}(\overline{\mathbf{Y}}, v^{\mathsf{x}} - w)$

**4** $\mathbf{Y}^+ \leftarrow \text{argmin}_{\widehat{\mathbf{Y}} \in \mathcal{Y}} \left\langle v^{\mathsf{y}} + \nabla_y r_{\mathcal{A}}^{(\alpha,\mu)}(\bar{x}, \overline{\mathbf{Y}}) - \nabla_y r_{\mathcal{A}}^{(\alpha,\mu)}(x, \mathbf{Y}), \widehat{\mathbf{Y}} \right\rangle + V_{\overline{\mathbf{Y}}}^{\beta H}(\widehat{\mathbf{Y}})$

**5** $x^+ \leftarrow \tilde{x}(\mathbf{Y}^+, v^{\mathsf{x}} - w)$

**6** $\overline{\mathbf{Y}}^+ \leftarrow \text{argmin}_{\widehat{\mathbf{Y}} \in \mathcal{Y}} \left\langle v^{\mathsf{y}} + \nabla_y r_{\mathcal{A}}^{(\alpha,\mu)}(x^+, \mathbf{Y}^+) - \nabla_y r_{\mathcal{A}}^{(\alpha,\mu)}(x, \mathbf{Y}), \widehat{\mathbf{Y}} \right\rangle + V_{\overline{\mathbf{Y}}}^{\beta H}(\widehat{\mathbf{Y}})$

**7 Return:** $(x^+, \mathbf{Y}^+, \overline{\mathbf{Y}}^+)$

---

**Corollary 5.** *For $\beta \geq \alpha \geq \frac{1}{2}$, $\mu \geq 0$, Algorithm 6 is an $(\alpha, \beta, \mu, 11\Delta)$-approximate extragradient step oracle.*

*Proof.* The proof is the same as Lemma 7, but we incur error due to the approximate computations of $w$, $\bar{x}$, and $x^+$. Let $r := r_{\mathcal{A}}^{(\alpha,\mu)}$ for convenience. Let $\bar{x}_\star$ minimize the subproblem defining $\bar{x}$ and let $x_\star^+$ minimize the subproblem defining $x^+$.

Lemma 7 combines inequalities (17), (18), and first-order optimality of $x^+$. As in Corollary 4, in place of (17) the approximation guarantee on $\bar{x}$ yields for any $u^{\mathsf{y}} \in \mathcal{Y}$,

$$
\begin{aligned}
\left\langle v^{\mathsf{y}} + \nabla_y r(x^+, \mathbf{Y}^+) - \nabla_y r(x, \mathbf{Y}), \mathbf{Y}^+ - u^{\mathsf{y}} \right\rangle &= \left\langle v^{\mathsf{y}} + \nabla_y r(\bar{x}, \overline{\mathbf{Y}}) - \nabla_y r(x, \mathbf{Y}), \mathbf{Y}^+ - \overline{\mathbf{Y}}^+ \right\rangle \\
&+ \left\langle v^{\mathsf{y}} + \nabla_y r(x^+, \mathbf{Y}^+) - \nabla_y r(x, \mathbf{Y}), \overline{\mathbf{Y}}^+ - u^{\mathsf{y}} \right\rangle \\
&+ \left\langle \nabla_y r(x^+, \mathbf{Y}^+) - \nabla_y r(\bar{x}, \overline{\mathbf{Y}}), \mathbf{Y}^+ - \overline{\mathbf{Y}}^+ \right\rangle \\
&\leq V_{\overline{\mathbf{Y}}}^{\beta H}(u^{\mathsf{y}}) - V_{\overline{\mathbf{Y}}^+}^{\beta H}(u^{\mathsf{y}}) - V_{\mathbf{Y}^+}^{\beta H}(\overline{\mathbf{Y}}^+) - V_{\overline{\mathbf{Y}}}^{\beta H}(\mathbf{Y}^+) \\
&+ \left\langle \nabla_y r(x^+, \mathbf{Y}^+) - \nabla_y r(\bar{x}, \overline{\mathbf{Y}}), \mathbf{Y}^+ - \overline{\mathbf{Y}}^+ \right\rangle \\
&\leq V_{\overline{\mathbf{Y}}}^{\beta H}(u^{\mathsf{y}}) - V_{\overline{\mathbf{Y}}^+}^{\beta H}(u^{\mathsf{y}}) - V_{\mathbf{Y}^+}^{\beta H}(\overline{\mathbf{Y}}^+) - V_{\overline{\mathbf{Y}}}^{\beta H}(\mathbf{Y}^+) \\
&+ \left\langle \nabla_y r(x^+, \mathbf{Y}^+) - \nabla_y r(\bar{x}_\star, \overline{\mathbf{Y}}), \mathbf{Y}^+ - \overline{\mathbf{Y}}^+ \right\rangle + 4\Delta.
\end{aligned}
$$

Similarly, in place of (18) we have

$$
\begin{aligned}
\left\langle \nabla_y r(x^+, \mathbf{Y}^+) - \nabla_y r(\bar{x}_\star, \overline{\mathbf{Y}}), \mathbf{Y}^+ - \overline{\mathbf{Y}}^+ \right\rangle &= \left\langle \nabla_y r(x_\star^+, \mathbf{Y}^+) - \nabla_y r(\bar{x}_\star, \overline{\mathbf{Y}}), \mathbf{Y}^+ - \overline{\mathbf{Y}}^+ \right\rangle \\
&+ \left\langle \nabla_y r(x^+, \mathbf{Y}^+) - \nabla_y r(x_\star^+, \mathbf{Y}^+), \mathbf{Y}^+ - \overline{\mathbf{Y}}^+ \right\rangle \\
&\leq V_{\mathbf{Y}^+}^{\beta H}(\overline{\mathbf{Y}}^+) + V_{\overline{\mathbf{Y}}}^{\beta H}(\mathbf{Y}^+) + 4\Delta.
\end{aligned}
$$

Finally, as in the proof of Corollary 4, we lose an additive $3\Delta$ in the optimality of $x^+$. $\qquad\square$

### 4.3 Implementation details

**Approximate best response oracles.** We first develop an implementation for the approximate best response oracles in Section 4.2. Specifically, we study problems of the form

$$
\min_{x \in \mathcal{X}} \langle v, x \rangle + \left\langle |\mathcal{A}|^*(\mathbf{Y}), x^2 \right\rangle + \frac{\mu}{2} \|x\|_2^2 \tag{22}
$$

where $v \in \mathcal{X}^*$, $\mathbf{Y} \in \mathcal{Y}$, $\{\mathbf{A}_i\}_{i \in [n]}$, and $\mu$ are fixed throughout. We further introduce the notation

$$\ell_v(q) := \min_{x \in \mathcal{X}} \langle v, x \rangle + \langle q, x^2 \rangle = \sum_{i \in [n]} \begin{cases} -\frac{v_i^2}{2q_i} & |v_i| \le 2q_i \\ q_i - |v_i| & |v_i| \ge 2q_i \end{cases},$$

minimized by $x = \mathbf{\Pi}_{\mathcal{X}}\left(-\frac{v}{2q}\right)$.

With this notation, the problem in (22) can be written as $\min_{x \in \mathcal{X}} \ell_v(|\mathcal{A}|^*(\mathbf{Y}) + \frac{\mu}{2}\mathbb{1}_n)$. Our best response oracle for $r_{\mathcal{A}}^{(\alpha,\mu)}$ that our algorithms require follows from the following structural fact.

**Lemma 11.** *Let $q \in \mathbb{R}_{\ge 0}^n$, and for a parameter $\epsilon \in (0,1)$, suppose that $\tilde{q} \in \mathbb{R}_{\ge 0}^n$ satisfies*

$$|q_i - \tilde{q}_i| \le \epsilon q_i \text{ for all } i \in [n]. \tag{23}$$

*Then letting $x_\star$ minimize $\ell_v(q)$ over $\mathcal{X}$ and $\tilde{x}_\star$ minimize $\ell_v(\tilde{q})$ over $\mathcal{X}$, for all $u^\times \in \mathcal{X}$,*

$$\|x_\star - \tilde{x}_\star\|_\infty \le \epsilon,$$
$$\langle v + 2q \circ \tilde{x}_\star, \tilde{x}_\star - u^\times \rangle \le 2\epsilon \|q\|_1.$$

*Proof.* The first claim decomposes coordinatewise: in light of our characterization of the minimizers of $\ell_v$, it suffices to show that for any scalars $a, b \ge 0$ and $v \in \mathbb{R}$ such that $|a - b| \le \epsilon a$,

$$\left|\text{med}\left(\frac{v}{a}, -1, 1\right) - \text{med}\left(\frac{v}{b}, -1, 1\right)\right| \le \epsilon.$$

This follows by a case analysis: if $v \in [-a, a]$, then the additive error after the median operation is at most $\epsilon$. Otherwise, if $v \ge a$, then the first corresponding term after taking a median is 1 and the second is in $[1, 1 + \epsilon]$, and a similar argument handles the case $v \le -a$.

For the second claim, note that first-order optimality of $\tilde{x}_\star$ implies

$$\langle v + 2\tilde{q} \circ \tilde{x}_\star, \tilde{x}_\star - u^\times \rangle \le 0,$$

and Hölder's inequality implies the conclusion where we use $\|2\tilde{q} \circ \tilde{x}_\star - 2q \circ \tilde{x}_\star\|_1 \le 2 \|q - \tilde{q}\|_1$. $\square$

Lemma 11 shows that to implement an oracle meeting Definition 8, it suffices to compute a multiplicative approximation to $q = |\mathcal{A}|^*(\mathbf{Y}) + \frac{\mu}{2}\mathbb{1}_n$. We will use an implementation trick which was observed by [AJJ$^+$22] to implicitly maintain $\mathbf{Y}$ exactly via its logarithm. In particular, we use recursive structure to maintain explicit vectors $w, w' \in \mathbb{R}^n$ and a scalar $b \in \mathbb{R}$, such that

$$\mathbf{Y} = \mathbf{\Pi}_{\mathcal{Y}}\left(\exp\left(\mathcal{A}(w) + |\mathcal{A}|(w') + b\mathbf{B}\right)\right). \tag{24}$$

Assuming this maintenance, we give a full implementation of an approximate best response oracle.

**Lemma 12.** *Let $\Delta, \delta \in (0, 1)$, $\mu \le \frac{1}{n}$, $\alpha \ge 0$, and suppose for explicit $w, w' \in \mathbb{R}^n$, $b \in \mathbb{R}$, $\mathbf{Y}$ satisfies (24). We can implement a $\Delta$-approximate best response oracle for $r_{\mathcal{A}}^{(\alpha,\mu)}$ and $\mathbf{Y}$ with probability $\ge \delta$ in one call to a $(\frac{\Delta}{2}, \delta, \frac{\mu\Delta}{2d})$-MEQ oracle for $\{|\mathbf{A}_i|\}_{i \in [n]}$ and $\mathbf{M} = \mathcal{A}(w) + |\mathcal{A}|(w') + b\mathbf{B}$.*

*Proof.* It suffices to apply Lemma 11 with $\tilde{q} = \frac{\mu}{2}\mathbb{1}_n$ plus the approximation to $|\mathcal{A}|^*(\mathbf{Y})$ from the MEQ oracle, which clearly meets the multiplicative approximation required by Lemma 11 with parameter $\epsilon = \frac{\Delta}{2}$, since $\text{Tr}(|\mathbf{A}_i|) \le d \|\mathbf{A}_i\|_{\text{op}} \le d$. For $q = |\mathcal{A}|^*(\mathbf{Y}) + \frac{\mu}{2}\mathbb{1}_n$ with $\mu \le \frac{1}{n}$, $\|q\|_1 \le 2$, and hence the guarantees of Lemma 11 imply that returning the minimizer to $\ell_v(\tilde{q})$ implements a best response oracle with parameter $\Delta$ as long as the MEQ oracle succeeds. $\square$

**Approximate gradient oracles.** We next note that appropriately-parameterized MEQ oracles allow us to straightforwardly implement the approximate gradient oracles for $g$ in (19) and $\nabla_x r_{\mathcal{A}}^{(\alpha,\mu)}(\cdot, \mathbf{Y})$ used in Section 4.2. We state this guarantee in the following.

**Lemma 13.** *Let $\Delta, \delta \in (0,1)$, $\mu \leq \frac{1}{n}$, $\alpha \geq 0$, and suppose for explicit $w, w' \in \mathbb{R}^n$ and $b \in \mathbb{R}$, $\mathbf{Y}$ satisfies (24). We can implement a $\Delta$-approximate gradient oracle for $g^{\times}(\cdot, \mathbf{Y})$ with probability $\geq 1 - \delta$ in one call to a $(\frac{\Delta}{2}, \delta, \frac{\Delta}{2d})$-MEQ oracle for $\{\mathbf{A}_i\}_{i \in [n]}$ and $\mathbf{M} = \mathcal{A}(w) + |\mathcal{A}|(w') + b\mathbf{B}$. We can also implement a $\Delta$-approximate gradient oracle for $\nabla_x r_{\mathcal{A}}^{(\alpha,\mu)}(\cdot, \mathbf{Y})$ in one call to a $(\frac{\Delta}{2}, \delta, \frac{\Delta}{2d})$-MEQ oracle for $\{|\mathbf{A}_i|\}_{i \in [n]}$ and $\mathbf{M} = \mathcal{A}(w) + |\mathcal{A}|(w') + b\mathbf{B}$.*

*Proof.* By Hölder's inequality, it suffices to approximate each relevant operator to an $\ell_1$ error of $\Delta$, which (again recalling $\sum_{i \in [n]} \text{Tr}(|\mathbf{A}_i|) \leq d$) is satisfied by the MEQ oracle whenever it succeeds. $\square$

**Maintaining the invariant** (24). Finally, we discuss how to maintain the implicit representation (24) for the $\mathcal{Y}$ iterates of our algorithm, under the updates of Corollary 4 or Corollary 5.

**Lemma 14.** *Suppose for explicit $w, w' \in \mathbb{R}^n$, $b \in \mathbb{R}$, the input $\mathbf{Y}$ to Corollary 4 satisfies (24), and suppose the input $g^{\mathsf{y}}$ satisfies for explicit $w_g, w'_g \in \mathbb{R}^n$, $b_g \in \mathbb{R}$,*

$$g^{\mathsf{y}} = \mathcal{A}(w_g) + |\mathcal{A}|(w'_g) + b_g \mathbf{B}.$$

*We can maintain $\mathbf{Y}'$ implicitly of the form (24) with $w, w', b$ replaced by $\hat{w}, \hat{w}', \hat{b}$, such that*

$$\max\left(\|\hat{w}\|_{\infty}, \|\hat{w}'\|_{\infty}, |\hat{b}|\right) \leq \max\left(\|w\|_{\infty}, \|w'\|_{\infty}, |b|\right) + \frac{1}{\beta} \max\left(\|w_g\|_{\infty}, \|w'_g\|_{\infty} + 1, |b_g|\right).$$

*Proof.* By Theorem 3.1 of [Lew96], $\nabla H(\mathbf{Y}) = \log \mathbf{Y} + \mathbf{I}_d$. Optimality of $\mathbf{Y}'$ then shows

$$\log \mathbf{Y}' - \log \mathbf{Y} = -\frac{1}{\beta}\left(g^{\mathsf{y}} + |\mathcal{A}|(\hat{x}^2) - |\mathcal{A}|(x^2)\right) + \iota \mathbf{I}_d$$

$$\implies \log \mathbf{Y}' = \mathcal{A}\left(w - \frac{1}{\beta}w_g\right) + |\mathcal{A}|\left(w' - \frac{1}{\beta}\left(w'_g + \hat{x}^2 - x^2\right)\right) + \left(b - \frac{1}{\beta}b_g\right)\mathbf{B} + \iota' \mathbf{I}_d,$$

for some scalars $\iota, \iota'$, by our implicit maintenance of $\mathbf{Y}$ and $g^{\mathsf{y}}$. Noting that the form of (24) is invariant to arbitrary additive shifts by the identity in the argument of exp yields the claim. $\square$

**Lemma 15.** *Suppose for explicit $w, w', \bar{w}, \bar{w}' \in \mathbb{R}^n$, $b, \bar{b} \in \mathbb{R}$, the input $\mathbf{Y}$ to Corollary 5 satisfies (24) and the input $\overline{\mathbf{Y}}$ satisfies (24) with $w, w', b$ replaced with $\bar{w}, \bar{w}', \bar{b}$, and suppose the input $g^{\mathsf{y}}$ satisfies for explicit $w_g, w'_g \in \mathbb{R}^n$, $b_g \in \mathbb{R}$,*

$$g^{\mathsf{y}} = \mathcal{A}(w_g) + |\mathcal{A}|(w'_g) + b_g \mathbf{B}.$$

*We can maintain $\mathbf{Y}^+$ and $\overline{\mathbf{Y}}^+$ implicitly of the form (24) with $w, w', b$ replaced with $w_+, w'_+, b_+$ and $\bar{w}_+, \bar{w}'_+, \bar{b}_+$ respectively, such that*

$$\max\left(\|w_+\|_{\infty}, \|\bar{w}_+\|_{\infty}, \|w'_+\|_{\infty}, \|\bar{w}'_+\|_{\infty}, |b_+|, |\bar{b}_+|\right) \leq \max\left(\|\bar{w}\|_{\infty}, \|\bar{w}'_{\infty}\|, |\bar{b}|\right)$$
$$+ \frac{1}{\beta}\max\left(\|w_g\|_{\infty}, \|w'_g\|_{\infty} + 1, |b_g|\right).$$

*Proof.* The proof is analogous to Lemma 14; dropping multiples of $\mathbf{I}_d$, the optimality conditions on $\mathbf{Y}^+$ and $\overline{\mathbf{Y}}^+$ imply that it suffices to take

$$\left(w_+, w'_+, b_+\right) \leftarrow \left(\bar{w} - \frac{1}{\beta} w_g, \bar{w}' - \frac{1}{\beta}\left(w'_g + \bar{x}^2 - x^2\right), \bar{b} - \frac{1}{\beta} b_g\right),$$

$$\left(\bar{w}_+, \bar{w}'_+, \bar{b}_+\right) \leftarrow \left(\bar{w} - \frac{1}{\beta} w_g, \bar{w}' - \frac{1}{\beta}\left(w'_g + (x^+)^2 - x^2\right), \bar{b} - \frac{1}{\beta} b_g\right).$$

$\square$

## 4.4 Algorithm

We now combine the results of Sections 4.1, 4.2, and 4.3 to give a full analysis for our box-spectraplex solver. We will actually prove a generalization of our claimed result Theorem 2, phrased in terms of MEQ oracles. Theorem 2 then follows by combining Theorem 3 with the oracle implementation in Proposition 2 (or exact oracles), applied with the stated required parameters in the following claim.

**Theorem 3.** *There is an algorithm which computes an $\epsilon$-approximate saddle point to* (2) *in*

$$T = O\left(\frac{L_{\mathcal{A}} \log d}{\epsilon}\right) \quad \text{iterations, where } L_{\mathcal{A}} := \||\mathcal{A}|(\mathbb{1}_n)\|_{\mathrm{op}},$$

*with probability $\geq 1 - \delta$, each using $O(1)$ calls to a $(\Theta(\frac{\epsilon}{L_{\mathcal{A}}}), \frac{\delta}{O(T)}, \Theta(\frac{\epsilon}{L_{\mathcal{A}} nd}))$-MEQ oracle (Definition 6), for $\{\mathbf{A}_i, |\mathbf{A}_i|\}_{i \in [n]}$ and $\mathbf{M} = \mathcal{A}(w) + |\mathcal{A}(w')| + \beta \mathbf{B}$, where $\|w\|_\infty, \|w'\|_\infty, |\beta| = O(T)$.*

*Proof.* As in Theorem 1, we can assume throughout (by rescaling) that $L_{\mathcal{A}} = 1$ for simplicity. We again use the parameters $\alpha = \beta = 2$, $\gamma = 4$, and $\eta = \frac{1}{3}$; we also use $\mu = \frac{1}{n}$. This is a valid choice of $\eta$ for use in Corollary 3, due to Corollary 2. Hence, using Lemma 10 to bound the initial divergence, the proof of Theorem 1 (substituting Corollary 4 and 5 appropriately) implies we need to obtain $\Theta(\epsilon)$-approximate best response oracles and approximate gradient oracles in each iteration.

We next observe that under the given parameter settings, the recursions stated in Lemmas 14 and 15 implies we can maintain every $\mathcal{Y}$ iterate used in calls to Corollary 3 and 2 as $\mathbf{\Pi}_{\mathcal{Y}}(\exp(\mathbf{M}))$, where $\mathbf{M}$ has the form stated in the theorem statement. In particular, using notation of Lemmas 14 and 15, it is clear $w_g \in \mathcal{X}$, $|b_g| = \Theta(1)$, and $w'_g = \mathbb{0}_n$ in every call. Under this maintenance, the conclusion follows by plugging in Lemmas 12 and 13 as our oracle implementations. $\square$

Combining Theorem 3 and Proposition 2 subsumes and slightly refines the first result in Theorem 2. The second result in Theorem 2 comes from an exact implementation of MEQ oracles.

**Corollary 6.** *There is an algorithm which computes an $\epsilon$-approximate saddle point to* (2) *in time*

$$O\left(\left(\mathcal{T}_{\mathrm{mv}}(\mathbf{B}) + \sum_{i \in [n]} \left(\mathcal{T}_{\mathrm{mv}}(\mathbf{A}_i) + \mathcal{T}_{\mathrm{mv}}(|\mathbf{A}_i|)\right)\right) \cdot \frac{L_{\mathcal{A}}^3 \sqrt{L_{\mathrm{tot}}} \log(d) \log^2\left(\frac{Lnd}{\delta \epsilon}\right)}{\epsilon^{3.5}}\right),$$

*with probability $\geq 1 - \delta$, where $L_{\mathrm{tot}} := \|\sum_{i \in [n]} |\mathbf{A}_i|\|_{\mathrm{op}} + \|\mathbf{B}\|_{\mathrm{op}}$ and $L_{\mathcal{A}} := \|\sum_{i \in [n]} |\mathbf{A}_i|\|_{\mathrm{op}}$.*

## 5 Applications

We finally discuss various applications of our new box-simplex solver in Theorem 1.

**Optimal transport.** The discrete optimal transportation problem is a fundamental optimization problem on discrete probability distributions. Given two input distributions $p, q \in \Delta^d$ and a cost matrix $\mathbf{C} \in \mathbb{R}_{\geq 0}^{d \times d}$, the problem is to find a matrix $\mathbf{X}$ solving the following linear program:

$$\min_{\mathbf{X}\mathbb{1}_d = p, \mathbf{X}^\top \mathbb{1}_d = q, \mathbf{X} \geq 0} \langle \mathbf{C}, \mathbf{X} \rangle.$$

In previous work, [JST19] observes that a $2\epsilon$-approximate optimal transport map can be recovered from an $\epsilon$-approximate saddle point of the following box-simplex game:

$$\min_{\mathbf{X} \in \Delta^{n^2}} \max_{y \in [-1,1]^{2n}} \langle \mathbf{C}, \mathbf{X} \rangle + 2\left\|\mathbf{C}\right\|_{\max} y^\top (\mathbf{B}\mathbf{X} - r). \tag{25}$$

We treat $\mathbf{X}$ as an element of $\Delta^{n^2}$ and define $\mathbf{B} : \mathbb{R}^{n^2} \to \mathbb{R}^{2n}$ is the linear operator which sends $\mathbf{X}$ to $[\mathbf{X}\mathbb{1}, \mathbf{X}^\top \mathbb{1}]$ and $r = [p, q]$. By applying Theorem 1 to (25), we obtain the following.

**Corollary 7.** *Let $\mathbf{C} \in \mathbb{R}_{\geq 0}^{n \times n}$, $p, q \in \Delta^n$ be given. There is an algorithm to compute an $\epsilon$-approximate optimal transport map from $p$ to $q$ with costs $\mathbf{C}$ running in time $O(n^2 \log n \left\|\mathbf{C}\right\|_{\max} \epsilon^{-1})$.*

**Min-mean cycle.** Additionally, our improved box-simplex game solver gives an improved algorithm for the minimum mean-cycle problem. In this problem, we are given an undirected weighted graph $G$ and we seek a cycle $C$ of length $\ell$ of minimum mean length $\frac{1}{\ell} \sum_{e \in C} w_e$. As noted in [AP20], the min-mean cycle problem is equivalent to the following primal-dual optimization problem:

$$\min_{x \in \Delta^m} \max_{y \in [-1,1]^n} w^\top x + 3dw_{\max} y^\top \mathbf{B} x \tag{26}$$

where $\mathbf{B}$ is the oriented graph incidence matrix of $G$, $w_{\max}$ is the maximum edge weight in $G$, and $d$ is the (unweighted) graph diameter of $G$. This problem is a box-simplex game, and $\left\|dw_{\max}\mathbf{B}\right\|_{1 \to 1} = O(dw_{\max})$. Further, by [AP20] $\epsilon$-approximate saddle points for this problem give $\epsilon$-approximate min-mean cycles: applying Theorem 1 to (26) then gives the following corollary.

**Corollary 8.** *Let $G$ be a (nonnegative weighted) graph with $n$ vertices, $m$ edges; let $w \in \mathbb{R}_{\geq 0}^m$ be the vector of edge weights. Let $w_{\max}$ be the maximum edge weight in $G$ and $d$ be the unweighted diameter of $G$. There is an algorithm to compute an $\epsilon$-approximate minimum-mean cycle in time*

$$O\left(\frac{mdw_{\max} \log n}{\epsilon}\right).$$

**Faster flow problems on graphs.** Finally, our framework implies faster approximation algorithms for the important combinatorial optimization problems of transshipment and maximum flow. Given a graph $G$ with edge weights $w$ and a demand $d$, these problems can written in the form

$$\min_{\mathbf{B}f = d} \left\|\mathbf{W}f\right\|_1 \quad \text{and} \quad \min_{\mathbf{B}f = d} \left\|\mathbf{W}f\right\|_\infty$$

respectively, where $\mathbf{B}$ is the graph incidence matrix. As used in previous work, these problems admit 'cost approximators': linear operators which approximate the optimal cost of the corresponding flow problem up to a polylogarithmic factor. The guarantee of these approximators is summarized below.

**Lemma 16** (Lemma 8 from [AJJ+22] and Theorem 4.4 from [She17]). *Let $G$ be a graph with $n$ vertices, $m$ edges, and nonnegative edge weights $w$. Let $\mathsf{opt}_p(d)$ denote the optimal value of the $\ell_p$-flow problem over $G$ with demands $d$:*

$$\mathsf{opt}_p(d) = \min_{\mathbf{B}f = d} \left\|\mathbf{W}f\right\|_p.$$

*For $p \in \{1, \infty\}$, there exists an algorithm which computes a matrix $\mathbf{R} \in \mathbb{R}^{K \times n}$ such that for parameters $\alpha, \beta, \gamma, K$,*

- *For any $d \in \mathbb{R}^n$ with $\mathbb{1}^\top d = 0$, $\mathsf{opt}_p(d) \le \|\mathbf{R}d\|_p \le \alpha\mathsf{opt}_p(d)$.*

- *The matrix $\mathbf{RB}$ has $O(m\beta)$ nonzero entries.*

- *The algorithm runs in $O(m\gamma)$ time and returns both $\mathbf{R}$ and $\mathbf{RB}$.*

*In addition, the parameters $\alpha, \beta, \gamma, K$ above can take the values $\alpha = \log^{O(1)} n$, $\beta = \log^{O(1)} n$, $\gamma = \log^{O(1)} n$, and $K = n \log^{O(1)} n$.*

As shown in [AJJ$^+$22], we may compute $(1+\epsilon)$-multiplicative approximations the transshipment problem by solving problems of the form

$$\min_{f \in \Delta^{2m}} \max_{y \in [-1,1]^K} ty^\top \mathbf{A}^\top f - b^\top y \tag{27}$$

to additive error $\epsilon t$, where $t \ge 0$ is a parameter, $b = \mathbf{R}d$, and (where $\mathbf{W}$ is a diagonal weight matrix)

$$\mathbf{A} = \begin{pmatrix} \mathbf{W}^{-1}\mathbf{B}^\top\mathbf{R}^\top \\ -\mathbf{W}^{-1}\mathbf{B}^\top\mathbf{R}^\top \end{pmatrix}.$$

Applying Lemma 16 with $p = 1$ to compute $\mathbf{R}$, we see that $t\|\mathbf{A}^\top\|_{1\to1} \le t\alpha$: employing Theorem 1 gives an algorithm for this task which uses $O(\frac{\alpha \log n}{\epsilon})$ matrix-vector products with $\mathbf{A}, \mathbf{A}^\top, |\mathbf{A}|$, and $|\mathbf{A}|^\top$. Similarly, [She17] obtains $(1 + \epsilon)$-approximate solutions to maximum flow by solving

$$\min_{f \in [-1,1]^m} \max_{y \in \Delta^{2K}} ty^\top \mathbf{A} f - b^\top y$$

to additive error $\epsilon t$, where again $b = \mathbf{R}d$ and

$$\mathbf{A} = \begin{pmatrix} \mathbf{RBW}^{-1} \\ -\mathbf{RBW}^{-1} \end{pmatrix}.$$

Applying Lemma 16 with $p = \infty$ to compute $\mathbf{R}$, we see that $t\|\mathbf{A}^\top\|_{1\to1} \le t\alpha$: employing Theorem 1 gives an algorithm for this task which uses $O(\frac{\alpha \log n}{\epsilon})$ matrix-vector products with $\mathbf{A}, \mathbf{A}^\top, |\mathbf{A}|$, and $|\mathbf{A}|^\top$. Combining these subroutines with the outer-loops described in [AJJ$^+$22, She17] leads to improved algorithms for these graph optimization problems.

Finally, though it is outside the scope of this paper, our Algorithm 4 is efficiently parallelizable and using the techniques of [AJJ$^+$22], it improves state-of-the-art (in some parameter regimes) semi-streaming pass complexities for maximum cardinality matching by a logarithmic factor.

## Acknowledgements

We would like thank our long-term collaborators Yujia Jin and Aaron Sidford for many helpful discussions at earlier stages of this project, which improved our understanding of area convexity. We also thank Victor Reis for his collaboration on related ideas to this work in [JRT23]. Finally, we thank anonymous reviewers for their helpful suggestions in improving our presentation.

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

# A  Box-simplex proximal subproblems via relative conditioning

In this section, we demonstrate the implications of Lemma 5 for solving the subproblems in extra-gradient methods for (1) using area convex regularizers. We reproduce Lemma 5 for convenience.

**Lemma 5.** *In the setting of Lemma 4, suppose for any $x \in \mathcal{X}$, $F(x, \cdot)$ (as a function over $\mathcal{Y}$) always is $r : \mathcal{Y} \to \mathbb{R}$ plus a linear function (where the linear function may depend on $x$). Then $r - f$ is convex, and $f - q$ is convex for any $q : \mathcal{Y} \to \mathbb{R}$ such that $F - q : \mathcal{X} \times \mathcal{Y} \to \mathbb{R}$ is jointly convex.*

Consider a subproblem encountered when running the extragradient method of Appendix B on the problem (1), using the regularizer $r_{\mathbf{A}}^{(\alpha)}$ in (5). It has the form, for some $(g^{\mathsf{x}}, g^{\mathsf{y}}) \in \mathcal{X}^* \times \mathcal{Y}^*$,

$$\min_{x \in [-1,1]^n, y \in \Delta^d} F(x, y) := \langle g^{\mathsf{x}}, x \rangle + \langle g^{\mathsf{y}}, y \rangle + \langle |\mathbf{A}|y, x^2 \rangle + \alpha h(y).$$

Recall from Lemma 1 that $F$ is jointly convex over $(x, y)$ for any $\alpha \geq \frac{1}{2}$. Hence, for $\alpha = 2$, we may apply Lemma 5 with $r(y) := 2h(y)$ to conclude that

$$f(y) := \min_{x \in [-1,1]^n} F(x, y)$$

is 2-relatively smooth with respect to $h$, as a function over $\mathcal{Y} = \Delta^d$. Moreover, applying Lemma 5 with $q(y) := h(y)$, and again using the joint convexity fact in Lemma 1, shows that $f$ is further 1-relatively strongly convex with respect to $h$. At this point, a direct application of Theorem 3.1 in [LFN18] (which gives an algorithm for optimization under relative smoothness and strong convexity) yields a linearly-convergent algorithm for minimizing $F$. We remark that in light of Lemma 4, we can implement gradient queries to $f$ by computing the best response argument for a given $y$.

Interestingly, the analysis in this section used nothing more than Lemma 5, joint convexity of our regularizer, and the ability to tune the parameter $\alpha$ to induce a small amount of relative strong convexity. This is in contrast to the analysis in [She17] (see Lemma 5 of [JST19] for a formal proof), which requires ad hoc multiplicative stability properties. An important consequence of this observation is that the same technique generalizes to the matrix setting via new joint convexity facts we prove in Proposition 1, where multiplicative stability breaks due to non-monotonicity of the matrix exponential. This gives a simple proof-of-concept solver for matching Theorem 3 up to logarithmic factors, which we improve via our approximation-tolerant extragradient methods.

# B  Unified extragradient convergence analysis

In this section, we show that our notion of relaxed relative Lipschitzness (Definition 1) extends area convexity and relative Lipschitzness, and demonstrate that it enables convergence of an extragradient method for variational inequalities. We begin by comparing these conditions.

**Lemma 17.** *Assume either of the following holds for operator $g : \mathcal{Z} \to \mathcal{Z}^*$ and convex $r : \mathcal{Z} \to \mathbb{R}$.*

- *$g$ is $\frac{1}{\eta}$-relatively Lipschitz with respect to $r$.*

- *$g$ is $\eta$-area convex with respect to $r$.*

*Then $g$ is $\frac{1}{\eta}$-relaxed relatively Lipschitz (Definition 1) with respect to $r$.*

*Proof.* The first case follows immediately from Definition 2: for any $(z, z', z^+) \in \mathcal{Z} \times \mathcal{Z} \times \mathcal{Z}$,

$$\eta \left\langle g(z') - g(z), z' - z^+ \right\rangle \leq V_z^r(z') + V_{z'}^r(z^+) \leq V_z^r(z') + V_{z'}^r(z^+) + V_z^r(z^+).$$

For the second case, direct computation and nonnegativity of the Bregman divergence yields

$$r(z) + r(z') + r(z^+) - 3r(c) = V_z^r(z^+) + V_z^r(z') - 3V_z^r(c) \leq V_z^r(z') + V_z^r(z^+) + V_{z'}^r(z^+).$$

□

With this fact in hand, we now show that the more general condition of relaxed relative Lipschitzness is sufficient to prove convergence of a simple extragradient method. In particular, we analyze Algorithm 7, a slight variant of the mirror prox algorithm [Nem04, CST21], and show that it gives rates for relaxed relative Lipschitz monotone variational inequalities that match the rates for relative Lipschitz or area convex operator-regularizer pairs up to constant factors. We will require the definition of a proximal oracle, which takes the standard iteration for mirror descent algorithms.

**Definition 11** (Proximal oracle). *For convex $r : \mathcal{Z} \to \mathbb{R}$, $z \in \mathcal{Z}$, and $g \in \mathcal{Z}^*$, $\mathrm{Prox}_z^r(g)$ outputs*

$$z' = \arg\min_{w \in \mathcal{Z}} \left\langle g, w \right\rangle + V_z^r(w).$$

---

**Algorithm 7:** RelaxedMirrorProx$(g, r, z_0, \eta, T)$

---

1 **Input:** Operator $g : \mathcal{Z} \to \mathcal{Z}^*$ satisfying $\frac{1}{\eta}$ relaxed relative Lipschitzness with respect to $r$, $T \in \mathbb{N}$, $z_0 \in \mathcal{Z}$
2 **for** $t = 0$ **to** $T - 1$ **do**
3 $\quad w_t = \mathrm{Prox}_{z_t}^r(\eta g(z_t))$
4 $\quad z_{t+1} = \mathrm{Prox}_{z_t}^r\left(\frac{\eta}{2} g(w_t)\right)$
5 **end**

---

We note that Algorithm 7 is exactly the same as Algorithm 1 of [CST21] (a rephrasing of the main result of [Nem04]), except there is a factor of 2 in the step size in Line 4. As the proof of Proposition 3 shows, this allows us to obtain an extra divergence term which is handled by the relaxed relative Lipschitzness condition. Notably, this extra divergence is also handled by operator-regularizer pairs satisfying area convexity, explaining the same step size change appearing in [She17].

**Proposition 3.** *Let $g : \mathcal{Z} \to \mathcal{Z}^*$, $r : \mathcal{Z} \to \mathbb{R}$, $z_0 \in \mathcal{Z}$, and assume that $g$ satisfies $\frac{1}{\eta}$-relaxed relative Lipschitzness with respect to $r$ for some $\eta > 0$. The iterates of Algorithm 7 satisfy (for any $u \in \mathcal{Z}$),*

$$\frac{1}{T} \sum_{t=0}^{T-1} \left\langle g(w_t), w_t - u \right\rangle \leq \frac{2}{\eta T} V_{z_0}^r(u).$$

*Proof.* Applying (4) to the optimality conditions implied by the steps of Algorithm 7, we obtain

$$\eta \left\langle g(z_t), w_t - z_{t+1} \right\rangle \leq V_{z_t}^r(z_{t+1}) - V_{w_t}^r(z_{t+1}) - V_{z_t}^r(w_t)$$

and

$$\frac{\eta}{2} \left\langle g(w_t), z_{t+1} - u \right\rangle \leq V_{z_t}^r(u) - V_{z_{t+1}}^r(u) - V_{z_t}^r(z_{t+1}).$$

Doubling the second inequality and adding it to the first, we have

$$\eta\left(\langle g(w_t), z_{t+1} - u\rangle + \langle g(z_t), w_t - z_{t+1}\rangle\right) \le 2V_{z_t}^r(u) - 2V_{z_{t+1}}^r(u) - V_{z_t}^r(z_{t+1}) - V_{w_t}^r(z_{t+1}) - V_{z_t}^r(w_t).$$

Rearranging the above yields

$$\begin{aligned}
\eta\langle g(w_t), w_t - u\rangle &\le 2V_{z_t}^r(u) - 2V_{z_{t+1}}^r(u) \\
&\quad - V_{z_t}^r(z_{t+1}) - V_{w_t}^r(z_{t+1}) - V_{z_t}^r(w_t) + \eta\langle g(w_t) - g(z_t), w_t - z_{t+1}\rangle \\
&\le 2V_{z_t}^r(u) - 2V_{z_{t+1}}^r(u)
\end{aligned}$$

where the inequality used relaxed relative Lipschitzness of $g$. Summing over all $T$ iterations and dividing by $\eta T$ then gives the desired

$$\frac{1}{T}\sum_{t=0}^{T-1}\langle g(w_t), w_t - u\rangle \le \frac{2}{\eta T}\left(V_{z_0}^r(u) - V_{z_T}^r(u)\right) \le \frac{2}{\eta T}V_{z_0}^r(u).$$

$\square$

We briefly compare Proposition 3 to the analyses of extragradient methods considered in [She17, CST21]. The extragradient method considered in [She17] can be viewed as a dual variant of Algorithm 7, and is based on dual extrapolation [Nes07] instead of mirror prox (see discussion in [CST21] for their relationship); the latter is the skeleton of our Algorithm 7. This affirmatively answers the question of whether there is a mirror prox-like algorithm which converges under area convexity, which to our knowledge was previously not known. The algorithm in [She17] obtains the same regret guarantee as in Proposition 3, also calling $O(1)$ proximal oracles in $r$ per iteration.

On the other hand, the mirror prox algorithm of [Nem04, CST21] run for $T$ iterations run on a $\frac{1}{\eta}$-relatively Lipschitz operator-regularizer pair yields

$$\sum_{t=0}^{T-1}\langle g(w_t), w_t - u\rangle \le \frac{1}{\eta T}V_{z_0}^r(u)$$

for any $u \in \mathcal{Z}$, where again each iteration requires $O(1)$ calls to a proximal oracle Definition 11. This result therefore improves Proposition 3's convergence rate by a factor of 2, at the cost of using the stronger Definition 2. Finally, we note that exact implementations of the proximal oracles required by Algorithm 7 satisfy the oracles used in Sections 3 and 4, i.e. those in Definitions 4 and 5, with $\beta = 0$, as seen by applying (4). To handle error introduced by not being able to exactly implement a proximal oracle for our regularizers (5) and (7), we relax our method to handle $\beta > 0$, and we satisfy the required relaxations via relative conditioning properties of the proximal subproblems.

## C  Proof of Lemma 8

In this section, we provide a proof of Lemma 8. We begin by recalling a technical claim on matrix relative entropy. We require the notion of *positive* maps on matrices.

**Definition 12.** *Let* $\Phi : \mathbb{R}^{d\times d} \to \mathbb{R}^{n\times n}$ *be a linear function on matrices. We say* $\Phi$ *is* positive *if* $\Phi(\mathbf{A}) \succeq 0$ *for any* $\mathbf{A} \in \mathbb{S}_{\succeq \mathbf{0}}^d$. *In addition, for any* $k \ge 1$ *define the (linear) map* $\Phi_k : \mathbb{R}^{kd\times kd} \to \mathbb{R}^{kn\times kn}$ *which sends*

$$\mathbf{A} = \begin{pmatrix} \mathbf{A}_{1,1} & \mathbf{A}_{1,2} & \dots & \mathbf{A}_{1,k} \\ \mathbf{A}_{2,1} & \mathbf{A}_{2,2} & \dots & \mathbf{A}_{2,k} \\ \vdots & \vdots & \ddots & \vdots \\ \mathbf{A}_{k,1} & \mathbf{A}_{k,2} & \dots & \mathbf{A}_{k,k} \end{pmatrix}$$

*to*

$$\Phi_k(\mathbf{A}) = \begin{pmatrix} \Phi(\mathbf{A}_{1,1}) & \Phi(\mathbf{A}_{1,2}) & \dots & \Phi(\mathbf{A}_{1,k}) \\ \Phi(\mathbf{A}_{2,1}) & \Phi(\mathbf{A}_{2,2}) & \dots & \Phi(\mathbf{A}_{2,k}) \\ \vdots & \vdots & \ddots & \vdots \\ \Phi(\mathbf{A}_{k,1}) & \Phi(\mathbf{A}_{k,2}) & \dots & \Phi(\mathbf{A}_{k,k}) \end{pmatrix}.$$

*We say $\Phi$ is $k$-positive if $\Phi_k$ is positive. We say $\Phi$ is* completely positive *if it is $k$-positive for any $k \geq 1$.*

We additionally require the following equivalent characterization of completely positive maps.

**Theorem 4** (Choi-Kraus Representation Theorem, [Cho75])**.** *Let $\Phi : \mathbb{R}^{d \times d} \to \mathbb{R}^{n \times n}$ be a linear map on matrices. Then it is completely positive if and only if there exist $\mathbf{V}_1, \mathbf{V}_2 \dots \mathbf{V}_k \in \mathbb{C}^{n \times d}$ such that*

$$\Phi(\mathbf{A}) = \sum_{i=1}^{k} \mathbf{V}_i \mathbf{A} \mathbf{V}_i^{\dagger}$$

*where $\mathbf{V}_i^{\dagger}$ denotes the Hermitian transpose.*

We refer the reader to Chapter 3 of [Bha97] for more discusion of this and related results. With this notion, we recall a useful fact concerning matrix relative entropy and completely positive maps.

**Lemma 18** (Theorem, [Lin75])**.** *Let $\Phi : \mathbb{R}^{d \times d} \to \mathbb{R}^{n \times n}$ be a completely positive map on matrices such that $\mathrm{Tr}(\mathbf{A}) = \mathrm{Tr}(\Phi(\mathbf{A}))$ for any $\mathbf{A} \in \mathbb{R}^{d \times d}$. Then for any $\mathbf{A}, \mathbf{B} \in \mathbb{S}_{\succ \mathbf{0}}^d$,*

$$V_{\mathbf{B}}^H(\mathbf{A}) \geq V_{\Phi(\mathbf{B})}^H(\Phi(\mathbf{A})).$$

We refer the reader to [Yu13, Bha97] for further discussion of this result. With these facts in hand, we are ready to prove Lemma 8: we recall it for convenience below.

**Lemma 8.** *Let $\mathcal{A} := \{\mathbf{A}_i\}_{i \in [n]} \subset \mathbb{S}_{\succeq \mathbf{0}}^d$ satisfy $\sum_{i \in [n]} \mathbf{A}_i = \mathbf{I}_d$. For any $\mathbf{M} \in \mathbb{S}^d$ and $\mathbf{Y} \in \mathbb{S}_{\succ \mathbf{0}}^d$ we have $\nabla^2 H(\mathbf{Y})[\mathbf{M}, \mathbf{M}] \geq \nabla^2 h(y)[m, m]$ for $y := \mathcal{A}^*(\mathbf{Y})$, $m := \mathcal{A}^*(\mathbf{M})$.*

*Proof.* Define the map $\Phi : \mathbb{R}^{d \times d} \to \mathbb{R}^{n \times n}$ by $\Phi(\mathbf{Y}) := \mathbf{diag}(\mathcal{A}(\mathbf{Y}))$. We begin by showing that $\Phi$ is trace-preserving: for any $\mathbf{Y} \in \mathbb{R}^{d \times d}$,

$$\mathrm{Tr}(\Phi(\mathbf{Y})) = \mathrm{Tr}(\mathbf{diag}(\mathcal{A}(\mathbf{Y}))) = \sum_{i=1}^{n} \langle \mathbf{A}_i, \mathbf{Y} \rangle = \langle \mathbf{I}_d, \mathbf{Y} \rangle = \mathrm{Tr}(\mathbf{Y}).$$

We now show that $\Phi$ is completely positive. As each $\mathbf{A}_i$ is positive semidefinite, there exist vectors $v_{i,1}, v_{i,2}, \dots v_{i,d}$ such that $\mathbf{A}_i = \sum_{j=1}^{d} v_{i,j} v_{i,j}^{\top}$. Given these vectors for every $\mathbf{A}_i$, we define the matrices $\mathbf{V}_{i,j} \in \mathbb{R}^{n \times d}$ as $\mathbf{V}_{i,j} = e_i v_{i,j}^{\top}$. We claim that

$$\Phi(\mathbf{Y}) = \sum_{i=1}^{n} \sum_{j=1}^{d} \mathbf{V}_{i,j} \mathbf{Y} \mathbf{V}_{i,j}^{\top}.$$

To see this, note that

$$\sum_{j=1}^{d} \mathbf{V}_{i,j} \mathbf{Y} \mathbf{V}_{i,j}^{\top} = \sum_{j=1}^{d} e_i (v_{i,j}^{\top} \mathbf{Y} v_{i,j}) e_i^{\top} = \left( \left\langle \sum_{j=1}^{d} v_{i,j} v_{i,j}^{\top}, \mathbf{Y} \right\rangle \right) e_i e_i^{\top} = \langle \mathbf{A}_i, \mathbf{Y} \rangle e_i e_i^{\top}$$

and therefore

$$\sum_{i=1}^{n}\sum_{j=1}^{d}\mathbf{V}_{i,j}\mathbf{Y}\mathbf{V}_{i,j}^{\top} = \sum_{i=1}^{n}\langle\mathbf{A}_i,\mathbf{Y}\rangle\, e_i e_i^{\top} = \mathbf{diag}\left(\mathcal{A}(\mathbf{Y})\right) = \Phi(\mathbf{Y}).$$

By Theorem 4, the existence of these $\mathbf{V}_{i,j}$ implies that $\Phi$ is completely positive. Now $\Phi$ satisfies the necessary conditions for Lemma 18: we therefore have for any $\mathbf{A},\mathbf{B}\in\mathbb{S}_{\succ\mathbf{0}}^{d}$

$$V_{\mathbf{B}}^{H}(\mathbf{A}) \geq V_{\Phi(\mathbf{B})}^{H}(\Phi(\mathbf{A}))$$

Now, for any $\mathbf{Y}\in\mathbb{S}_{\succ\mathbf{0}}^{d}$ and $\mathbf{M}\in\mathbb{S}^{d}$, there exists a constant $\alpha$ such that $\mathbf{Y}+t\mathbf{M}\in\mathbb{S}_{\succ\mathbf{0}}^{d}$ for all $|t|\leq\alpha$. Therefore, for any such $t$ we have

$$V_{\mathbf{Y}}^{H}(\mathbf{Y}+t\mathbf{M}) \geq V_{\Phi(\mathbf{Y})}^{H}(\Phi(\mathbf{Y}+t\mathbf{M})).$$

Taking the limit of $t\to 0$, a second-order Taylor expansion yields

$$\nabla^2 H(\mathbf{Y})[\mathbf{M},\mathbf{M}] = \frac{2}{t^2}\lim_{t\to 0}V_{\mathbf{Y}}^{H}(\mathbf{Y}+t\mathbf{M}),$$

$$\nabla^2 h(y)[m,m] = \frac{2}{t^2}\lim_{t\to 0}V_{y}^{h}(y+tm)$$

$$= \frac{2}{t^2}\lim_{t\to 0}V_{\mathbf{diag}(y)}^{H}(\mathbf{diag}\,(y+tm)) = \frac{2}{t^2}\lim_{t\to 0}V_{\Phi(\mathbf{Y})}^{H}(\Phi(\mathbf{Y}+t\mathbf{M})).$$

The second-to-last equality uses Theorem 3.3 of [LS01], a formula for the gradient of a spectral function (a function from matrices to scalars depending only on the eigenvalues), and the fact that $\Phi(\mathbf{Y})$ and $\Phi(\mathbf{Y}+t\mathbf{M})$ commute. Combining the above displays shows the desired claim. $\qquad\square$

# D   Matrix exponential products

In this section we provide a collection of tools used for approximate computations against a matrix exponential. We begin by providing an approximation to its trace, using some helper tools.

**Proposition 4** (Approximate top eigenvalue, Theorem 1, [MM15]). *Let $\mathbf{M}\in\mathbb{S}_{\succeq\mathbf{0}}^{d}$, and let $\delta,\epsilon\in(0,1)$. There is an algorithm which with probability $\geq 1-\delta$ returns a value $\widehat{\widetilde{\lambda}}$ such that $|\widehat{\lambda}-\lambda_{\max}(\mathbf{M})|\leq\epsilon\lambda_{\max}(\mathbf{M})$, where $\lambda_{\max}$ is the largest eigenvalue of a matrix in $\mathbb{S}^{d}$, in time*

$$O\left(\mathcal{T}_{\mathrm{mv}}(\mathbf{M})\cdot\frac{\log\frac{d}{\delta\epsilon}}{\sqrt{\epsilon}}\right).$$

**Proposition 5** (Polynomial approximation to exp, Theorem 4.1, [SV14]). *Let $\mathbf{M}\in\mathbb{S}_{\succeq\mathbf{0}}^{d}$ have $\mathbf{M}\preceq R\mathbf{I}$, and let $\epsilon\in(0,1)$. There is a polynomial $p$ satisfying*

$$\exp(-\mathbf{M})-\epsilon\mathbf{I}\preceq p(\mathbf{M})\preceq\exp(-\mathbf{M})+\epsilon\mathbf{I},\ \mathrm{degree}(p) = O\left(\sqrt{R\log\frac{1}{\epsilon}}+\log\frac{1}{\epsilon}\right).$$

**Proposition 6** (Johnson-Lindenstrauss transform, Theorem 2.1, [DG03]). *Let $\mathbf{Q}\in\mathbb{R}^{k\times d}$ have rows formed by independently random unit vectors in $\mathbb{R}^{d}$ scaled by $k^{-\frac{1}{2}}$, and let $\epsilon,\delta\in(0,1)$. If $k=\Omega(\frac{1}{\epsilon^2}\log\frac{d}{\delta})$ for an appropriate constant, for any $v\in\mathbb{R}^{d}$ (independent of $\mathbf{Q}$), with probability $\geq 1-\delta$,*

$$(1-\epsilon)\|v\|_2^2 \leq \|\mathbf{Q}v\|_2^2 \leq (1+\epsilon)\|v\|_2^2.$$

**Approximating the exponential trace.** We first show how to use these tools to approximate the trace of the exponential of a bounded matrix efficiently.

**Lemma 19.** *Let $\epsilon, \delta \in (0, 1)$, let $R \geq \log \frac{1}{\epsilon}$, and let $\mathbf{M} \in \mathbb{S}^d$ satisfy $\|\mathbf{M}\|_{\mathrm{op}} \leq R$. We can compute an $\epsilon$-multiplicative approximation to $\operatorname{Tr} \exp \mathbf{M}$ with probability $\geq 1 - \delta$ in time*

$$O\left(\mathcal{T}_{\mathrm{mv}}(\mathbf{M}) \cdot \frac{\sqrt{R} \log^{1.5}(\frac{Rd}{\delta \epsilon})}{\epsilon^2}\right).$$

*Proof.* Assume for simplicity $\epsilon \leq \frac{1}{2}$ as this affects the result by at most a constant. By applying Proposition 4 to the matrix $\mathbf{M} + 2R\mathbf{I}$ with $\epsilon \leftarrow \frac{1}{3R}$, we obtain a value $\widehat{\lambda}$ such that $\lambda_{\max}(\mathbf{M}) \leq \widehat{\lambda} \leq \lambda_{\max}(\mathbf{M}) + 1$ with probability $\geq 1 - \frac{\delta}{2}$, within the allotted time budget. Next, note that for

$$\widetilde{\mathbf{M}} := \widehat{\lambda} \mathbf{I} - \mathbf{M},$$

we have $\operatorname{Tr} \exp(\mathbf{M}) = \exp(\widehat{\lambda}) \cdot \operatorname{Tr} \exp(-\widetilde{\mathbf{M}})$, so it suffices to approximate $\operatorname{Tr} \exp(-\widetilde{\mathbf{M}})$. We observe $\widetilde{\mathbf{M}} \in \mathbb{S}^d_{\succeq 0}$ and it has an eigenvalue at most 1, by definition of $\widehat{\lambda}$. This latter fact implies $\operatorname{Tr} \exp(-\widetilde{\mathbf{M}}) \geq \frac{1}{e}$. Next take $\mathbf{Q}$ from Proposition 6 with $k = O(\frac{1}{\epsilon^2} \log \frac{d}{\delta})$ such that with probability $\geq 1 - \frac{\delta}{2}$, for all $j \in [d]$ (by a union bound), we have

$$\left(1 - \frac{\epsilon}{4}\right) \left\|\left[\exp\left(-\frac{1}{2}\widetilde{\mathbf{M}}\right)\right]_{j:}\right\|_2^2 \leq \left\|\mathbf{Q}\left[\exp\left(-\frac{1}{2}\widetilde{\mathbf{M}}\right)\right]_{j:}\right\|_2^2 \leq \left(1 + \frac{\epsilon}{4}\right) \left\|\left[\exp\left(-\frac{1}{2}\widetilde{\mathbf{M}}\right)\right]_{j:}\right\|_2^2.$$

Union bounding over the success of the two randomized steps gives the failure probability. Since

$$\operatorname{Tr} \exp(-\widetilde{\mathbf{M}}) = \sum_{j \in [d]} \left\|\left[\exp\left(-\frac{1}{2}\widetilde{\mathbf{M}}\right)\right]_{j:}\right\|_2^2,$$

$$\sum_{j \in [d]} \left\|\mathbf{Q}\left[\exp\left(-\frac{1}{2}\widetilde{\mathbf{M}}\right)\right]_{j:}\right\|_2^2 = \operatorname{Tr}\left(\exp\left(-\frac{1}{2}\widetilde{\mathbf{M}}\right) \mathbf{Q}^\top \mathbf{Q} \exp\left(-\frac{1}{2}\widetilde{\mathbf{M}}\right)\right)$$

$$= \operatorname{Tr}\left(\mathbf{Q} \exp(-\widetilde{\mathbf{M}}) \mathbf{Q}^\top\right) = \sum_{\ell \in [k]} \left\|\exp\left(-\frac{1}{2}\widetilde{\mathbf{M}}\right) \mathbf{Q}_{\ell:}\right\|_2^2,$$

combining the above two displays means it suffices to give an $\frac{\epsilon}{4ek}$-additive approximation to the squared norm of each $\exp(-\frac{1}{2}\widetilde{\mathbf{M}})\mathbf{Q}_{\ell:}$. This would result in an overall $\frac{\epsilon}{4}$-multiplicative loss in approximating $\operatorname{Tr} \exp(-\widetilde{\mathbf{M}})$, due to application of $\mathbf{Q}$, and an additional $\frac{\epsilon}{4e}$-additive approximation, which is also a $\frac{\epsilon}{4}$-multiplicative approximation factor due to $\operatorname{Tr} \exp(-\widetilde{\mathbf{M}}) \geq \frac{1}{e}$.

Finally, recall that all $\|\mathbf{Q}_{\ell:}\|_2^2 = \frac{1}{k}$, so applying Proposition 5 with error parameter $\frac{\epsilon}{4e}$, we have a polynomial $p$ of degree $\sqrt{R \log \frac{1}{\epsilon}}$ such that $p(\widetilde{\mathbf{M}})$ approximates $\exp(-\widetilde{\mathbf{M}})$ to an additive $\frac{\epsilon}{4e}\mathbf{I}$. Hence, the quadratic forms of $\mathbf{Q}_{\ell:}$ through $p(\widetilde{\mathbf{M}})$ approximate each

$$\left\|\exp\left(-\frac{1}{2}\widetilde{\mathbf{M}}\right) \mathbf{Q}_{\ell:}\right\|_2^2 = \mathbf{Q}_{\ell:}^\top \exp\left(-\widetilde{\mathbf{M}}\right) \mathbf{Q}_{\ell:}$$

to an additive $\frac{\epsilon}{4ek}$ as desired. The runtime of this step comes from computing all $k = O(\frac{1}{\epsilon^2} \log \frac{d}{\delta})$ quadratic forms with a degree $O(\sqrt{R \log \frac{1}{\epsilon}})$ polynomial in $\mathbf{M}$. $\qquad \square$

**Approximating an inner product.** Next, we build upon the proof of Lemma 19 to approximate an inner product against the exponential of a bounded matrix.

**Lemma 20.** *Let $\epsilon, \gamma, \delta \in (0,1)$, $\{\mathbf{A}_i\}_{i\in[n]} \subset \mathbb{S}^d$, $R \geq \log\frac{1}{\gamma}$, and let $\mathbf{M} \in \mathbb{S}^d$ satisfy $\|\mathbf{M}\|_{\mathrm{op}} \leq R$. We can compute values $\{V_i\}_{i\in[n]}$ such that for all $i \in [n]$,*

$$|V_i - \langle \mathbf{A}_i, \exp(\mathbf{M}) \rangle| \leq \epsilon \langle |\mathbf{A}_i|, \exp(\mathbf{M}) \rangle + \gamma \mathrm{Tr}(|\mathbf{A}_i|) \mathrm{Tr}\exp(\mathbf{M})$$

*with probability $\geq 1 - \delta$ in time*

$$O\left( \mathcal{T}_{\mathrm{mv}}(\mathbf{M}) \cdot \frac{\sqrt{R}\log^{1.5}(\frac{Rnd}{\gamma\delta\epsilon})}{\epsilon^2} + \left( \sum_{i\in[n]} \mathcal{T}_{\mathrm{mv}}(\mathbf{A}_i) \right) \cdot \frac{\log\frac{nd}{\delta}}{\epsilon^2} \right).$$

*Proof.* We prove the result for a single $\mathbf{A}_i$ with failure probability $\delta \leftarrow \frac{\delta}{n}$ and then use a union bound to obtain the result. Denote $\mathbf{A} \leftarrow \mathbf{A}_i$. As in the proof of Lemma 19, assume that $\epsilon \leq \frac{1}{2}$ and we have obtained $\widehat{\lambda}$ such that $\lambda_{\max}(\mathbf{M}) \leq \widehat{\lambda} \leq \lambda_{\max}(\mathbf{M}) + 1$ with probability $\geq 1 - \frac{\delta}{2}$ within the allotted time, and define $\widetilde{\mathbf{M}} := \widehat{\lambda}\mathbf{I} - \mathbf{M}$. Also, by the spectral theorem there exists $\mathbf{B} \in \mathbb{R}^{d\times d}$ and a diagonal matrix $\mathbf{D} \in \mathbb{S}^d$ with diagonal entries in $\{\pm 1\}$ such that $\mathbf{A} = \mathbf{B}^\top \mathbf{D}\mathbf{B}$ and $\mathbf{B}^2 = |\mathbf{A}|$. Assume for $\mathbf{Q} \in \mathbb{R}^{k\times n}$ with $k = O(\frac{1}{\epsilon^2}\log\frac{d}{\delta})$ from Proposition 6, that for all $j \in [d]$,

$$\left| \left\| \mathbf{Q}\left[ \mathbf{B}\exp\left( -\frac{1}{2}\widetilde{\mathbf{M}} \right) \right]_{j:} \right\|_2^2 - \left\| \left[ \mathbf{B}\exp\left( -\frac{1}{2}\widetilde{\mathbf{M}} \right) \right]_{j:} \right\|_2^2 \right| \leq \frac{\epsilon}{e} \left\| \left[ \mathbf{B}\exp\left( -\frac{1}{2}\widetilde{\mathbf{M}} \right) \right]_{j:} \right\|_2^2.$$

Conditioning on these events gives the failure probability. Observe that

$$\left| \left\langle \mathbf{A}, \exp(-\widetilde{\mathbf{M}}) \right\rangle - \mathrm{Tr}\left( \mathbf{Q}\exp\left( -\frac{1}{2}\widetilde{\mathbf{M}} \right) \mathbf{A}\exp\left( -\frac{1}{2}\widetilde{\mathbf{M}} \right) \mathbf{Q} \right) \right|$$

$$= \left| \mathrm{Tr}\left( \exp\left( -\frac{1}{2}\widetilde{\mathbf{M}} \right) \mathbf{A}\exp\left( -\frac{1}{2}\widetilde{\mathbf{M}} \right) \right) - \mathrm{Tr}\left( \mathbf{Q}\exp\left( -\frac{1}{2}\widetilde{\mathbf{M}} \right) \mathbf{A}\exp\left( -\frac{1}{2}\widetilde{\mathbf{M}} \right) \mathbf{Q} \right) \right|$$

$$\leq \sum_{j\in[d]} \left| \left\| \mathbf{Q}\left[ \mathbf{B}\exp\left( -\frac{1}{2}\widetilde{\mathbf{M}} \right) \right]_{j:} \right\|_2^2 - \left\| \left[ \mathbf{B}\exp\left( -\frac{1}{2}\widetilde{\mathbf{M}} \right) \right]_{j:} \right\|_2^2 \right| \tag{28}$$

$$\leq \frac{\epsilon}{4e} \sum_{j\in[d]} \left\| \left[ \mathbf{B}\exp\left( -\frac{1}{2}\widetilde{\mathbf{M}} \right) \right]_{j:} \right\|_2^2 = \frac{\epsilon}{e} \left\langle |\mathbf{A}|, \exp(-\widetilde{\mathbf{M}}) \right\rangle.$$

The first inequality above used the decomposition $\mathbf{A} = \mathbf{B}^\top \mathbf{D}\mathbf{B}$ and the triangle inequality, and the second inequality used our assumption on $\mathbf{Q}$. Next, we apply Proposition 5 with accuracy parameter $\frac{\gamma}{3e}$ to obtain a polynomial $p$ of degree $O(\sqrt{R\log\frac{1}{\gamma}})$ such that for $\mathbf{E} := p(\frac{1}{2}\widetilde{\mathbf{M}}) - \exp(-\frac{1}{2}\widetilde{\mathbf{M}})$,

$$\|\mathbf{E}\|_{\mathrm{op}} \leq \frac{\gamma}{3e}.$$

Let $q$ be some row of $\mathbf{Q}$ with $\|q\|_2^2 = \frac{1}{k}$, and let $u = \mathbf{E}q$ so that $\|u\|_2 \leq \frac{\gamma}{3e\sqrt{k}}$. We bound

$$\left| q^\top p\left( \frac{1}{2}\widetilde{\mathbf{M}} \right) \mathbf{A}p\left( \frac{1}{2}\widetilde{\mathbf{M}} \right) q - q^\top \exp\left( -\frac{1}{2}\widetilde{\mathbf{M}} \right) \mathbf{A}\exp\left( -\frac{1}{2}\widetilde{\mathbf{M}} \right) q \right| \leq 2\left| q^\top \exp\left( -\frac{1}{2}\widetilde{\mathbf{M}} \right) \mathbf{A}u \right|$$

$$+ \left| u^\top \mathbf{A}u \right|.$$

We further may upper bound

$$\left| u^\top \mathbf{A} u \right| \le \|\mathbf{B}u\|_2^2 \,, \quad \left| q^\top \exp\left(-\frac{1}{2}\widetilde{\mathbf{M}}\right) \mathbf{A} u \right| \le \|\mathbf{B}u\|_2 \left\| \mathbf{B}\exp\left(-\frac{1}{2}\widetilde{\mathbf{M}}\right) q \right\|_2 \,,$$

by the triangle inequality and Cauchy-Schwarz inequality. By positive semidefiniteness of $\widetilde{\mathbf{M}}$ we have $\| \exp(-\frac{1}{2}\widetilde{\mathbf{M}})q\|_2 \le \|q\|_2$, and hence

$$\left\| \mathbf{B}\exp\left(-\frac{1}{2}\widetilde{\mathbf{M}}\right) q \right\|_2^2 \le \left\langle qq^\top, |\mathbf{A}| \right\rangle \le \frac{1}{k}\mathrm{Tr}|\mathbf{A}|,$$

$$\|\mathbf{B}u\|_2^2 = \left\langle uu^\top, |\mathbf{A}| \right\rangle \le \frac{\gamma^2}{9e^2 k}\mathrm{Tr}|\mathbf{A}|.$$

Combining the above three displays, and summing over each row of $\mathbf{Q}$, we obtain

$$\left| \mathrm{Tr}\left( \mathbf{Q}\exp\left(-\frac{1}{2}\widetilde{\mathbf{M}}\right) \mathbf{A}\exp\left(-\frac{1}{2}\widetilde{\mathbf{M}}\right) \mathbf{Q} \right) - \mathrm{Tr}\left( \mathbf{Q}p\left(\frac{1}{2}\widetilde{\mathbf{M}}\right) \mathbf{A}p\left(\frac{1}{2}\widetilde{\mathbf{M}}\right) \mathbf{Q} \right) \right| \le \frac{\gamma}{e}\mathrm{Tr}|\mathbf{A}|. \tag{29}$$

We choose to return $V = \exp(\widehat{\lambda})\mathrm{Tr}(\mathbf{Q}p(\frac{1}{2}\widetilde{\mathbf{M}})\mathbf{A}p(\frac{1}{2}\widetilde{\mathbf{M}})\mathbf{Q})$. The approximation guarantee follows from combining (28) and (29), and multiplying by $\exp(\widehat{\lambda}) \le e\mathrm{Tr}(\exp(\mathbf{M}))$. The runtime comes from applying $p(\frac{1}{2}\widetilde{\mathbf{M}})$ to each row of $\mathbf{Q}$ first, and then computing $k$ quadratic forms through $\mathbf{A}$. We note that the applications of $p(\frac{1}{2}\widetilde{\mathbf{M}})$ to rows of $\mathbf{Q}$ can be precomputed, and reused for all $\{\mathbf{A}_i\}_{i\in[n]}$. $\quad\square$