# OpenReview forum: "Revisiting Area Convexity: Faster Box-Simplex Games and Spectrahedral Generalizations"
_NeurIPS.cc/2023/Conference — NeurIPS 2023 poster_

### Official Review · Reviewer_Cfry · 2023-07-03

**Soundness:** 3 good
**Presentation:** 3 good
**Contribution:** 2 fair
**Rating:** 4
**Confidence:** 2

**Summary:**

This paper focuses on box-simplex games that are min max of degree 2 polynomials being bilinear in the inner and outer optimization variables. The inner problem is over the simplex and the outer problem is over the unit hypercube. The goal of the paper is to unify the previous framework derived in [She17] with standard Bregman divergence domination conditions.
This latter framework [She17] consists on exploiting the primal-dual nature of the min-max problem to setup a regularizer over the unit hypercube. This satisfies the so-called "area convexity" condition instead of strong convexity. The outer loop method relies on an extragradient method and the inner loop relies on an alternating minimization sub-procedure.
In order to analyze this framework, the authors explain the connections with relative smoothness and extragradient methods.
This yields an algorithm to solve box-simplex games that improves the one of [She17] by a log factor.


**Strengths:**

The paper is well-written and I believe quite clear for non-experts in this field.
The authors are willing to bridge the gap between area convexity and more standard convexity tools, that makes the contribution original.
The proof sketches sound correct.


**Weaknesses:**

I am not a specialist in this field and with a very limited amount of time (too short to read the 39 pages of supplementary material), it is completely impossible to judge whether the framework is correct or not. The authors stated the main theorems and did provide proof sketches and hints that could possibly convince some readers. I have doubts that this work, possibly sound and surely interesting, could be a good fit with NeurIPS.

There are neither concrete algorithmic implementation nor benchmarking in the main submission document.


**Questions:**

- after l72 (2): could you please define Delta^(dxd) (the set of trace 1 PSD matrices)?

- In (4) what is h?


**Limitations:**

There is no limitation section provided by the authors in the main submission.
The authors do not address any limitation and do not provide any further research directions.

---

> ### Author Rebuttal · Authors · 2023-08-07
>
> Thank you for your reviewing efforts. We are happy to hear that you found our paper well-written and interesting, and that you felt it was understandable for non-experts.
>
> We would like to note to the reviewer that the format and scope of our paper, which addresses a fundamental theoretical problem of interest in machine learning applications, is similar to many of the theoretical papers which have appeared in past NeurIPS conferences, and we believe it is a good fit for the venue. If the reviewer has specific concerns about the correctness of any portion of our paper, we are of course happy to answer any questions or provide an explanation. Our proofs of correctness are self-contained and provided in full in the supplement.
>
> Regarding practical implementations, our main motivation in this work was clarifying and improving a theoretical tool, area convexity, which has found use in practical problems we believe are of value to the NeurIPS community (optimal transport, matching, min-mean-cycle, etc.), and showing how the same tool extends to solve semidefinite programming variants. While promising empirical evaluations of area convex algorithms for solving these applications on real-world examples have been conducted in prior works (see e.g. [JST19]), we agree that an important next step is to benchmark our improved area convex algorithm against them.
>
> The set name $\Delta^{d \times d}$ is reserved for the spectraplex, and the function name $h$ is reserved for entropy; both are defined in Section 1.3, but we will make this more clear when they appear. We agree that the inclusion of $\Delta^{d \times d}$ in (2) before it is defined is an oversight which we will address; thank you for pointing this out.
>
> Regarding limitations and extensions, we give a brief outline of how future work can improve Theorem 2 by using low-rank sketching tools after its statement (with an extended discussion in Section 1.2 of the supplement), but agree this can be more explicitly stated. We also believe Theorem 1 may be optimal up to constant factors due to the lack of progress in the historically easier simplex-simplex case (see Footnote 6), which Theorem 1 matches; however, proving a formal lower bound is an important future endeavor. Moreover, area convex algorithms appear to be specialized to bilinear settings such as linear and semidefinite programming currently; finding nonlinear generalizations (as suggested by another reviewer) would be an exciting extension of these tools. Finally, finding more applications and improving practical implementations of our algorithms are important extensions of this line of research. We will add these points in a revision of our paper; thank you for raising them.
>
> We hope this discussion was clarifying to the reviewer and improves your opinion of our paper.

---

> > ### Comment · Reviewer_Cfry · 2023-08-17
> >
> > Thank you for your responses, I will retain my score.

---

### Official Review · Reviewer_Vb7x · 2023-07-04

**Soundness:** 4 excellent
**Presentation:** 3 good
**Contribution:** 3 good
**Rating:** 7
**Confidence:** 3

**Summary:**

The authors study first-order algorithms for box-simplex games, a special kind of two-player zero-sum bi-affine constrained games. The constraints of these games dictate that the first player selects an action represented by a vector within the n-dimensional box ($[-1,1]^n$), while the second player chooses an action defined by a vector from the d-dimensional simplex ($\Delta_d$).

The authors show an algorithm for the computation of an $\epsilon$-approximate saddle point of a box-simplex bi-affine game, expressed as $min_{x\in[-1,1]^n,y\in\Delta_d} x^T A y^T - <b,x> + <c,y>$, in time $O(nnz(A)\cdot L \cdot\log(d)/\epsilon)$, where L is the maximum 1-norm of any column of A, and nnz(A) denotes the number of non-zero entries in matrix A. They advance over the previously best-known first-order method by Sherman, removing logarithmic terms.

Box-simplex games has some important applications, and the authors' algorithm introduces a new state-of-the-art, including applications like approximate max flow and optimal transport.

In the latter part of the paper, the authors broaden their focus to include a new class of two-player games: box-spectraplex games. This is a generalization of box-simplex games where one player still picks an action in a n-dimensional box ($[-1,1]^n$), but the other player chooses as an action, a positive semi-definite matrix Y with trace equals to 1. An applications of these kind of games include spectral sparcification. The authors present a first-order algorithm to computing an approximate saddle point of a box-spectraplex game.


After rebuttal: The author's rebuttal successfully addressed my concerns, and as a result, I have chosen to maintain my original score.

**Strengths:**

This paper studies an interesting class of games with important applications, thereby contributing to the current state-of-the-art algorithm. Moreover, the authors novel reinterpretation of the area-convexity as the more standard relative smoothness allows for the use of more conventional analysis methods, which is interesting on its own. I looked at the statement of the lemmas and they seem reasonable. I only verified the proofs in the main body.

**Weaknesses:**

In the spirit of constructive criticism, I believe that the writing could be improved for better readability. Currently, the appendix appears more like a journal version of the paper rather than serving its intended purpose. A few recommendations to enhance the paper include:
- On line 63, defining the notation nnz(A) and $||A||_{1\rightarrow 1}$ earlier might improve clarity.
- On lines 105-16, it could be beneficial to define the operator $T_{mv}$ sooner.
- Several references to 'outer' and 'inner loop' throughout the paper lacked clarity. Lines 38, 215, and 224 are examples of this.

Finally, the move towards box-spectraplex games seems to be more theoretical than practical, with limited application examples. Please see the question section.

**Questions:**

Could the authors provide more examples of problems that could be simplified to box-spectraplex games?

I was a bit confused in the proof sketch of Theorem 1. For the telescoping sum to be valid, it appears necessary that $\bar{y}_{t+1}$ must equal $\bar{y}^+_t$ (where $\bar{y}_t$ represents the input to the extra gradient step-oracle at step t, and $\bar{y}^+_t$ indicates the output of the extra gradient oracle at step t). However, upon reviewing Algorithm 3 in the supplementary material, different notation seems to be used. Could the authors provide clarification on this?

**Limitations:**

The authors list any assumptions for their theorems, and I do not believe this work can have negative societal impact.

---

> ### Author Rebuttal · Authors · 2023-08-07
>
> Thank you for your insightful feedback; we appreciate that you found both the problems we study and our insights to be interesting. We agree with your suggestions on presentation, and discussed some potential directions for incorporating them in the global response. We are happy to incorporate any of your further suggestions. We also agree with your notation and naming convention suggestions, and will address these in a revision.
>
> Regarding Theorem 1, indeed, $\bar{y}_{t + 1}$ is set to be the third argument of the output of Algorithm 2 (denoted $\bar{y}^+$) in each iteration. In Algorithm 3 in the supplement, we performed this computation explicitly in terms of the inputs of Algorithm 2, which can be verified to be consistent. However, we agree it would be more clear to have a “conceptual” variant of Algorithm 3 which simply explains how the points in the analysis of Theorem 1 are derived from the abstract oracles in Definitions 4 and 5 of the main body, and will provide this in a revision.
>
> Regarding further applications of box-spectraplex games, we note that the community has previously not provided any nontrivial algorithms for this problem beyond black-box applications of non-tailored methods (or even defined it explicitly to the best of our knowledge), likely due to the lack of progress even in the box-simplex case until recently [She17]. We find it encouraging that recent work [JRT23] was already able to use our result to obtain improvements for a fundamental problem, and are hopeful that other previous applications of positive SDP solvers (as outlined in Page 3 of the main submission) may also be amenable to our tools. At the moment we do not have other examples of applications of box-spectraplex games, but we believe this is outside the scope of our submission (whose goal was to provide the first near-linear time solver), and consider this an exciting direction opened up by our result.

---

> > ### Comment · Reviewer_Vb7x · 2023-08-14
> >
> > Thank you for the clarification. I'm pleased with your response and will retain the score I've assigned.

---

### Official Review · Reviewer_E7hD · 2023-07-07

**Soundness:** 3 good
**Presentation:** 1 poor
**Contribution:** 2 fair
**Rating:** 5
**Confidence:** 4

**Summary:**

This paper looks at the bilinear min-max problem Area Convexity framework proposed by Sherman through the lens of Relative Lipschitzness condition of Cohen et al. which relates to the standard Bregman analysis in mirror-type algorithms in first-order optimization. Leveraging this connection, they provide the standard analysis of two-timescale Extragradient method for area convex regularizer based Bregman divergence. Tuning a parameter in the analysis and using the structure of the area convex regularizer, they implement an approximate version of Extragradient where each approximate iteration requires O(nnz(A)) computations. This allows them to remove the log(1/\eps) factor Sherman incurred in the analysis of inner loop. The extend similar result to the case where

**Strengths:**

The paper had a new ideas. E.g., tuning parameter alpha/beta in the regularizer was critical step that allowed for approximate implementation as done in Line 5-9 and Line 12-17 of Algorithm 3.

**Weaknesses:**

The main weakness of the paper is in its presentation. The way main paper differs from the supplementary almost feels like reading a new paper with different orders, equation numbers etc. Also, there are concepts introduced which may not be directly relevant to the problem at hand: E.g. Definition 1 is not used throughout the main paper. I had to dig until Appendix B to understand why there was a fuss about it on the very first technical section, i.e., Section 2.1.
The paper borrows key ideas developed by Sherman (which are popular by now) and some previous developments such as Lu et al, and Cohen et al. In particular, the idea of Area Convexity is very much exclusively used for bilinear games and there has been no major movement towards general convex-concave problems. In that sense, this is an incremental work in my opinion.


**Questions:**

My questions below are referring to the supplementary material (or actual paper).

1. It would be best if you can explain how steps 7,8,9 relate to steps in Algorithm 1 with exact parameters used for it. Leaving it as "By observation" may not work for large fraction of readers. Same goes for steps 14-17. It would also be worthwhile to remind the reader that h is the entropy divergence function used here.
2. The proof of Lemma 6 needs to deliberate more on what F to use define f(y-hat) in (13) such that Lemma 4 can be applied.
3. Use x_br for two different purposes in the same section is a bad notation. In particular, Lemma 4 defines x_br as argmin F(x,y) for F defined Section 2.2 of the main paper (again, I could not find it in the supplementary paper). Is this the same used in Lemma 6?
4. Remind the reader about H at the beginning of Section 4.
5. In the Proof of Lemma 8, you use Theorem 3.3 of LS01. It would be better if you provide the result as there is already quite a bit of jumping around while reading the paper. I haven't been able to verify the claims in the proof of this lemma.
6. I have read some of the references in the previous paper of this area. Can you also comment BSW19 paper? As I understand, they also implement Shermans MWU algorithm for inner subproblems and hence incur log(1/\eps) additional factor similar to Sherman or JST19.
7. Can you also compare the result for box-spectraplex case with BBN13, AL17, CDST19? In particular, it will be better to state clearly in what regimes Algorithm in Section 4.4 would be better than box-spectraplex variations of the above papers.


BSW19: Faster width-dependent algorithm for mixed packing and covering LPs

**Limitations:**

I am giving this paper borderline accept since it was not very well written and almost seems like reading two papers at a time which borrows/uses existing literature heavily (sometimes without giving context). If properly written, this paper can be improved substantially. But that will require significant rewriting and hence, I don't see how it will be justified in this conference set up.

---

> ### Author Rebuttal · Authors · 2023-08-07
>
> Thank you for your careful reviewing efforts and many helpful comments. We agree with the revisions suggested by your Questions 1, 2, 3, 4, and 5, and will incorporate them. Regarding Question 1, in the main body of our revision, we will include an “abstract” variant of Algorithm 3 in the supplement which calls Definitions 4 and 5 directly and precisely states what parameters are used in the calls (these parameters are stated in the proof of Theorem 1, though we agree they can be made more explicit). Regarding Question 2, the function $F$ in the application of Lemma 4 is the joint function of $x$ and $y$ defined in (11) of the supplement (before minimizing over $x$), but we will clarify this. Regarding Question 3, we agree the overloaded notation is confusing and this will be clarified; the use of notation $x_{\text{br}}$ in Lemma 6 is  consistent with the use in Lemma 4 with a specific choice of $F$. Regarding Question 5, we will explain in self-contained terms the result from [LS01], which is that the gradient of a spectral function (on matrices) agrees with its vector gradient on the eigenvalues.
>
> Regarding Question 6, we agree that [BSW19] is a relevant paper to this line of work and will include a citation, and that it suffers from the same logarithmic overhead in runtime due to its reliance on alternating minimization as in [She17]. In the process of completing this work, we actually reached out to the authors of [BSW19] because they did not include a proof of correctness that their alternating minimization converges, which we had some doubts about because their setup has different properties than [She17]. They have not yet gotten back to us with a proof of correctness, so we were unsure how to address this point in the submission.
>
> Regarding Question 7, consider the problem of solving a box-spectraplex game with Lipschitz constant $L$, as defined in Theorem 2 of our paper. Compared to [BBN13, AL17, CDST19], our algorithm incurs an overhead of $L/\epsilon$ in the runtime, but saves a factor of $d^2$ (the dimension of the matrices) due to the size of $\ell_1$-strongly convex regularizers over the $\ell_\infty$ ball (the box, as opposed to the simplex which [BBN13, AL17, CDST19] are tailored to handle). By using “half-regularized” strategies as discussed in the introduction of [She17], we believe this $d^2$ overhead can be improved to a $d$, but the analysis of low-rank sketches used in [BBN13, AL17, CDST19] becomes more challenging. In summary, we lose a $L/\epsilon$ factor but save at least a $d$ (and possibly a $d^2$ factor) over these methods, which is favorable in the setting where first-order methods are preferred over e.g. interior point methods in the first place (when $L/\epsilon$ is small compared to $d$).
>
> We acknowledge your concerns on readability and will take steps to address them, summarized in our global response. We believe our conceptual contributions and algorithmic improvements can be understood from the shortened version of our paper, and thus our work is appropriate for publication at NeurIPS; we hope that aligning our presentation more closely between the shortened version and the supplement will help bridge the gap in readability. Regarding the incrementality of our results, we conjecture our Theorem 1 is optimal within a constant factor as it matches state-of-the-art deterministic algorithms in the easier simplex-simplex setting (which have not been improved in 20 years) under a more challenging problem geometry; we hence believe there is further merit in closing this gap. Finally, the tools we introduce to obtain our result generalize readily to the box-spectraplex setting, overcoming obstacles from prior analyses as explained in Section 2.2.
>
> We hope this discussion clarifies and elevates the merits of our paper to you. Thank you again for all the detailed feedback from a reviewer familiar with the line of work on area convexity.

---

> > ### Comment · Reviewer_E7hD · 2023-08-17
> >
> > Thank you for your reply. After carefully reading your response, I have decided to retain my score.

---

### Official Review · Reviewer_Wg5p · 2023-07-23

**Soundness:** 3 good
**Presentation:** 2 fair
**Contribution:** 3 good
**Rating:** 7
**Confidence:** 3

**Summary:**

In this paper, the authors consider box-simplex games, a bilinear min-max optimization problem with box and simplex constraints. The current best-known method for solving such problems is Sherman's algorithm, which is based on the concept of "area convexity". The key insight of this work is to reinterpret area convexity as a more general notion of relative Lipschitzness in the optimization literature. Using this new perspective, they streamline the proof of Sherman's algorithm, propose an improved subproblem solver that eliminates the additional log overhead, and further extend the algorithm to box-spectraplex games.

**Strengths:**

- The authors show that area convexity implies a Bregman divergence bound similar to relative Lipschitzness, which is a simple yet insightful observation. I appreciate that it better clarifies the connection between Sherman's algorithm and the classical extragradient methods, making it conceptually easier to understand and extend.
- The proposed subproblem solver improves a $\log(1/\epsilon)$ factor from the original Sherman's algorithm. As a corollary, it leads to the state-of-the-art runtime complexity for optimal transport among first-order algorithms.

**Weaknesses:**

I only wish the presentation of the paper could be more clear. In particular, the proposed extragradient algorithm is not fully described in the main text, but only implicitly in the proof of Theorem 1. It would be better if the authors can present the pseudocode and better explain the steps in the algorithm. Also, it is a bit confusing that the supplement is an extended version of the paper rather than the appendix, which makes it a bit hard to track down the proof.

**Questions:**

- In Definition 4, the notation $V_z^{\alpha}$ is undefined.
- If I understand correctly, both the GradStepOracle and XGradStepOracle are designed for solving the same kind of subproblem (at the bottom of page 6). Could you explain why they are implemented differently? And how are they related to the algorithm in [LFN18]?

**Limitations:**

The authors addressed the limitations by specifying the considered problem class.

---

> ### Author Rebuttal · Authors · 2023-08-07
>
> Thank you for your encouraging comments, and we are glad that you found our technical contributions insightful. We agree with your suggestions regarding the supplement, and outline some directions for improvement in the global response (though we are happy to incorporate any further suggestions as well). Regarding pseudocode for Theorem 1, our exclusion was largely due to space restrictions (in particular, Algorithm 3 in the supplement contains full pseudocode for our algorithm). However, we will provide a more explicit explanation before the proof for clarity. To our understanding the final conference version would have an additional allowed page, so if our paper is accepted we will use it to add clarity to this section of the paper.
>
> Thank you for pointing out our oversight in Definition 4, which we will also address in a revision.
>
> Regarding your question on asymmetry in the steps of our method, this reflects an asymmetry in the analyses of extragradient methods. In these, the regret is bounded for the “gradient oracle” points, but the regret upper bound is stated in terms of the divergences of the “extragradient oracle” points (which our inexact oracles need to compensate for). Our oracle implementations are not directly related to the algorithm in [LFN18], which was designed for iterative convex minimization (as opposed to our method, a one-shot subproblem solver for a minimax optimization problem), but certainly our analysis is inspired by theirs through the commonality of the tool we use (relative smoothness). We will add this discussion for clarity in a revision as well.

---

> > ### Comment · Reviewer_Wg5p · 2023-08-14
> >
> > I thank the authors for their response. Please make sure to incorporate the changes in the revision.

---

### Official Review · Reviewer_mAce · 2023-07-24

**Soundness:** 3 good
**Presentation:** 2 fair
**Contribution:** 3 good
**Rating:** 5
**Confidence:** 3

**Summary:**

This paper explores the relationship between area convexity and extragradient methods, and provides improved solvers for subproblems required by variants of the algorithm. The paper also presents a state-of-the-art first-order algorithm for solving box-simplex games and a near-linear time algorithm for a matrix generalization of box-simplex games. The authors demonstrate that their algorithms improve runtimes for combinatorial optimization problems and provide new insights into numerical linear algebra. The contributions of the paper include the development of efficient algorithms with improved convergence rates and computational efficiency, as well as the application of these algorithms to various optimization problems in fields such as optimal transport and min-mean-cycle.

**Strengths:**

Through a deeper understanding of the relationship between box-simplex games and matrix generalization problems, the authors propose improved solvers that leverage the lens of relative smoothness and convex analysis to design efficient algorithms with faster runtimes. These proposed algorithms not only improve runtimes for combinatorial optimization problems but also provide new insights into numerical linear algebra and have applications in various combinatorial optimization problems such as approximate maximum flow, optimal transport, and min-mean-cycle.

In addition to its technical contributions, the paper is well-written, well-structured, and effectively communicates its main ideas and contributions. The authors present a rigorous analysis of the proposed algorithms, complete with detailed proofs, explanations, and appropriate use of mathematical notation. The inclusion of appendix sections, examples, illustrations, and a comprehensive literature review further enhance the quality, readability, and accessibility of the work.

Overall, the strengths of this paper lie in its thorough exploration of area convexity, its innovative use of relative smoothness, and the development of improved algorithms for solving optimization problems.


**Weaknesses:**

1.	The author claims to have improved the runtime for the Box-simplex problem, resulting in increased performance of related applications. However, a detailed comparison with previous results is not provided. Providing such a comparison would help readers better understand the level of improvement and contribution of the paper.
2.	The paper mentions the use of a proximal oracle in Algorithm 6, but more details on its implementation and computational complexity would be helpful. It would be valuable for the authors to discuss how the proximal oracle contributes to the overall efficiency of the algorithm and whether there are any limitations or challenges in its practical implementation.
3.	The paper could benefit from experimental evaluation to validate the proposed approach. Conducting experiments on real-world or synthetic datasets would provide empirical evidence of the effectiveness and efficiency of the alternating minimization scheme.


**Questions:**

Regarding Weakness 1: The author claims to have improved the runtime of multiple applications by solving the Box-simplex problem. However, a detailed comparison with previous results is not provided. Providing such a comparison would help readers better understand the level of improvement and contribution of the paper.

Regarding Weakness 2: The paper mentions the use of a proximal oracle in Algorithm 6. Could the authors provide more details on the implementation and computational complexity of this oracle? It would be helpful to understand how the proximal oracle contributes to the overall efficiency of the algorithm and whether there are any limitations or challenges in its practical implementation.


**Limitations:**

The paper does not discuss the limitations and any potential negative consequences of their work. However, since their algorithm is focused on optimization problems, the impact on society would largely depend on the specific applications and use cases. A brief discussion on the potential real-world implications and ethical considerations would provide valuable context for readers to better understand the broader implications of the research.

---

> ### Author Rebuttal · Authors · 2023-08-07
>
> Thank you for your careful reviewing efforts. We are glad that you found our technical contributions interesting and our paper easy to read.
>
> Regarding prior box-simplex game solvers, our paper is most directly comparable to [She17], which it improves upon by a logarithmic factor in the runtime by removing the need for high-accuracy alternating minimization. The [She17] result was viewed as a breakthrough in the optimization community (and has not been improved since prior to our work), as previously for box-simplex games, first-order algorithms incurred an additional overhead of $\min(\epsilon^{-1},\sqrt{d})$ in the iteration count. Higher-order algorithms such as interior point methods also incurred polynomial overhead in runtime due to the need to solve linear systems. We agree this comparison is valuable context and not explicit, and will add it in a revision. We also note that we summarize our new results for applications in Section 5 of the supplement, though there too we will add a more explicit comparison to prior work (in all cases, we remove an extraneous log factor).
>
> While the improvement of Theorem 1 over [She17] is somewhat modest, it improves the runtime of box-simplex games to be comparable to state-of-the-art deterministic algorithms for the easier simplex-simplex setting (which have not been improved in 20 years) under a more challenging problem geometry, so we conjecture it is optimal up to constant factors; we believe closing this gap has additional merit. Finally, the tools we introduce to obtain our result generalize readily to the box-spectraplex setting, overcoming obstacles from prior analyses as explained in Section 2.2.
>
> Regarding Algorithm 6, we note that this is just a conceptual framework for extragradient algorithm design we introduce, and for specific problems care must be taken to handle any inexactness in the proximal oracle implementation. For the box-simplex and box-spectraplex game applications we study, we follow the framework of Algorithm 6 and provide end-to-end inexactness analyses and (near-linear) runtime guarantees for our results.
>
> Regarding practical implementations, our main motivation in this work was clarifying and improving a theoretical tool, area convexity, which has found use in practical problems we believe are of value to the NeurIPS community (optimal transport, matching, min-mean-cycle, etc.), and showing how the same tool extends to solve semidefinite programming variants. While promising empirical evaluations of area convex algorithms for solving these applications on real-world examples have been conducted in prior works (see e.g. [JST19]), we agree that an important next step is to benchmark our improved algorithm against them.
>
> We hope this response was clarifying, and elevates your opinion of our paper. Thank you again for your helpful suggestions, and we will take care to incorporate them in future versions.

---

> > ### Comment · Reviewer_mAce · 2023-08-11
> >
> > Thank you for your detailed response. I have carefully evaluated your response, as well as the feedback provided by other reviewers. After careful consideration, I have decided to maintain my current score.

---

### Author Rebuttal · Authors · 2023-08-07

Reviewers Wg5p, E7hD, and Vb7x asked about inconsistencies between our main submission and supplementary material which hindered readability. We thank you for raising this important concern and completely agree; we will make efforts in a revision to make the two more consistent. Specifically, after all deferred proofs or contents in the main submission, we will add an explicit pointer to where the corresponding part of the supplement can be found. We will also order the two parts in a more compatible way (trying to preserve theorem names, etc. whenever possible), and make the main submission more self-contained by eliminating unused parts.

---

### Decision · Program_Chairs · 2023-09-21

**Decision:**

Accept (poster)

**Comment:**

The paper provides new insights into area convexity, and connects it to more standard gradient descent analyses. Building on these insights, the paper obtains modest but meaningful improvements in the running times for well-motivated combinatorial problems. There were concerns about the presentation of the paper and whether the algorithmic framework is correct and theoretically sound, which were sufficiently addressed by the author response and the subsequent discussion.